

# Chemical characterization and source apportionment of submicron aerosols measured in Senegal during the 2015 SHADOW campaign

Laura-Hélèna Rivellini[1,2] [*], Isabelle Chiapello[2], Emmanuel Tison[1], Marc Fourmentin[3], Anaïs Féron[4], Aboubacry Diallo[5], Thierno N'Diaye[5], Philippe Goloub[2], Francesco Canonaco[6], André Stephan Henry Prévôt[6], and Véronique Riffault[1] [*]

[1]IMT Lille Douai, Univ. Lille, SAGE - Département Sciences de l'Atmosphère et Génie de l'Environnement, 59000 Lille, France
[2]Laboratoire d'Optique Atmosphérique, Université de Lille - CNRS, Villeneuve d'Ascq, 59655, France
[3]Laboratoire de Physico-Chimie de l'Atmosphère, Université du Littoral Côte d'Opale, Dunkerque, 59140, France
[4]Laboratoire Interuniversitaire des Systèmes Atmosphériques, CNRS - Université Paris Est Créteil - Université Paris Diderot, Créteil, 94010, France
[5]Institut de Recherche pour le Développement, M'Bour, Senegal
[6]Laboratory of Atmospheric Chemistry, Paul Scherrer Institute, 5232 Villigen, Switzerland

[*] *Correspondence to*:

Véronique Riffault
Tel.: +33 327 712 604, Fax: +33 327 712 914, e-mail: veronique.riffault@imt-lille-douai.fr

Laura-Hélèna Rivellini
e-mail: laura.rivellini@imt-lille-douai.fr

**Abstract.** The present study offers the first chemical characterization of the submicron ($PM_1$) fraction in West Africa at a high time resolution, thanks to collocated measurements of non-refractory (NR) species with an Aerosol Chemical Speciation Monitor (ACSM), black carbon and iron concentrations derived from absorption coefficient measurements with a 7-wavelength aethalometer, and total $PM_1$ determined by a TEOM-FDMS for mass closure. The field campaign was carried out during four months (March to June 2015) as part of the SHADOW (SaHAran Dust Over West Africa) project at a coastal site located in the outskirts of the city of M'Bour, Senegal. With an average mass concentration of 5.4 µg m$^{-3}$, levels of NR-$PM_1$ in M'Bour were three to ten times lower than cities like Paris or Beijing. Nonetheless the first half of the observation period was marked by intense but short pollution events (concentrations higher than 15 µg m$^{-3}$), sea breeze phenomena and Saharan desert dust outbreaks ($PM_{10}$ up to 900 µg m$^{-3}$). During the second half of the campaign, the sampling site was mainly under the influence of marine air masses. The air masses on days under continental and sea breeze influences were dominated by organics (36-40%), whereas sulfate particles were predominant (40%) for days under oceanic influence. Overall, measurements showed that about 3/4 of the total $PM_1$ were explained by NR-$PM_1$, BC and Fe (a proxy for dust) concentrations, leaving ~1/4 for other refractory species. A mean value of 4.6% for the Fe/$PM_1$ ratio was obtained. Source apportionment of the organic fraction, using Positive Matrix Factorization (PMF) highlighted the impact of local combustion sources, such as traffic and residential activities, which contribute on average to 52% of the total organic fraction. A new





organic aerosol (OA) source, representing on average 3% of the total OA fraction, showed similar variation as non-refractory particulate chloride. Its rose plot and daily pattern pointed out to local combustion processes, that is to say two open waste burning areas located about 6 and 11 km away from the receptor site and to a lesser extent a traditional fish smoking place. The remaining fraction was identified as oxygenated organic aerosols (OOA), a factor

that prevailed regardless of the day type (45%) and was representative of regional but also local sources due to enhanced photochemical processes.

## 1 Introduction

Atmospheric aerosols play a key role in the Earth's radiative forcing, by interacting with incoming solar and

outgoing terrestrial radiations (direct effect) and influencing cloud formation, growth and lifetime (indirect effects). Important uncertainties related to such effects remain, as reported in the IPCC reports (IPCC, 2007, 2013). Besides, several epidemiological and toxicological studies (Brook et al., 2004; Kelly and Fussell, 2012) also highlight the sanitary impacts of particulate matter (PM) depending on their size, chemical composition and exposure time. In 2012, around 3.7 million deaths were attributed to cardiovascular and respiratory diseases caused by outdoor PM

exposure (Brauer et al., 2012). As a response, the World Health Organization (WHO) established in 2006 air quality thresholds for PM: a daily average of 25 (respectively 50) $\mu g\ m^{-3}$ and a mean annual limit of 10 (respectively 20) $\mu g\ m^{-3}$ for $PM_{2.5}$ (PM with diameter < 2.5 $\mu m$) ($PM_{10}$) (WHO, 2006). During past decades, high-time resolution monitoring instruments have been implemented in Europe within the ACTRIS (Aerosols, Clouds, and Trace gases Research Infrastructure network) program, in America with the IMPROVE (Interagency Monitoring of Protected

Visual Environments) network (Prenni et al., 2016; Schurman et al., 2015) or more recently in Asia, in order to better characterize fine particles. Yet, the African continent remains poorly documented in terms of particulate pollution. Several studies in West Africa (mostly sub-Saharan regions) have focused on the coarse fraction, since this area is strongly influenced by natural sources, especially the Sahara desert that injects high amounts of mineral aerosols into the atmosphere (e.g Chiapello et al., 1995). Among field campaigns, a large effort was performed on aerosol

measurements during the AMMA (African Monsoon Multidisciplinary Analysis) program carried out between 2002 and 2010, with intensive observation periods in 2006 (Redelsperger et al., 2006). During the dry season (extending from November to May) the transport of desert dust (DD) and biomass burning (BB) aerosols were observed to occur into two layers, one dominated by DD close to the surface (< 1 km), the one containing BB aerosols being located at a higher altitude, between 2 and 4 km (Haywood et al., 2008; Osborne et al., 2008). During one of the

AMMA special observing periods (SOP) (February 2006), a chemical characterization of particles at the ground level was performed near M'Bour (Senegal) using filter sampling and individual particle analysis (Deboudt et al., 2010; Flament et al., 2011). The analysis evidenced both internal and external mixing of DD, sea salt (SS) and carbonaceous aerosols (Deboudt et al., 2010) within the surface layer. The compositions of the coarse (2-10 $\mu m$) and fine (< 2 $\mu m$) fractions were established at the surface; DD dominated the coarse fraction while organic matter was

the major constituent of the fine one. Similar analysis were performed on samples collected during aircraft flights (Formenti et al., 2008a) to determine the chemical composition of aerosols at different altitudes. The altitude layer




was mainly composed of BB and DD external mixture (Chou et al., 2008; Hand et al., 2010). From January to March 2006 (AMMA SOP-0), several types of events from mixtures of DD and BB aerosols to pure mineral dust events were identified based on column aerosol retrievals from AERONET sun/sky photometer measurements in M'Bour, Senegal (Derimian et al., 2008). Other field campaigns such as POLCA ("POLlution des Capitales Africaines" on the

Pollution of African Capitals) highlighted the presence of anthropogenic sources of aerosols linked to the strong demographic growth, limited traffic pollution regulations and traditional activities such as slash-and-burn cultivation that generates huge amounts of BB aerosols (Liousse et al., 2010). Its aim was rather on the chemical characterization of $PM_{2.5}$, especially Black Carbon (Doumbia et al., 2012), and the impact on health (Val et al., 2013). Nevertheless, to the best of our knowledge, only one other study has performed on-line and real-time

chemical characterization of major $PM_1$ constituents on the African continent. It was conducted during one year in Welgegund, South Africa, at a site influenced by anthropogenic sources and devoid of mineral dust or marine contribution (Tiitta et al., 2014). Their results have shown that the submicron fraction was dominated by organic and sulfate species, with a chemical composition similar to those encountered in megacities like Beijing (Sun et al., 2012), Mexico (Salcedo et al., 2006) or Pittsburgh (Zhang et al., 2005). Such a pattern differs strongly from those

recently measured in European urban cities, where the main submicron pollutants are organic and nitrate species (Petit et al., 2015; Schlag et al., 2015).

In this study we offer the first insight of a real-time, continuous and long-term chemical characterization of $PM_1$ in West Africa, performed in M'Bour, Senegal from March to June 2015 as part of the first intensive observation period (IOP-1) of the SHADOW (SaHAran Dust Over West Africa) field campaign. An Aerosol Chemical Speciation

Monitor (ACSM) was chosen for the quantification and chemical characterization of non-refractory submicron particles, the instrument being better suited for unattended and long term monitoring. Parallel measurements of BC and total $PM_1$ allowed for mass closure. This paper reports the chemical composition of the submicron fraction and the temporal behaviour of particulate species. The sources and processes responsible for the concentrations encountered at the M'Bour site over the period have been assessed using source-receptor modeling as well as

dynamic meteorological measurements parameters and back-trajectory analyses.

## 2 Instrumentation and methods

### 2.1 Site description and the SHADOW campaign

The sampling site is located within the '*Institut de Recherche et Développement*' (IRD) facility in M'Bour (14°23'38''N, 16°57'32''W), Senegal. This site, located near the seashore at 80 km south of Dakar (Figure 1), is

known to be under the influence of Saharan dust, sea salt, and biomass burning during part of the dry season (from December to March). The site may also be affected by regional anthropogenic emissions from surrounding cities including M'Bour and Dakar, and from traditional fish-smoking activities and open waste burning areas. These potential aerosol source locations are reported in Figure 1. The M'Bour station, as part of the international AERONET network (Holben et al., 1998) is routinely equipped with active and passive remote sensing instruments

for cloud, aerosol and meteorological monitoring (CIMEL sun/sky photometer, CIMEL micro-Lidar, fluxmeters and





a weather station). Since the AMMA campaigns in 2006, M'Bour is also one of the three ground-based stations of the "Sahelian Dust Transect" where $PM_{10}$ mass concentrations are measured continuously (every 5 minutes) thanks to a Tapered Element Oscillating Microbalance (TEOM) (Marticorena et al., 2010; Kaly et al., 2015).

SHADOW main objectives are to better determine the physical and chemical properties of particles in this region largely influenced by high concentrations, and to establish a link between them, the atmospheric dynamics and the aerosol load and optical properties. A large panel of high-performance instruments has therefore been added to the AERONET station (Holben et al., 1998) implemented in M'Bour since 1996 (Derimian et al., 2008). Optical and microphysical aerosol measurements (results not presented) were also performed on site by active and passive remote-sensing instruments, like the LiLAS LIDAR (Bovchaliuk et al., 2014; Veselovskii et al., 2016) and a PLASMA airborne sun photometer (Karol et al., 2013). $PM_{10}$ in situ optical measurements at the surface were carried out by a mono-wavelength aethalometer and a nephelometer. Fine and coarse particle size distributions were recorded by a GRIMM optical particle counter, while ground and airborne filter sampling were collected through a 4-stage DEKATI cascade impactor to be analyzed off-line by individual particle analysis. A Doppler Lidar was implemented to improve our understanding of the atmospheric dynamics between the surface and up to 2-3 km over the IRD station in order to provide more accurate micro-meteorological information.

The online chemical composition measurements presented here were acquired during IOP-1, which took place from March 20 to June 22, 2015. Results discussed in this paper focus on the chemical characterization of surface $PM_1$. The instruments presented hereafter were set up in an air-conditioned room located underneath the flat roof of the IRD main building. Co-located wind speed and direction were measured by an ultrasonic anemometer (model USA-1, METEK GmbH) deployed on the rooftop (about 12 m above ground).

### 2.2 Instrumentation

### 2.2.1 ACSM

The chemical characterization of non-refractory submicron particles (NR-$PM_1$), that is to say material vaporizing around 600°C under close-to-vacuum conditions, was performed on-line and in real time every 30 minutes by an ACSM (Aerodyne Research Inc.). This instrument is based on the same principle as the aerosol mass spectrometers (AMS), without providing size distribution information. A full description of the instrument is presented in Ng et al. (2011). Basically it is composed of an aerodynamic lens that focuses the particle beam (with vacuum aerodynamic diameters below 1 μm) and directs it through three vacuum chambers, the last one being a detection chamber in which particles are vaporized by impaction on a surface heated at 600°C. Non-refractory species, such as organic matter (OM), sulfate ($SO_4^{2-}$), nitrate ($NO_3^-$), ammonium ($NH_4^+$) and non-refractory chloride (Chl), are vaporized at this temperature and then ionized by electron impact (70 eV). Particles are then detected by mass spectrometry thanks to a residual gas analyser (RGA). Because of a simplest operating system and the use of a quadrupole mass spectrometer, the instrument has lower time-resolution and sensitivity but remains better suited for long-term monitoring. A $PM_{2.5}$ cut-off inlet (URG Cyclone 2000-30EH, Chapel Hill, NC, USA) was placed on the roof at the entrance of the sampling line with a primary flow of 3 LPM and vertically connected to the instrument. A Nafion dryer (PD-200T-12 MPS, Perma Pure) upstream of the inlet reduces the sample relative humidity (RH). Particle





losses were evaluated using the Particle Loss calculator (Von der Weiden et al., 2009) and were inferior to 2% between 50 nm and 1 μm.

Several calibrations were performed with ammonium nitrate, ammonium sulfate and ammonium chloride individual solutions (at 0.005 mol L$^{-1}$ in purified water) prior to IOP-1. An average of all previous calibrations with this instrument gives a mean NO$_3$ response factor (RF) of $3.63 \times 10^{-11}$ and mean relative ionization efficiencies (RIE) of 5.72, 0.58 and 2.26 for ammonium, sulfate and chloride, respectively (see Supplementary Information S1 – Figure S1(a-d) and Zhang et al., in prep., for more details). Organic and nitrate RIE default values of 1.4 and 1.1, respectively, were used (Canagaratna et al., 2007). Detection limits in μg m$^{-3}$ were determined by Ng et al. (2011) to be 0.284 for ammonium, 0.148 for organic matter, 0.024 for sulfate, 0.012 for nitrate and 0.011 for chloride. The ACSM intercomparison which took place in Paris during winter 2013 has also allowed to determine expanded reproducibility uncertainties of 15, 9, 28 and 36% for NO$_3$, OM, SO$_4$ and NH$_4$ mass concentrations, respectively (Crenn et al., 2015). In addition to the relative ion transmission efficiency correction applied using a naphthalene internal standard, the Collection Efficiency (CE) due to particle losses induced by an incomplete vaporization and/or transmission through the aerodynamic lens was also determined for the whole dataset. Those parameters are mainly influenced by particle shape (size, sphericity) and acidity, ammonium nitrate fraction and RH in the sampling line. Middlebrook et al. (2012) have developed a correction algorithm based on AMS datasets which is applied on ACSM mass concentrations and which considers both RH and aerosol chemical composition to obtain a time-dependent correction of CE values ranging from 0.45 to 0.83 (Figure S1e). Nonetheless due to the presence of the Nafion dryer at the entrance of the instrument no RH corrections were necessary as RH values were below 30%. A minor fraction (3.2%) of the data was excluded from the dataset due to unstable parameters, which were generally observed after restarting the instrument following power outages.

**2.2.2 AE33 aethalometer**

Real-time measurements of aerosol absorption were performed every minute by a seven-wavelength (370, 470, 520, 590, 660, 880 and 950 nm) aethalometer (AE33, Magee Scientific Inc.). The instrument was equipped with a PM$_1$ impactor inlet (BGI model SCC-0.732, Waltham, MA, USA) and sampled at 5 L min$^{-1}$. The aethalometer principle is based on the measurement of light transmission through a filter onto which aerosols deposit. The attenuation is then converted into an aerosol absorption coefficient $\sigma_{aer}$ for each wavelength. BC concentrations are retrieved by applying a specific mass absorption cross section of 7.77 m$^2$ g$^{-1}$ to the absorption coefficient at 880 nm. The AE33 instrument uses internal corrections based on the Weingartner et al. (2003) algorithm to account for multiple scattering by the filter and a dual spot technology (Drinovec et al., 2015) to compensate the loading effect. Even if DD absorption is limited at 880 nm, its occurrence in high concentrations on site might cause an overestimation of BC concentrations derived from absorption measurements. The method developed by Fialho et al. (2005, 2006, 2014) was used to correct BC concentration data from mineral dust interference. This method is further explained in section II.3.2.





### 2.2.3 Other instruments

To achieve mass closure in the submicron fraction and account for the expected refractory material (mineral dust and sea salt), ambient air was sampled at 16.7 L min$^{-1}$ through a PM$_{10}$-inlet (Thermo Fisher Scientific Inc.) mounted on a PM$_1$ cyclone (SCC 2.229, BGI Inc., Waltham, MA). Gravimetric measurements of the total mass concentrations

were performed every 6 minutes using a TEOM operating at a temperature of 30°C and equipped with a filtered dynamic measurement system (TEOM-FDMS 1405-F, Thermo Scientific) that can account for semi-volatile material by maintaining temperature under 30°C and relative humidity (RH) below 25 %, as described by Grover et al. (2005). Some data had to be discarded, mainly due to high temperatures encountered during dust events leading to FDMS failure to maintain proper operating conditions.

Micro-meteorological parameters (wind speed and direction) at the surface (~10 m high) were obtained from 15-minutes accumulation (at 20 Hz) with an ultrasonic anemometer (model USA-1 by METEK GmbH) with a resolution of 0.01 m s$^{-1}$ and 1°, respectively. A weather station (Campbell Scientific) provided precipitation, RH and temperature data every 10 min.

### 2.3 Analysis strategy

**2.3.1 Classification of air masses**

The station of M'Bour is under the influence of a typical Sahelian climatic cycle composed of a dry and wet season. The dry season is generally defined from November to April and corresponds to a period for which no significant precipitation is recorded (Kaly et al., 2015). The period from May to October, defined as the wet season, corresponds to the period when precipitation is measured, although significant precipitation is generally limited to a few months,

i.e. between July and September in M'Bour during which heavy rain and thunderstorms occur (Kaly et al., 2015). The site is influenced by two important wind flows: the Harmattan conditions, with winds coming from North to North-East (N-NE, 0-45°) during the dry season, and the monsoon conditions during the wet season associated to winds from South-West to North-West (SW-NW, 225-315°). Our study taking place from March to June allowed for the observation of both the late part of the dry season (March-April) and the beginning of the wet season (May-June).

During IOP-1 two main prevailing directions were found (Fig. 2a). The first one corresponds to an oceanic influence characterized by surface winds coming from West-South-West to North-West (210-300°) with a total frequency of 56% and wind speeds between 2 and 4 m s$^{-1}$. The second predominant direction is observed for winds originating from NW to NE (300 to 60°) with a total frequency of around 42% and similar wind speeds (2-4 m s$^{-1}$). The remaining wind sector (60-210°) is a negligible fraction (2%) (Fig. 2a). The maximum wind speed, 6.8 m s$^{-1}$, was

measured on June 21, 2015, with values above 6 m s$^{-1}$ recorded between 2 and 6 a.m. and associated to SW direction. Our measurements during IOP-1 are generally consistent with monthly average frequencies of M'Bour surface wind directions reported by Kaly et al. (2015) between 2006 and 2010. Indeed, their climatology has shown that spring months are generally influenced by winds coming predominantly from two main sectors, North to East (0-90°, prevailing in March-April) and North-West to South-West (315-225°, dominant in May-June).

Each sampling day of IOP-1 was classified according to the locally measured surface wind directions. Three categories of days were indeed identified: (i) days exclusively under northern (N) trade wind influences, i.e. within ±



45° of the North (0°) direction, associated with continental influence, have been defined as "continental" (Fig. 2b);(ii) days dominated by westerly winds corresponding to oceanic air masses have been classified as "marine" days (Fig. 2d); (iii) an intermediate category called "sea breeze" days has been observed during which measurements show winds coming from the NE before 2 pm, then shifting from the N to W directions between 2 and 7 pm, and

returning to the NW in the evening (Fig. 2c). This phenomenon has been previously observed in M'Bour during the AMMA campaigns as reported by Derimian et al. (2008), Léon et al. (2009) and Deboudt et al. (2012). In summary, among the 91 days of IOP-1, 19% were classified as continental days, 32% as sea breeze days and 49% as marine days. The frequent occurrence of marine days can be explained by the transition from dry to wet season from April to June in the Sahelian region (Redelsperger et al., 2006; Slingo et al., 2008).

**2.3.2 Interpretation of absorption measurements**

M'Bour being largely under the influence of mineral dust, a possible overestimation of the amount of BC derived from absorption coefficients measured by the aethalometer (due to DD absorbing properties at shorter wavelengths) has to be considered (Bond and Bergstrom, 2006). These interferences may be enhanced during the dry season by internal mixture of BB and DD, as encountered and evidenced during AMMA by Deboudt et al. (2010) at the surface

and by Hand et al. (2010) and Paris et al. (2010) at higher altitude. Consequently, BC absorption coefficients have been recalculated following the method developed by Fialho et al. (2005, 2006, 2014) in order to obtain BC concentrations unbiased by DD influence. It consists in a deconvolution of the wavelength-dependent aerosol absorption coefficient over time, $\sigma_{aer}(\lambda, t)$, into two terms that take into account DD and BC contributions, through the following equations:

$$\sigma_{aer}(\lambda, t) = \sigma_{BC}(\lambda, t) + \sigma_{DD}(\lambda, t)$$
(1)

where $\sigma_{BC}(\lambda, t)$ and $\sigma_{DD}(\lambda, t)$ are BC and DD absorption coefficients, respectively, which can be expressed as a function of the species concentrations $\langle C_i(t) \rangle$:

$$\sigma_{BC}(\lambda, t) = K_{BC}\lambda^{\alpha}\langle C_{BC}(t) \rangle$$
25 (2)

$$\sigma_{DD}(\lambda, t) = K_{DD}\lambda^{\beta}\langle C_{DD}(t) \rangle$$
(3)

$K_{BC}$ and $K_{DD}$ are empirical constants depending of instrument characteristics, and $\alpha$ and $\beta$ are respectively BC and DD absorption exponents. $\alpha$ and $\beta$ values of - 1 and - 4, respectively, have been determined in the visible range (from

470 to 660 nm) by Fialho et al. (2014) with a dataset acquired in the Cape Verde Islands located at approximately 500 km from our sampling site.

Fialho et al. (2006) have replaced dust by iron in order to calculate an iron concentration from dust absorption:

$$K_{Fe}\langle C_{Fe}(t) \rangle = K_{DD}\langle C_{DD}(t) \rangle$$
(4)

Indeed, DD absorption is known to be mainly influenced by the iron content (Lafon et al., 2006) even if this element presents rather low mass contribution to the mineral dust total mass. As mentioned by Fialho et al. (2014), this





method allows to estimate elemental iron concentrations only in the absence of brown carbon. The m/z 60 signal ($C_2H_4O_2^+$, fragment characteristic of levoglucosan from biomass burning) was mostly absent of the ACSM dataset during IOP-1, representing on average only 0.003 (0.007 and 0.012 for the 95[th] and 99[th] percentiles, respectively, see Figure S5) of the total OM, which is identical to the threshold value of 0.3% suggested by Cubison et al. (2011) to

5    assess the presence of BB aerosols. This point is discussed further in the source apportionment results, in which we tend to attribute m/z 60 emissions to other sources. A few data points (less than 1% of the dataset), under the influence of other combustion processes from specific local activities that may cause a bias in the deconvolution algorithm, have been excluded to derive Fe concentrations (see further discussion in section III.2.2).

Combining Eq.(4) with Eq.(1, 2 and 3) leads to the following expression:

$$\frac{\sigma_{aer}(\lambda,t)}{\lambda^\alpha} = K_{BC}\langle C_{BC}(t)\rangle + K_{Fe}\langle C_{Fe}(t)\rangle\,\lambda^{(\beta-\alpha)}$$

(5)

Eq. (5) can be plotted as a linear equation at each time t to determine the intercept at the origin, a(t), and the slope, b(t). BC and Fe concentrations are then derived using $K_{BC} = 14.625\ \mu m\ m^2\ g^{-1}$ in Eq. (6) and $K_{Fe} = 0.234\ \mu m^4\ m^2\ g^{-1}$ in Eq. (7), respectively:

$$\langle C_{BC}(t)\rangle = \frac{a(t)}{K_{BC}}$$

(6)

$$\langle C_{Fe}(t)\rangle = \frac{b(t)}{K_{Fe}}$$

(7)

### 2.3.3 Positive Matrix Factorization (PMF)

20    The PMF model is a statistical source-receptor model developed by Paatero and Tapper (1994), largely employed in source apportionment of atmospheric pollutants when the source profiles and contributions are not known *a priori*. In this study, PMF was applied on the ACSM organic and chloride mass spectra by using the Multilinear Engine (Paatero and Tapper, 1999) and the version 5.3 of the Source Finder (SoFi) described in Canonaco et al. (2013) and operated with IgorPro 6.37 (Wavemetrics). PMF is based on a bilinear model described by the following equations:

$$X = GF + E$$

(8)

$$x_{ij} = \sum_{k=1}^{p}(g_{i,k}f_{k,j}) + e_{i,j}$$

(9)

where X, corresponding to model entries, represents the matrix of mass fragment spectra measured, and G and F are

30    matrices of a *k*-factor concentration time series and m/z profile, respectively, where *i* denotes the time step and *j* the mass fragment. The number of factors *p* is formerly determined by the user. The residual matrix E, containing the unexplained fraction of the PMF solution, is minimized by iterations of the model using the Q function:

$$Q = \sum_i \sum_j \left(\frac{e_{ij}}{s_{ij}}\right)^2$$

(10)





where $s_{ij}$ represents the measurement uncertainties of fragment $j$ at time $i$. The Q value is then normalized by $Q_{exp}$, representing the degrees of freedom of the model solution. This normalization is used as an indicator for the solution reliability. Thus a ratio $Q/Q_{exp}$ equal to one means that both variability and uncertainties are totally explained by the model.

## 3 Results and discussion

### 3.1 Chemical characterization and temporal behavior

#### 3.1.1 NR-PM$_1$, PM$_1$ and PM$_{10}$ mass concentrations

As previously mentioned, Senegal is widely influenced by DD events transported from arid and semi-arid regions of Sahara and Sahel. Moreover, M'Bour being a coastal site, the influence of sea salt (SS) particles on the measured aerosol mass concentrations may be significant. Thus, the contributions of these two aerosol types, in both the coarse and fine fractions of aerosol, have been investigated. PM$_1$ chemical mass closure was checked for all day types by subtracting from the TEOM-FDMS measurements the mass concentrations of species determined by ACSM (NR-PM$_1$) and aethalometer (BC + Fe); the fraction of unaccounted material therefore corresponded to DD and SS contributions.

Figure 3a shows the mass concentration time series of NR-PM$_1$ measured by ACSM, total PM$_1$ by TEOM-FDMS, and PM$_{10}$ by TEOM during IOP-1. It must be noted that total PM$_1$ data acquired between March 28 and April 10 had to be invalidated due to instrument overheating during a dust storm event when the outside temperature reached 42°C and PM$_{10}$ concentrations exceeded 600 µg m$^{-3}$.

The temporal evolutions of the three aerosol fractions do not show any particular correlations (the highest correlation coefficient is obtained between the PM$_1$ and PM$_{10}$ mass concentrations with r = 0.39 for n = 2666). The weak correlation (r = 0.26, n = 2946) between NR-PM$_1$ and PM$_1$ might be explained by the contribution of refractory material (DD and SS) to the total PM$_1$, while the absence of correlation (r = 0.08, n = 3424) between NR-PM$_1$ and PM$_{10}$ was expected, as well as the low average contribution (8%) of NR-PM$_1$ to PM$_{10}$. Indeed, the coarse fraction of aerosol (PM$_{10}$) at this site is dominated by DD and SS (Flament et al., 2011). NR species represent on average 71% of the total PM$_1$, which underlines the significant influence of refractory material in the fine fraction of aerosol measured at M'Bour. During IOP-1, as shown in Figure 3b, despite an important variability (values ranging between 4 and 25%) submicron particles (both refractory and non-refractory fractions) represent on average 11% of the PM$_{10}$ fraction whatever the day type. This pattern has already been observed at other sites influenced by Saharan emissions, and consequently under the influence of DD like Granada (Titos et al., 2015) or Cape Verde (Pio et al., 2014).

Despite similar orders of magnitude between the values of NR-PM$_1$ and total PM$_1$, their ratio exhibits different trends depending on day type (Figure 3c). When comparing PM$_1$ and NR-PM$_1$ concentrations for marine days, NR species appeared to account for most of the PM$_1$ fraction over IOP-1 (slope: 0.71, r = 0.82, n = 452), suggesting a minor influence of sea salt in the PM$_1$ fraction. This conclusion is consistent with the analysis of Flament et al. (2011) during the AMMA SOP-0 in M'Bour, with a reported PM$_2$ fraction composed of 18 to 77% of DD and less than 20%



of soluble ions (dominated by NaCl). Additionally for continental and sea breeze scatter plots also reported in Figure 3c, the discrepancies between NR- and total $PM_1$ mass concentrations are higher (slopes of 0.49 and 0.56, respectively), underlying that continental air masses – predominant during those days – carried additional refractory material such as DD in the submicron fraction. The unaccounted fraction (determined as the difference between the

gravimetrically measured $PM_1$ mass concentration and the sum of chemical species from ACSM and aethalometer measurements) corresponds to 27%, 26% and 16% of the $PM_1$ mass for continental, sea breeze and marine days, respectively (see Figure S2). A more significant difference could have been expected for continental compared to other days, but this might be explained by the absence of $PM_1$ mass measurements during the more intense dust events, as mentioned previously. Nevertheless, these results stress the need to apply Fialho et al. (2014)

deconvolution in order to separate DD and BC absorption contribution.

### 3.1.2 Estimation of absorbing compound concentrations in $PM_1$

Fe and BC concentrations obtained after correction led to averaged values of (0.55 ± 0.85) and (0.36 ± 0.37) µg m$^{-3}$, respectively, over the whole IOP-1 dataset. The deconvolution led to an average decrease of 45% of BC concentrations (factor of 2.2), which is much higher than the 11% decrease observed by Doumbia et al. (2012) in

Dakar where local BC sources are predominant. In our study the decrease even reached 83% under the influence of Saharan dust events and therefore the influence of DD on absorption measurements could never be neglected. As depicted in Figure 4, BC concentrations hardly exceeded 3 µg m$^{-3}$, except throughout punctual and short-term episodes during which they reached higher values, with a half-hourly average maximum of 3.6 µg m$^{-3}$ reached on April 2, 2015. The higher concentrations can be attributed to local anthropogenic combustion processes as BC

concentrations present a significant correlation (r = 0.79) with the ACSM m/z 57 combustion tracer. BC concentrations measured in M'Bour are on average much lower than those measured in Dakar during POLCA (Doumbia et al., 2012), where BC yearly average was (10.5 ± 3.5) µg m$^{-3}$. The lower concentrations measured here can be explained by the differences in sampling site type and population with the IRD center being located at the outskirts of the city of M'Bour (180 000-200 000 inhabitants) area while the site in Dakar (about 3 million

inhabitants within the metropolitan area) was located in an urban area. It can be noted that the range of BC concentrations (0.01-3.6 µg m$^{-3}$) obtained during IOP-1 in M'Bour is within the same order of magnitude than the one reported by Liousse et al. (2010) for the rural site of Djougou, Benin between Dec. 2005-Feb. 2006 (0.4-8.2 µg m$^{-3}$).

Fe concentrations estimated from $PM_1$ absorption measurements are considered as an indicator of DD in the fine

fraction. The following comparisons were carried out only on coincident measurements, due to missing data in the $PM_1$ and $PM_{10}$ datasets, corresponding to lower average concentrations of Fe (0.39 and 0.53 µg m$^{-3}$ for the $PM_1$ and $PM_{10}$ data comparison, respectively). Although a weak correlation (r = 0.55) was found between Fe and total $PM_1$ concentrations, Fe concentrations showed higher correlations with $PM_{10}$ (r = 0.70, see Figure 4). This could be explained by DD domination in the coarse fraction, while the fine fraction is mainly driven by NR and BC species

during most of the IOP-1 (Figure 3c). As depicted in Figure 4, most of the low iron concentrations were related to days under marine influence, while the highest Fe concentrations (> 8.0 µg m$^{-3}$) are generally associated with



continental and sea breeze days. These maxima also coincide with $PM_{10}$ highest concentrations (> 400 µg m$^{-3}$) and confirm iron as a constituent of mineral dust emitted by the Saharan and Sahel regions.

Fe contributions to $PM_{10}$ estimated in M'Bour (average Fe/$PM_{10}$ ratio of 0.51% over IOP-1 and 0.89% for continental days) are in excellent agreement with the ~1% of elemental iron in the $PM_{100}$ fraction reported by Joshi et al. (2017) for a DD sample collected at the same location in 2015 after a dust storm. It is however significantly lower than the average content of 11-12% of elemental iron estimated by Formenti et al. (2008) in the $PM_{40}$ dust fraction for samples collected at the ground level in Banizoumbou (Niger). Nonetheless they measured for the same samples an averaged iron concentration of 10 µg m$^{-3}$, in the same order of magnitude as our maximum concentration of 11.2 µg m$^{-3}$ in $PM_1$. Besides, Lafon et al. (2006) measured a total iron content of 6 to 8% in $PM_{10}$ bulk samples collected in Niger and Cape Verde, while Moreno et al. (2006) observed a 4% contribution of iron oxides in western Saharan soil samples. Concerning the finer fraction, a mean value of 4.7% was obtained for the Fe/$PM_1$ ratio over IOP-1 and 7.7% for continental days. The latter value is very close to the 8% of iron content (relative to all oxide mass) observed by Lafon et al. (2006) in the fine mode dust of Nigerian soil samples.

### 3.1.3 $PM_1$ average chemical composition

Table 1 reports ACSM measurements performed at M'Bour between March 20 and June 22, 2015, as well as other field campaigns carried out worldwide. This study shows an average value of 5.4 µg m$^{-3}$ for NR-$PM_1$ with a maximum half-hourly average of 68.3 µg m$^{-3}$. The total $PM_1$ mean concentration (including BC and Fe) is 8.2 µg m$^{-3}$ (maximum half-hourly average: 143 µg m$^{-3}$), a value which is about 10 times lower than the $PM_{10}$ mean concentration of 103.5 µg m$^{-3}$ over the same period. This $PM_{10}$ average is consistent with the range of monthly averages reported by Kaly et al. (2015) at M'Bour for the March-June period of 2006-2010 (between 63 and 126 µg m$^{-3}$). When compared to similar studies conducted around the world, M'Bour clearly appears much less polluted than megacities like Beijing (Sun et al., 2012) or Paris area during winter (Petit et al., 2014). The mean NR-$PM_1$ measured at M'Bour is close to the only other value reported in Africa for Welgegund, South Africa (Tiitta et al., 2014), regional background sites such as MontSec, Spain (Ripoll et al., 2015) or during summer in Paris with an average of 4.5 µg m$^{-3}$ (Petit et al., 2015).

The average NR-$PM_1$ chemical composition measured in M'Bour was composed of OM (39%), $SO_4$ (35%), $NH_4$ (15%) and $NO_3$ (9%), which is again rather similar to what was observed in South Africa by (Tiitta et al., 2014). Indeed, the OM and $SO_4$ predominance has been also observed in the majority of cities where ACSM (see Table 1) and AMS (Zhang et al., 2007) campaigns were implemented. Nonetheless, during winter, some European sites like Paris (Petit et al., 2015), Cabauw (Schlag et al., 2015) or Zurich (Lanz et al., 2010) present a fine fraction mainly dominated by nitrate and organic species. In our case these differences could be explained both by the semi-volatile nature of $NH_4NO_3$ combined with the limited use of fertilizers that prevent $NH_3$ emissions and ammonium nitrate formation, and more sources of $SO_2$ such as marine DMS oxidation processes.

Figure 5 represents the average contributions of NR-$PM_1$ species, BC and Fe in $PM_1$ over the entire IOP-1 and more specifically for continental, sea breeze and marine days. Total $PM_1$ pie charts, including unaccounted-for aerosol



species obtained by chemical mass closure, have been reported in the supplementary material section (Figure S2). Figure 5 shows that although an almost equal proportion of organic and sulfate species (31 and 32% respectively) was measured on average over the whole IOP-1, strong differences regarding day types can be highlighted: continental and sea breeze days present similar averaged compositions, with a major contribution of OM (36-40%)

followed by SO$_4$ with 21-24%. Similarities between the two profiles are probably due to a longer influence of northern wind compared to western ones during sea breeze days. On the other hand, for marine days the dominant fraction is sulfate (40%) while OM averaged contribution decreased to 25%. These changes could be explained by (i) oceanic air masses known to carry higher amount of sulfate species from the oxidation of dimethylsulfide (DMS) and organosulfur gases (Charlson et al., 1987; Fitzgerald, 1991) and (ii) long-range transport of polluted air masses from

the continent, carried back to M'Bour through oceanic air masses. BC and Fe from anthropogenic and continental origins, respectively, are also less abundant during marine days compared to continental/sea breeze days with fractions decreasing from 7 to 3% and 14-16% to 3%, respectively.

Comparable dynamics was already observed at M'Bour during AMMA SOP-0. For instance, Haywood et al. (2008) highlighted that BB aerosols carried in altitude (~3 km) from the African continent to the Atlantic Ocean were then

driven back through south-westerly winds. Similar contributions have been observed in Welgegund (Tiitta et al., 2014) where PM$_1$ measured during the dry season were dominated by OM (57%), followed by SO$_4$ (16%) and BC (10%), and the wet season marked by high contributions of SO$_4$ (42%) and lower contributions of OM and BC (35 and 4%, respectively). In M'Bour, we obtained a significant correlation (r = 0.74, Figure S7) between the OM and BC time series and an average BC/OM ratio of 0.15, whatever the day type, suggesting similar emission sources and

therefore pointing to continental origins either directly linked to combustion processes, as both species displayed a more important contribution during continental and sea breeze days, or due to the mixing of anthropogenic emissions with biogenic precursors or secondary organic aerosols (SOA).

Aerosol acidity can be considered as an indicator of the age of particles as they will get neutralized during their stay in the atmosphere. In order to estimate the degree of neutralization of ACSM inorganic species, NH$_4$ measured

concentrations (as NH$_2^+$, NH$_3^+$ and NH$_4^+$ ions) are compared to predicted ones, which are equal to the amount of NH$_4$ required to fully neutralize sulfate, nitrate and chloride anions according to the following equation (Zhang et al., 2007):

$$NH_{4,pred} = M(NH_4) \times \left( 2 \times \frac{SO_4^{2-}}{M(SO_4^{2-})} + \frac{NO_3^-}{M(NO_3^-)} + \frac{Cl^-}{M(Cl^-)} \right)$$

(10)

with SO$_4^{2-}$, NO$_3^-$ and Cl$^-$ the mass concentrations of inorganic species in µg m$^{-3}$ and M(X) their molecular weights (NH$_4$ 18; SO$_4$ 96; NO$_3$ 62; and Cl 35.5 g mol$^{-1}$).

The slope obtained between NH$_4$ measured and predicted (1.02, r = 0.92; see Figure S3a) underlined that most of inorganic species are neutralized over IOP-1, and additionally that there is no strong bias in the calibration values used for ACSM measurements. Nonetheless, a few points digress from the 1:1 line indicating partially neutralized

species (as mentioned above). Points significantly above the 1:1 ratio suggest the presence of NH$_4$ under other forms than (NH$_4$)$_2$SO$_4$, NH$_4$NO$_3$ and NH$_4$Cl. As these points also correspond to higher levels of OM, BC and chloride, circled in red (Figure S3b), they might be related to nitrogen-containing species associated with combustion





processes at low RH (<50%; Figure S3a), where chlorine could come from the combustion of sea salt and/or chlorine-containing materials such as plastics.

### 3.1.4 Variations of PM$_1$ chemical species

Figure 6 displays the 30-minute temporal variability of NR-PM$_1$ species during IOP-1. At such a timestamp a total of

13 pollution events – characterized by NR-PM$_1$ concentrations exceeding a threshold value of 15 μg m$^{-3}$, corresponding to three times the average – were detected. In terms of chemical composition, the highest average - over the whole IOP-1- concentrations were obtained for OM and SO$_4$ with respective values of 2.12 and 1.85 μg m$^{-3}$ (n = 3931). Moreover OM concentrations could present a high variability over short periods, like on May 12, 2015 when OM jumped from 0.9 to 61.7 μg m$^{-3}$ over an interval of 30 minutes, while the sum of other NR-PM$_1$ species

remained below 10 μg m$^{-3}$.

OM and SO$_4$ daily fractions exhibit opposite trends, with periods dominated by OM (> 30%) at the beginning of IOP-1 and from May 20 to June 8, 2015, corresponding to days under continental or sea breeze influences. OM concentrations encountered during these periods are generally related to punctual intense episodes, suggesting an influence linked to emissions by rather local anthropogenic activities rather than long-range transport sources.

Periods from April 25 to May 21 and after June 9, 2015, associated with oceanic air masses, are dominated by SO$_4$ (> 30%), with generally more moderate concentrations and less intense peaks, except for May 13$^{th}$ 2015, with NR-PM$_1$ reaching its maximum at 68 μg m$^3$.

The daily profiles of all identified PM$_1$ species and meteorological parameters (wind speed, temperature and wind roses) are presented in Figure 7 according to day types. Strong similarities can be observed between the continental

and sea breeze daily profiles (Fig. 7a-b), both presenting a morning peak around 8 am marked by a distinct rise in OM (> 4 μg m$^{-3}$) and BC (~1 μg m$^{-3}$) concentrations and to a lesser extent in NH$_4$ and chloride concentrations. Continental and sea breeze daily profiles also show a common peak at 8 pm, less intense than the morning one, but characterized by an increase in OM and BC. It can be noticed that this evening peak is less pronounced for continental days (OM ~ 3.5 μg m$^{-3}$ and BC ~ 0.7 μg m$^{-3}$) than for sea breeze days (OM ~ 4.5 μg m$^{-3}$ and BC ~ 0.9 μg

m$^{-3}$). As mentioned previously, the BC/OM ratio is on average of 0.13-0.14 for sea breeze and continental days (see Figure S7) but during the morning and evening peaks these ratios can reach a maximum of ~0.25 meaning that BC emissions are enhanced during these hours. A specific pattern of continental days is a peak measured at noon for OM and BC concentrations (constant BC/OM ratio ~ 0.16), probably combining local emissions and reduced dispersion due to dynamic phenomena as suggested by the slight wind speed decrease. On the other hand, for sea breeze days a

peak occurring at 3 pm, with an intense increase in OM, a moderate one in SO$_4$ and almost no variation of BC (ratio of 0.1), coincide with the sea breeze establishment (discernible through wind plots and the slight drop in temperature). BC for continental and sea breeze daily profiles show a maximum intensity of 1.1 μg m$^{-3}$ for the continental morning peak (8 am), a value which is quite consistent with maximal hourly concentrations (up to 4 μg m$^{-3}$) measured at 8 am, 4 pm and 8 pm in M'Bour by Deboudt et al. (2010). These peaks are measured for air masses

coming from the continent and correspond to traffic and/or cooking hours. Chloride peaks associated to intense OM



and BC ones tend to confirm combustion sources for these species, as previously observed during winter in Beijing by Sun et al. (2013).

The marine averaged daily profiles show a very distinctive pattern, compared to continental/sea breeze days (Fig. 7a), and are characterized by a sharp decrease of OM, BC and Fe, with nitrate, ammonium and chloride presenting

rather constant profiles, while sulfate exhibits a higher and almost constant concentration of 2.4 µg m$^{-3}$. This pattern as well as the sulfate peak associated with the sea breeze tends to confirm the regional transport of sulfate to M'Bour through oceanic air masses. It is noticeable that for marine days, OM and BC profiles reached maximum concentrations of 2.5 and 0.35 µg m$^{-3}$ during the night (between 8 pm and 2 am), in coincidence with low wind speed (Fig. 7d). Thus, this night time increase of concentrations measured for marine days might be due to lower dilution

of aerosols species from local emissions. The results obtained at M'Bour for marine days are in agreement with those reported by Topping et al. (2004) in Korea, with inorganic species such as $SO_4$, $NO_3$ and $NH_4$ increasing significantly during days under marine influence.

In Figure 7b, Fe and BC profiles showed distinctive behaviors depending on day types. As expected, both profiles for marine days are characterized by very low concentrations of both BC and Fe in comparison to those measured for sea

breeze and continental days. In the latter cases, a major difference can be observed for Fe concentrations which are almost twice higher than BC ones. Fe exhibits an additional intense peak around 3 pm and concentrations above 1.5 µg m$^{-3}$ from noon to 8 pm during continental days, which can be attributed to the intense dust events occurring around the end of March and on April 10[th] when comparing average and median profiles. These patterns suggest a transport of BC and DD (through Fe) by continental air masses – from the areas north and north-east of M'Bour.

Most probably air masses loaded in DD coming from the Saharan region (Fe continental profile) are then enriched in BC during their transport above cities. Nonetheless local traffic activities may also be a non-negligible source of dust through resuspension processes, which would also explain the synchronized peaks of BC and Fe observed for continental profiles even when excluding desert dust events.

**3.2 Origins and sources of aerosols**

**3.2.1 Geographical origins of chemical species**

As observed previously, most of the winds reaching M'Bour during IOP-1 were associated with the north and west sectors, carrying respectively air masses from the continent and the ocean, with an averaged wind speed of 2.6 m s$^{-1}$. These results are consistent with the five-year measurements of wind direction reported by Kaly et al. (2015) in M'Bour for the dry season. Figure 8 presents pollution roses obtained combining the time series of the different $PM_1$

components and co-located surface wind direction/speed measurements, thus offering a first insight into the geographical origins of chemical species. Similar spatial origins can be observed for several species; in particular, OM, BC and Chl maximum concentrations are rather associated with north-east (NE) winds during morning peaks and north-west (NW) winds at the end of the day (Fig. 7), pointing out to a common origin of the higher concentrations observed for these species, that would be linked to combustion processes due to the presence of BC.

Figure 8 also highlights an oceanic origin for some OM (with moderate intensity), $SO_4$ (as already observed in Fig. 5), $NO_3$ and $NH_4$. There is no clear trend linked to the wind speed (slower along the coast and higher when





perpendicular to it), meaning particulate chemical species are rather transported or formed above the ocean than directly emitted by sea spray (Ovadnevaite et al., 2012). Emissions of DMS and organosulfur gases by microorganisms (as mentioned previously) could explain part of the $SO_4$ and OM concentrations observed when the site is under western influence. These species may also be released by anthropogenic activities in distant cities like

Dakar, whose emissions may be carried toward the ocean and brought back to M'Bour by western winds. A long-term $PM_{10}$ chemical characterization conducted in Cape Verde by Fomba et al. (2014) has also shown that marine air masses were mostly composed of sulfate (10% on average) rather than organic matter (3.5%) but the proportion in $PM_1$ is more balanced.

Regarding the iron pollution rose plot reported in Figure 8, maxima are measured when the site is under the influence

of NE winds. For such directions we also observe maximum values of the total $PM_1$ concentrations (> 60 µg m$^{-3}$), which correspond to the April 10, 2015, most probably a resuspension event following previous dust storms, as observed on $PM_1$ and Fe concentrations time series (reported respectively on Figure 3 and 4).

### 3.2.2 Source apportionment

PMF was first applied without constraints on the whole IOP-1 organic database and then separately on continental,

sea breeze and marine extracted datasets (see also Supplementary Information Appendix S4). Three to ten factors were tested and for each run, only solutions with a normalized $Q/Q_{exp}$ ratio close or inferior to 1 were taken into account. Mass spectra were compared to reference profiles for identification and the consistency with daily profiles was checked. Without any constraints applied to the model, one or two factors identified as oxygenated organic aerosols (OOA, dominated by the $CO_2^+$ fragment at m/z 44) were easily retrieved for all datasets. An unknown factor

with peaks at m/z 58, 60, 83 and 91 appears for the whole IOP-1 when the number of factors increases (and is mixed with one of the OOA in the reduced datasets). This factor was associated with one of the OOA for the continental, sea breeze and marine 4-factor unconstrained solutions (Appendix S4, Figure S4.1). Besides, the model encountered difficulties to separate hydrocarbon-like (HOA) from cooking-like (COA) organic aerosol for the entire IOP-1 dataset and the continental and marine days. Both COA and HOA share similar m/z peaks at 27, 41, 55 ($C_nH_{2n-1}$) and

29, 43, 57 ($C_nH_{2n+1}$) that correspond to hydrocarbon fragmentation. Nonetheless, they can be differentiated based on 41/43 and 55/57 ratios, more important for COA (Fröhlich et al., 2015a; Mohr et al., 2012), while HOA presents more specific and intense peaks at 69-71. Moreover under continental influence, they tend to be emitted from the same wind directions corresponding to the urbanized area from NW to NE. A good separation with characteristic factor profiles was nonetheless obtained for sea breeze days which present a more variable dynamics, with 4 factors

identified, namely HOA, COA, as well as more oxidized and less-oxidized OOA (mixed with the unknown factor) (Appendix S4, Figure S4.1c). The unknown factor presents a rose plot and a daily profile consistent with local emissions from the open waste burning areas of Gandigal (NW) and Saly Douté (NE) and a fish-smoking area also located in the NE of M'Bour (Appendix S4, Figure S4.2), hence it has been designated as LCOA (Local Combustion Organic Aerosol). Since the behavior of Chl had also been suspected to come from the same sources, PMF solutions

adding the m/z 36 signal in the input matrix were investigated, and a solution is presented in supplementary (Appendix 6). No biomass burning OA (BBOA) profile appeared neither in our unconstrained (up to 10 factors) runs,





nor when HOA and COA were themselves constrained. Furthermore, local practices tend to use dry leaves and branches to feed livestock rather than as combustion fuel, and cooking in the urban area is mainly done on gas stoves. IOP-1 occurring at the transition between dry to wet seasons, the possibility of long-range emissions of BB from Savannah fire, like the one observed for example in South Africa (Tiitta et al., 2014) and northern Australia

(Milic et al., 2016), or even other BB regional sources, was not discarded at first. Therefore, an average BBOA profile (Ng et al., 2011b) was used to constrain the PMF solution with different a-values. Using a rather strong constraint of a = 0.3 led to a satisfactory BBOA profile but either stable solutions presenting a mixture between LCOA and part of the OOA factor but no COA, or when LCOA, OOA and COA were well separated, the BBOA profile presented missing fragments and a high run-to-run instability. Moreover, solutions for more moderate

constraints (a ≥ 0.5) led to a BBOA profile with unusually low or missing important fragments (such as m/z 60) and an increasing correlation with COA (Crippa et al, 2013). Besides, the important photochemistry observed on site coupled with levoglucosan (m/z 60, 73) fast reactivity under photo-oxidative conditions (Hennigan et al., 2010) might lead to a fast transformation of BBOA into SOA (which would appear as an OOA factor in our PMF solution). Finally, additional in situ or remote sensing optical instruments specifically implemented for the SHADOW

campaign, such as ground-based and airborne sunphotometers or the LILAS multiwavelength Mie-Raman lidar (Veselovskii et al., 2016), did not detect the presence of BB aerosol influence. For all these reasons, we decided not to constrain BBOA in our final solutions. In order to refine the solution, and due to the possible specificity of local emissions, the PMF model was run with mild constraints on the primary factor profiles observed in the different unconstrained runs (see Fig. S.4.1) that is to say COA (Crippa et al., 2013) and HOA (Ng et al., 2011b) using the a-

value approach with 30 to 90% of freedom (0.3 < a < 0.9) as explained in Canonaco et al. (2013). Indeed HOA are generally emitted by diesel or fuel exhaust, although in rural cities they have also been related to cooking activities (Sun et al., 2012). It has also been reported that African fuels supplied directly by Western company-owned petrol stations present higher levels or sulfur and polyaromatics, and that others are generally a mixture of refinery intermediate products and other (mostly unknown) chemicals, which could hamper the finding of a satisfactory

solution. The robustness of the optimal solution was checked by 50 random seed iterations.

A final constrained solution of 4 factors is presented in Figure 9 through their respective mass spectra with associated pollution rose plots and daily profiles for continental, sea breeze and marine days.

The three primary (POA) factors linked to anthropogenic activities, that is to say HOA, COA, and LCOA corresponded to 18, 30 and 3% of the organic fraction, respectively. HOA and COA contributions to the OA fraction

tend to be more important for continental and sea breeze days with respectively 21-24% and 24-31% than for marine days (11-20%), while LCOA slightly increases from 2 to 7%. The HOA profile is well correlated with BC (r = 0.79) and m/z 57 (r = 0.98). No specific external tracers have been identified nonetheless COA temporal variability appeared to correlate well with BC (r = 0.73). The HOA rose plot shows marked peaks in the directions of the two open waste burning areas and of the fish-smoking area located northeast of the site in the outskirts of M'Bour. COA

main origin in the northwest corresponds to the touristic area of Saly where a number of restaurants are located. BC concentrations previously linked to OM species present close correlations with both HOA and COA but this might be explained by the correlation of 0.82 between both time series suggesting concomitant emissions between traffic and



cooking sources. The variability depicted by their continental and sea breeze daily profiles (Fig. 9b) highlights anthropogenic emissions of COA and HOA with pronounced peaks during morning and evening (0.46 to 0.55 µg m⁻³) corresponding to human activities. The COA and HOA profiles present an additional peak after 11 am, reaching a maximum around noon then decreasing after 2 pm for continental days. The COA, OOA – and to a much lower

extent, the HOA – sea breeze profiles differed from the continental ones by a delay in the midday peak which rises only after 2 pm. This difference could be explained by anthropogenic aerosols emitted in the morning, then transported toward the ocean, and carried back on site through sea breeze phenomena. A similar pattern could have been expected from HOA, but the low concentrations observed suggest faster oxidation processes occurring over the ocean which would contribute to the higher OOA peak. Both COA and HOA factors present low background

concentrations during marine days.

As for LCOA which is a very small fraction of the total OA, the fact that it nonetheless consistently appears in the PMF analysis under both unconstrained and constrained conditions, suggests a specific behavior uncorrelated with other sources. Its robustness has been tested under various starting conditions (50 seed iterations) and through rotational ambiguity exploration (F-peak tested between -5 and +5 with a step of 0.5). Despite the fact that Chl

species were presenting similar wind sectors origins, its correlation with LCOA remains low (r = 0.44) while LCOA presents higher correlation with m/z 36 and 58 (r = 0.55 and 0.84, respectively). This is mainly due to the influence of refractory chloride (NaCl) in our Chl measurements; which has been observed through m/z 35 negative values likely resulting from a slow vaporization both during filter and non-filter measurements (Nuaaman et al., 2015). Nonetheless m/z 36 (HCl⁺) was mostly positive (0.03 ± 0.03, see Figure S5) confirming that non-sea salt chloride

was also detected. Indeed, if sea salt was predominant in LCOA, NaCl could explain the m/z 58 and 60 signals, but one would have expected a maximum on marine days, whereas LCOA rose plot clearly points out toward the local combustion areas already mentioned previously, and also identified in the chloride rose plot (Figure 8). Although coal combustion used for cooking has been identified as a potential source of chloride emissions (Ianniello et al., 2011; McCulloch et al., 1999), this practice has not been observed in the area. However, chloride species may be

emitted from household waste burning/smoldering, for which particle-bound polychlorinated dibenzo-p-dioxins and dibenzofurans (PCDD/Fs) and polychlorinated biphenyls (PCBs) have been measured in previous studies (Gullett et al., 2001; Lemieux et al., 2004; Tue et al., 2016; Zhang et al., 2009); or from sea salt particles subject to high-temperature combustion processes (since this factor is also observed to a lesser extent in the direction of the fish-smoking area).

Additionally, the measured NH₄ was higher than the predicted one in the directions of the waste burning areas - especially from the North-East (Figure S3b), also suggesting the formation of nitrogen-containing compounds that could lead to R-N⁺ or R-NO⁺ fragments at these same masses although high-resolution measurements would be needed to confirm this hypothesis. LCOA also present its most intense peaks in the morning (0.04 µg m⁻³) and some more moderate in the evening (< 0.03 µg m⁻³) for continental and sea breeze daily profiles, characteristics of point

sources and/or local emissions. These may be linked to the start of the fires in the morning for the open waste burning areas and an evening increase of the OM due to temperature drop that would shift the partitioning of semi-volatile species toward the particulate phase. On the contrary its marine profile shows constant but higher





concentrations all day long (~0.03 µg m$^{-3}$) with an increase after 8 am, which explains the slightly higher contribution to the OA fraction for the latter.

OOA are often considered as secondary organic aerosols (SOA) formed through gas-conversion processes of Volatile Organic Compounds (VOCs) or photochemical oxidation of primary OA emitted by biogenic (such as plants or algae) and/or anthropogenic sources. The hot temperatures and intense solar irradiation encountered in the region enhance these processes and can explain the major contribution (45%) observed for the OOA factor during IOP-1. Its profile showed intense peaks at m/z 18 and 44 ($f_{44}$ = 0.36) and low 43/44 ratios (0.04), which underlined the high level of oxidation characteristic of aged particles, and presented a correlation of 0.88 with the OOA averaged profile reported by Ng et al. (2011). In our study, the temporal variability of OOA shows a higher correlation with NO$_3$ than SO$_4$ (r = 0.56 against r = 0.24). As NO$_3$ species are generally semi-volatile and associated with less aged air masses (Mohr et al., 2012), OOA might not be only emitted by long distant sources. This is highlighted by daily continental and sea breeze profiles rising around 8 am, noon and 8 pm which emphasize direct emission of oxidized organic aerosols from anthropogenic activities. Also, it has been mentioned previously that NO$_3$ might not be locally emitted (Figure 8) and OOA highest concentrations being associated with wind speeds above 3 m s$^{-1}$, we attribute also a regional origin to this factor. Both the OOA correlation with NO$_3$ and the fact that its daily cycles progressively increase/decrease with sunrise/sunset suggest an important and fast photochemistry (Robinson et al., 2007). The OOA rose plot shows rather similar concentrations regardless of wind sectors, even if some concentrations superior to 1.5 µg m$^{-3}$ are connected to western wind influences (Figure 9). This oceanic origin is also highlighted by OOA sea breeze profile, which shows a maximum of 0.7 µg m$^{-3}$ reached around 3 pm (Figure 9). It indicates that this SOA formation is enhanced during OA transport over the ocean which under specific atmospheric conditions can be initiated by chlorine atoms (Hallquist et al., 2009). OOA increasing during daytime regardless of day type, their high oxidation level and important contribution to OA, especially during marine days (62%), support emissions by regional sources.

Previous observations of OM from daily filter measurements in the PM$_2$ fraction during AMMA SOP-0 were associated maximum values with winds coming from the coast (NW) and the ocean (W) (Flament et al., 2011) and attributed to emissions by local sources and cities along the Senegalese coast, suggesting they could originate from wood burning, cooking and dry leaf/grass burning. Only considering biomass burning and fossil fuel combustion as contributing to the absorption measurements during their field campaign in Dakar, Doumbia et al. (2012) also reported 12% of biomass burning emitted by regional sources against 88% by fossil fuel. Our measurements over a large period of four months at high-time resolution suggest, as mentioned before, that biomass burning from local sources is rather negligible in the M'Bour area during the transition period from dry to wet seasons. Cooking, traffic and other local combustion sources such as traditional fish smoking and household waste burning have been identified as the primary local anthropogenic sources of OM.

## 4 Conclusions

The deployment of high time resolution instruments at the West African coastal site of M'Bour (Senegal) during four months of the 2015 dry season allowed to investigate the temporal variability and chemical composition of the





poorly characterized $PM_1$ aerosols in this region. The average $NR-PM_1$ concentration (5.4 µg m$^{-3}$) was relatively low, comparable to levels generally reported in rural environments and also close to the one observed at a South African site. Although marine influence was dominant, various dynamic conditions were encountered during the campaign, with intense dust events, sea breeze phenomena and several anthropogenic episodes ($NR-PM_1$ reaching up to 68 µg

m$^{-3}$ over a 30-min period). $PM_1$ concentrations were on average 8.2 µg m$^{-3}$ representing 11% of $PM_{10}$, and were dominated by NR species (71%) with minor contributions of absorbing species (9%). The remaining unaccounted fraction (20% on average, up to 75% during identified dust events) was mainly attributed to mineral dust, strongly suggesting a minor contribution of sea salt in the submicron fraction. Fe concentrations (5%) obtained from the deconvolution of absorption measurements were consistent with the dust events observed on site. Among the species

identified in the $PM_1$ fraction, sulfate and organics represented 32 and 31%, respectively, followed by ammonium 14%, nitrate 9%, BC 5%, Fe 9% and Chl <1%. OM dominated the $NR-PM_1$ fraction when the sampling site was under continental air mass influences (north-eastern winds), whereas marine air masses (from western winds) preferentially brought higher concentrations of sulfate species on site.

Source apportionment of the organic fraction allowed to identify four types of OA. The organic fraction is composed

of a highly-oxidized OOA (45%), whose regional origin is underlined by its important contribution to the organic fraction during marine days (62%) but also by its increasing concentration during daytime with a maximum under sea breeze influence. Nonetheless, its higher correlation with $NO_3$ as well as morning and evening peaks observed for continental and sea breeze daily profiles tend to associate it partially to direct emissions or fast oxidation processes of anthropogenic compounds. Three primary OA linked to anthropogenic activities from nearby sources

were also identified: HOA (22%), COA (28%) and a new factor LCOA (3%) related to local combustion sources (emissions from open-waste burning and fish smoking areas), for which a good correlation with particulate chloride (m/z 36) was consistently found. Non-refractory chloride fragments from waste burning or fish smoking areas were suggested to originate from plastic smoldering (for the former) and/or sea salt (for both) submitted to high temperatures. This factor, although minor on average, could represent as high as 7% on a 30-minute time period

when the air masses were blowing from the waste burning areas, and very likely resulted in the concomitant emissions of highly-toxic compounds such as dioxins that would require further investigation.

Overall, our study suggests that natural sources strongly influence $PM_1$ levels in M'Bour, mainly due to the large influence of marine conditions associated with high sulfate levels, and additional significant influence of desert dust. Our measurements carried out at a suburban site away from megacities allowed to provide new insight into the

complex mixture between local anthropogenic sources and regional background aerosols in the $PM_1$ fraction in West Africa at the end of the dry season.

**Data availability**

Data from ACSM and aethalometer instruments are available upon request to the corresponding author, V. Riffault. $PM_{10}$ data can be accessed by request to A. Féron (LISA).



**Acknowledgements**

LHR's PhD grant and the SHADOW campaign are financially supported by the CaPPA project (Chemical and Physical Properties of the Atmosphere), which is funded by the French National Research Agency (ANR) through

5    the PIA (*Programme d'Investissement d'Avenir*) under contract "ANR-11-LABX-0005-01", by the Regional Council "*Hauts-de-France*" and the European Regional Development Fund (ERDF).

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



**Table 1. Averaged NR-PM1 concentrations (in µg m-3) - with maximum in parentheses when available - and respective contributions of NR compounds for this study compared to other ACSM campaigns from different locations.**

| NR-PM$_1$ avg (µg m$^{-3}$) | Relative contributions (%) | | | | | Period | Location | Type of site | Reference |
|---|---|---|---|---|---|---|---|---|---|
| | OM | SO$_4$ | NO$_3$ | NH$_4$ | Chl | | | | |
| 50.0 | 40 | 18 | 25 | 16 | <1 | Jun. - Aug. 2011 | Beijing (China) | Urban | Sun et al. (2012) |
| 25.9 | 58 | 23 | 7 | 11 | <1 | Sept.-Dec. 2015 | Hong Kong (China) | Urban | Sun et al. (2016) |
| 15.7 (80) | ~44 | ~6 | ~38 | ~11 | ~1 | Jan. - Mar. 2012 | Paris (France) | Suburban | Petit et al. (2014) |
| 15.3 | 72 | 15 | 6 | 8 | < 1 | Summer - Fall 2011 | Atlanta (USA) | Urban | Budisulistiorini et al. (2014) |
| 14.2 | 58 | 12 | 21 | 8 | 1 | Mar. 2013 – Mar 2014 | Ispra (Italy) | Rural background | Bressi et al. (2016) |
| 10.8 (~75) | 45 | 30 | 11 | 13 | 1 | Sept. - Oct. 2013 | Menyuan, Tibetan Plateau (China) | Rural | Du et al. (2015) |
| 9 | 31 | 12 | 41 | 14 | 2 | Jul. 2012 -Jun. 2013 | Cabauw (The Netherlands) | Rural | Schlag et al. (2015) |
| 7.5 (89) | 48 | 32 | 7 | 13 | <1 | Sept. 2010 - Aug. 2011 | Welgegund (South Africa) | Rural | Tiitta et al. (2014) |
| 7.2 | 52 | 29 | 7 | 12 | < 1 | Jul. - Sept 2010 | New-York (USA) | Urban | Ng et al. (2011) |
| 7.0 | 57 | 12 | 21 | 9 | < 1 | Nov. 2010 - Jun. 2012 | Southern Great plains (USA) | Rural | Parworth et al. (2015) |
| ~7.0 | 54 | 19 | 11 | 11 | < 5 | Jun. 2012 - Jul. 2013 | Montseny (Spain) | Rural | Minguillón et al. (2015) |
| 5.4 (68.3) | 39 | 35 | 9 | 15 | <1 | Mar. - Jun. 2015 | M'Bour (Senegal) | Suburban | This study |
| 4.9 | 53 | 25 | 7 | 12 | < 1 | Jul. 2011 - Apr. 2012 | Montsec (Spain) | Rural | Ripoll et al. (2015) |
| 0.3 (9.6) | 43 | 30 | 7 | 17 | / | Oct. 2012 - Oct. 2013 | Jungfraujoch (Switzerland) | Rural | Fröhlich et al. (2015) |



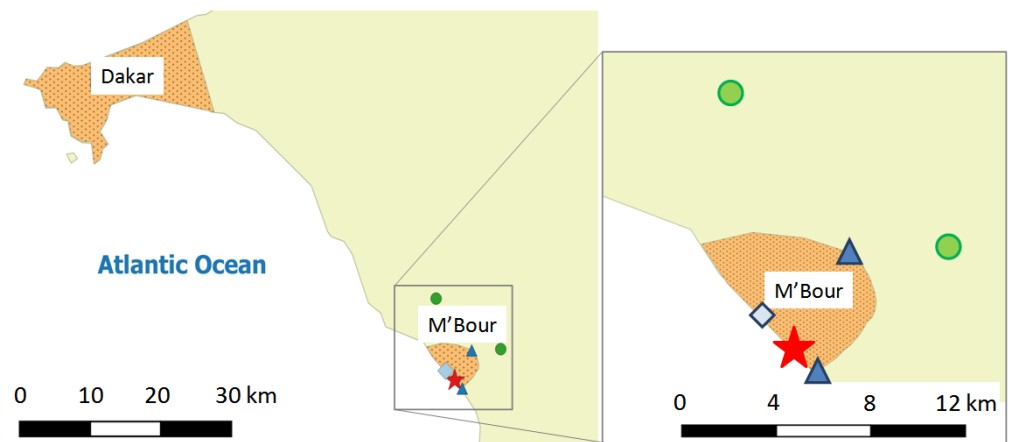

**Fig. 1.** (left) Dakar and M'Bour locations with city delimitations in orange and (right) local sources located around the IRD sampling site (red star), with open waste burning areas (green circles), fish-smoking sites (blue triangles) and the M'Bour port (light blue diamond).

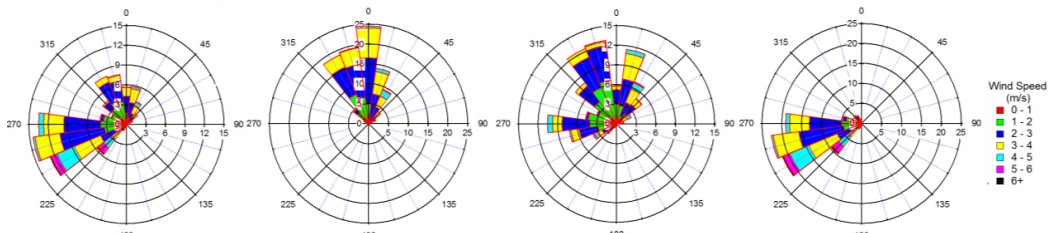

**Fig. 2.** Rose plots of wind direction divided into 15° sectors and averaged on 30 minutes (ACSM time step) with wind frequencies as radius (in %) and colored by wind speed intervals measured in M'Bour for (a) the whole dataset, (b) days classified as continental (n = 17), (c) sea breeze (n = 29) and (d) marine (n = 45).



**Fig. 3. (a) NR-PM$_1$, total PM$_1$ and PM$_{10}$ time series (30-minute averages) measured over IOP-1, (b) 2h averaged scatter plots of total PM$_1$ vs PM$_{10}$ and (c) NR-PM$_1$ vs PM$_1$ for (left) continental, (middle) sea breeze and (right) marine days (dotted lines are visual markers representing ratios between the different variables).**

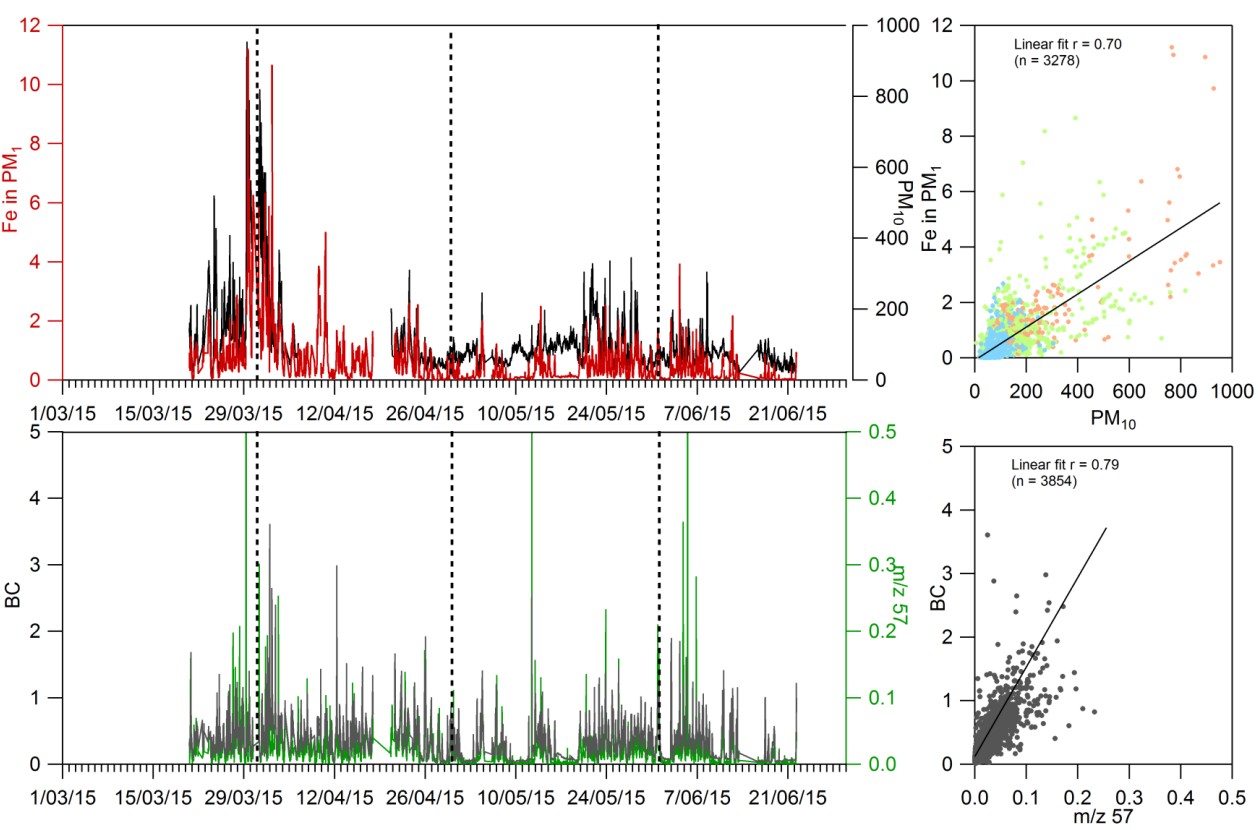

**Fig. 4. (Left)** Time series of **(top)** Fe and PM$_{10}$ concentrations (in µg m$^{-3}$) and **(bottom)** BC concentration (in µg m$^{-3}$) and m/z 57 (ng m$^{-3}$) 30 min average. **(Right)** Corresponding scatter plots and their respective linear fits with Fe and PM$_{10}$ data colored in red for continental, green for sea breeze and blue for marine days and BC and m/z 57 in grey dots.



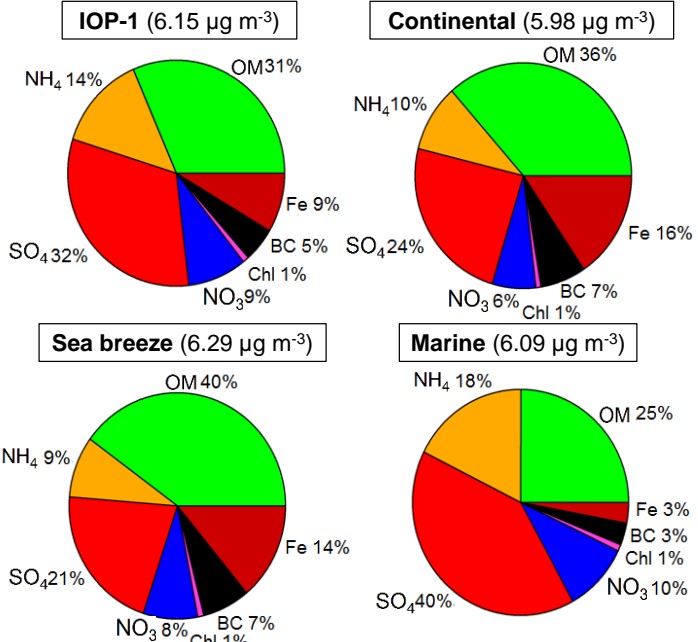

Fig. 5. Averaged contributions of NR-PM₁, BC and Fe for IOP-1 (n = 3771), continental (n = 307), sea breeze (n= 799) and marine days (n = 1843) (with average total concentration in µg m⁻³ in parenthesis).




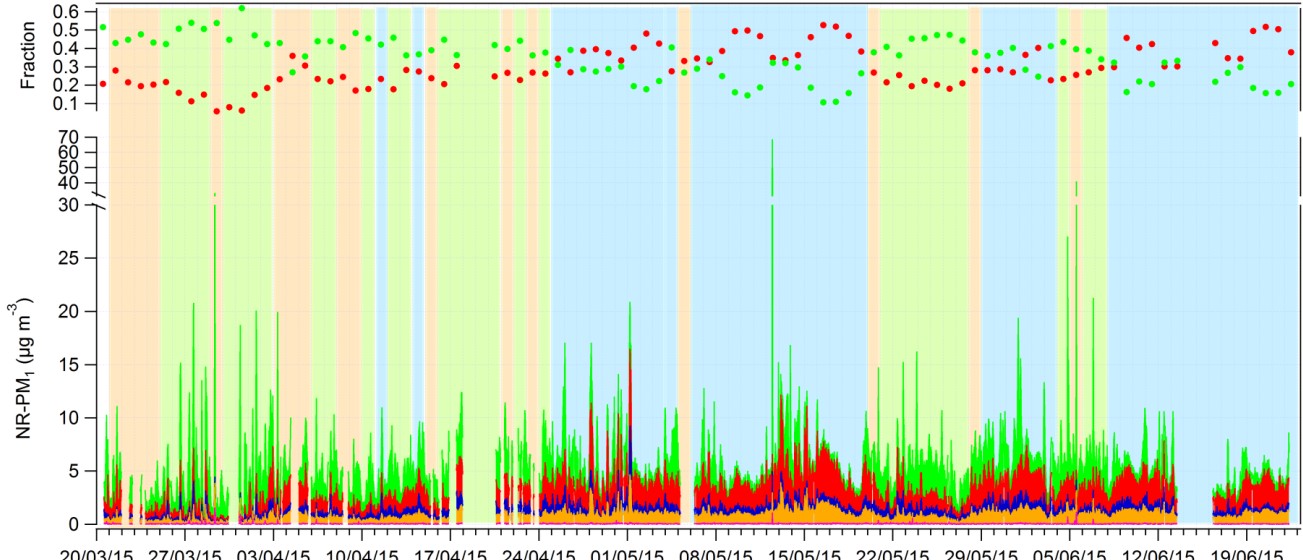

**Fig. 6. (Bottom) IOP-1 stacked time series of OM (green), SO₄ (red), NO₃ (blue), NH₄ (orange) and Chl (pink) on ACSM time step (30 min) and (top) daily averaged fraction of OM and SO₄. Tinted areas correspond to continental days in light pink, sea breeze days in light green and marine days in light blue.**



**Fig.7. Average daily profiles of (a) NR-PM₁, (b) BC and Fe concentrations, (c) wind speed and temperature for (from left to right) continental (dotted lines: medians), sea breeze and marine days. (d) Associated wind roses (radius, wind speed in m s-1) colored by time of day in UTC.**




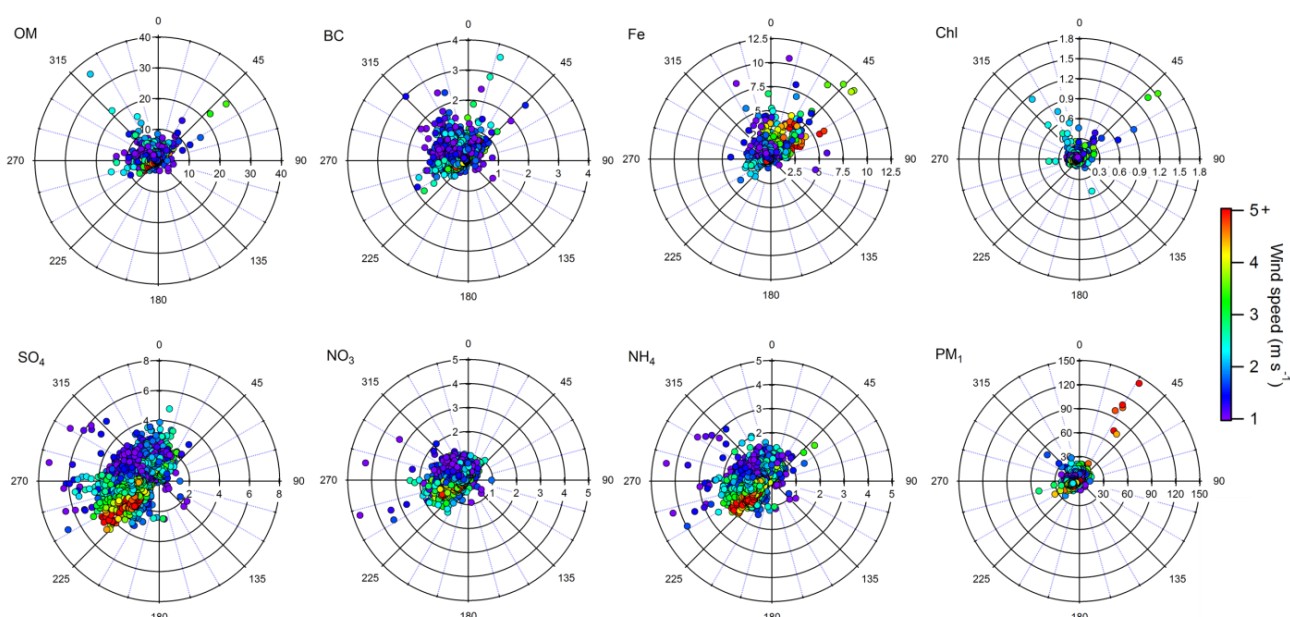

**Fig. 8.** Pollution rose plots of OM, BC, Chl, Fe, SO$_4$, NO$_3$, NH$_4$ and total PM$_1$ for the whole IOP-1, with concentrations (µg m$^{-3}$) as radius and colored by wind speed (measurements inferior to 1 m s$^{-1}$ in grey).





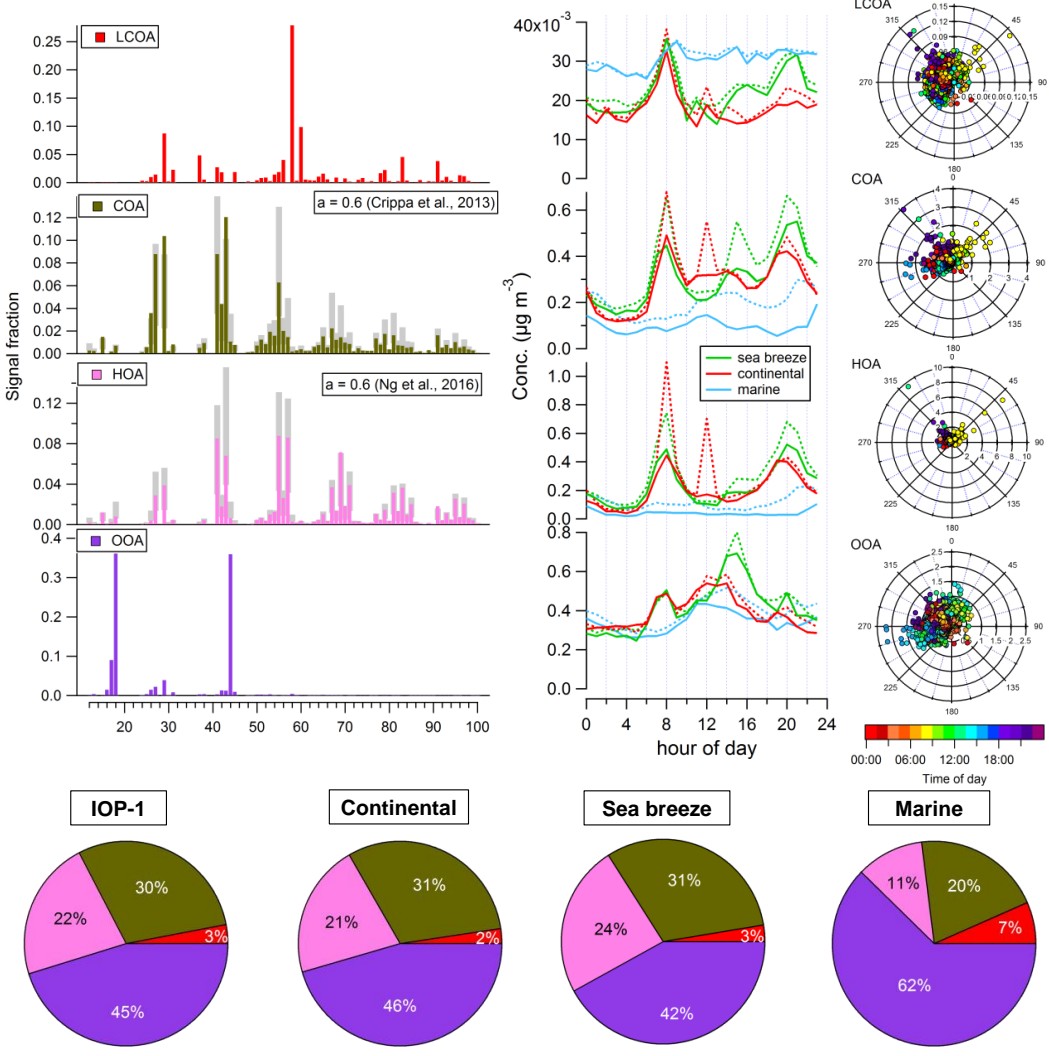

**Fig. 9. PMF constrained 4-factor solution: (left) factor profiles of LCOA, COA, HOA (the two latter constrained), OOA; (middle) corresponding daily cycles according to day types (solid lines: median; dotted lines: average); and (right) pollution rose plots colored by time of day. (bottom) Average pie charts of the contributions to the total organic fraction for IOP-1, continental, sea breeze and marine days.**