# Peer review of "Chemical characterization and source apportionment of submicron aerosols measured in Senegal during the 2015 SHADOW campaign"

_Atmospheric Chemistry and Physics, 2016_

## Referee Comment (RC1) · Anonymous Referee #2 · 28 Mar 2017

I recommend this manuscript for publication in ACP.

The manuscript presents interesting results on the submicron aerosol in Senegal. The datasets are new, the data analysis is very complete and thorough, and the manuscript is well structured and well written. There is one main issue to be addressed (first one listed below) and some additional comments to be addressed before the work can be published in ACP.

1. Please consider revising the calculated Fe concentrations and/or the PM1 concentrations and/or the ACSM concentrations (RF for NO3). These data together have

inconsistencies that need to be addressed or commented and the possible sources of uncertainty should be stated. The authors report a mean value of 4.6% for the Fe/PM1 ratio. This means 16% for the Fe/RefractoryPM1, since 71% of PM1 is NR-PM1. (Or 20% for the ratio Fe/(PM1-ACSM-BC) according to Figure S2, considering 5% of Fe and 25% of PM1-ACSM-BC). Hence, it is this 16% (or 20%) which should be compared to the data in the literature, given that the literature data that the authors quote make reference to Dust (and not total PM, regardless of the size fraction). Please see some additional comments related to this one below.

2. The sampling period is actually 3 months (20 March to 22 June), and not 4 as stated.

3. Please homogenize the dry and wet period definition: in the introduction it says that the dry season extends from November to May; in section 2.1 it says that the dry season is from December to March; in section 2.3.1 it says that dry season is generally defined from November to April; in the same section 2.3.1 it is coherent within the section and it says that "Our study taking place from March to June allowed for the observation of both the late part of the dry season (March-April) and the beginning of the wet season (May-June)"; in the conclusions section it says "during four months of the 2015 dry season". Given that precipitation data for the specific campaign is available, according to section 2.2.3, could you please provide this data, or make the classification based on these data? (Although the info in literature about the usual dry-wet periods can still be included).

4. Section 2.2.1. Please confirm that no major changes (such as filament replacement) occurred to the ACSM during the time between Feb 2014 and Jan 2015, for which the calibration constants were taken. This is necessary to be able to take the average of all calibrations. Especially between Dec 2014 and Jan 2015 there is a big difference for the RF of NO3 (Fig S1). This is a source of uncertainty that should be acknowledged in the manuscript, especially considering that the absolute concentrations are used and that the differences with the bulk PM1 concentrations are taken as very valid and interpreted.

5. Section 2.3.1. "During IOP-1 two main prevailing directions were found (Fig. 2a). The first one [...] and North-West to South-West (315-225°, dominant in May-June)" Maybe you could indicate which wind direction prevails for the first and second periods (Mar-Apr and May-Jun), since the wind directions for the periods are given based on literature info, but the data for the specific campaign is available and then the wind roses are commented for the entire period.

6. Section 2.3.1. "In summary, among the 91 days of IOP-1, 19% were classified as continental days, 32% as sea breeze days and 49% as marine days". You could say these percentages for the dry and wet periods? Or the rain for each of the 3 types of days? Somehow the info of the day types and the info on rain (dry-wet) should be linked. This is related to Fig 2 as well.

7. Section 3.1.1. "the fraction of unaccounted material therefore corresponded to DD and SS contributions". Note that if Fe is 2-5% of DD, then according to Fe concentrations, DD>unaccounted mass.

8. Section 3.1.1. "The unaccounted fraction (determined as the difference between the gravimetrically measured PM1 mass concentration and the sum of chemical species from ACSM and aethalometer measurements) corresponds to 27%, 26% and 16% of the PM1 mass for continental, sea breeze and marine days, respectively (see Figure S2)". Please specify that for these numbers you already applied the model from Fialho for this calculation, so that you derived already BC and Fe concentrations from the aethalometer measurements.

9. Section 3.1.1. Related with comment 4, please comment on the uncertainty of ACSM measurements since you took RF of NO3 as an average of previous calibrations and not determined on site.

10. Section 3.1.2. "Although a weak correlation (r = 0.55) was found between Fe and total PM1 concentrations, Fe concentrations showed higher correlations with PM10 (r = 0.70, see Figure 4)". You could check the correlation between Fe and PM1-ACSM-BC.

11. Section 3.1.2. "the highest Fe concentrations (> 8.0 $\mu$g m-3) are generally associated with continental and sea breeze days. These maxima also coincide with PM10 highest concentrations (> 400 $\mu$g m-3)". Note that 8 $\mu$g m-3 of Fe corresponds to 160 $\mu$g m-3 of DD (if Fe is 5% of DD on average according to literature values). Even if we assume a high percentage of refractory PM1 for this data point (higher than the average 29% according to data in page 9, line 24), let's estimate 60% of PM1 is refractory, this would mean that the PM1 is 267 $\mu$g m-3. If PM10 is 400 $\mu$g m-3, then the ratio PM1/PM10 for this event would be 67%, much higher than the average 10% reported. Is this the case? Please check for consistency. Either Fe is overestimated, or PM1 is underestimated, or both.

12. Section 3.1.2. Last paragraph. "Fe contributions to PM10 estimated in M'Bour (average Fe/PM10 ratio of 0.51% over IOP-1 and 0.89% for continental days)". The Fe concentrations determined in this study correspond to PM1, since the aethalometer was equipped with a PM1 inlet. If this is correct, then the authors are taking the Fe in PM1 with respect to bulk PM10, whereas the Fe concentration in PM10 corresponding to the Fe concentrations in PM1 determined in the present study would be much higher. Hence the comparisons with the ratios of Fe/DD or Fe/soil in the literature are not direct. Regarding the sentence "Nonetheless they (Formenti et al in PM40) measured for the same samples an averaged iron concentration of 10 $\mu$g m-3, in the same order of magnitude as our maximum concentration of 11.2 $\mu$g m-3 in PM1"; this is not directly comparable, Fe in PM40 with Fe in PM1.

13. Section 3.1.2. To compare with % determined in DD or soil samples, the ratios that should be taken from this study are the Fe/(PM1-ACSM-BC), assuming PM1-ACSM-BC a proxy for DD if we disregard sea salt, as DD or soil samples do not have the NR components that we have in the PM1 in this study. This ratio (Fe/(PM1-ACSM-BC)) is 20% aprox for IOP-1 and about 23% for continental days (according to Figure S2).

14. Section 3.1.3. "regional background sites such as MontSec, Spain". Consider replacing regional by continental, since Montsec site is defined as continental back-

ground site in Ripoll et al, 2015.

15. Section 3.1.4. The wind is always from the North (NW-NE), so the variation along the day cannot be explained by transport only, since the transport takes place the entire daytime (during the night the wind velocity is lower, so this can partially explain some variation).

16. Section 3.2.2. "The HOA rose plot shows marked peaks in the directions of the two open waste burning areas and of the fish-smoking area located northeast of the site in the outskirts of M'Bour". Can you please give a tentative explanation for this?

17. Section 3.2.2. Could you comment on the 5-factors solution constraining HOA and COA? Do you get LCOA and 2 different OOA factors? If this is the case, you could see different origins for OOA, ideally locally formed versus transported? Or is the 5-factors solution resulting in a mix LCOA-OOA factor not well defined?

18. Section 3.2.2. The COA profile does not meet the 41>43 characteristic of the COA. Do you have any comment on this?

19. Section 3.2.2. "Our measurements over a large period of four months". Please re-write to state the 3 months period. Maybe 3 months cannot be considered a large period (true it is larger than typical 3 weeks campaign for AMS, but it is not very large).

20. Conclusions section. "during four months of the 2015 dry season". Please revise if you wanted to say dry season, or dry+wet. Please correct the duration to 3 months.

21. Conclusions section. "This factor (LCOA), although minor on average, could represent as high as 7% on a 30-minute time period when the air masses were blowing from the waste burning areas". Isn't it even more as a maximum? The average contribution of LCOA for marine events is 7%, so there must be some individual 30-min data points with a higher contribution. Or are these points you mention when air masses blow from the waste burning areas not taking place during marine-classified days? Maybe worth to clarify this. Moreover, maybe worth to clarify also the differences in absolute

contributions, although the percentage is higher for marine days (by a factor of 2 or 3 with respect to other days), the absolute contribution is not so much higher. Still the absolute contribution for marine is higher (continental: 5.98 $\mu$g m-3 x 36% of OA x 2% of LCOA= 0.04 $\mu$g m-3 of LCOA; sea breeze: 6.29 $\mu$g m-3 x 40% of OA x 3% of LCOA=0.08 $\mu$g m-3 of LCOA; marine: 6.09 $\mu$g m-3 x 25% of OA x 7% of LCOA=0.11 $\mu$g m-3 of LCOA) (calculations to be improved with the corresponding decimals and not rounded values taken from the plots). Why the absolute average contribution of LCOA is higher for marine days even that during marine days the wind is never coming from the identified waste burning sources, according to map and wind rose in Figs 1 and 2?

22. Figure 5. Maybe choose a different color for Fe (in print it looks same as sulfate).

23. Figure 7a. Consider choosing a different scale for OM and (SO4, NH4, NO3 and Chl) to help seeing the variations, not very evident now for components different from OM.

24. Figure 7b. Fe correlates with BC. This is an indication that the Fe calculation should be revised. The explanation of the co-transport of BC and Fe may not explain completely this parallel behavior. You could isolate the dust events and see the differences in Fe and BC ratios.

25. Figure S6. Should legend in first plot read LCOA instead of WCOA?

---

## Referee Comment (RC2) · Anonymous Referee #1 · 3 Apr 2017

Manuscript is a well-written and organized and it offers important information about the aerosols and their sources in West-Africa, Senegal. I recommend it for publishing in ACP after addressing some comments below.

1. Aerosol acidity approach (Chapter 3.1.3, equation 11?) is valid if the influence of metal ions, as well as organic acids and bases on $NH_4^+$ concentration is negligible (Zhang et al., 2007). When the sulfate to $NH_4^+$ ratio is high, the amount of atmospheric ammonium is not sufficient to neutralize all $SO_4^{2-}$, $NO_3^-$ and $Cl^-$ anions. In such a case at least a fraction of $NO_3^-$ and $Cl^-$ anions must be associated with cations other

than NH4+ and Eq. (11) is no longer valid. This should be discussed and clarified in the revised manuscript.

2. Recently, it has been demonstrated that some inorganic salts (e.g. (NH4)2SO4) have a positive bias on the CO2+-signal through reaction on the aerosol mass spectrometer vaporizer (Pieber et al., 2016). This interference is highly variable between instruments and with measurement history. How big is this interference value for your ACSM and possible impacts on data analyses including PMF.

3. How can authors explain high NO3-concentrations in air masses from South-West? The air mass history can be investigated e.g. using backtrajectories calculated with the HYbrid Single-Particle Lagrangian Integrated Trajectory (HYSPLIT).

4. What are the following steps for aerosol research in this region of the world? Are these results valuable for policy makers to guide cleanup and decision making for future industry? Please, improve discussion.

References

Pieber, S. M., El Haddad, I., Slowik, J. G., Canagaratna, M. R., Jayne, J. T., Platt, S. M., Bozzetti, C., Daellenbach, K. R., Fröhlich, R., Vlachou, A., Klein, F., Dommen, J., Miljevic, B., Jimenez, J. L., Worsnop, D. R., Baltensperger, U., Prévôt, A. S. H. Inorganic Salt Interference on CO2+ in Aerodyne AMS and ACSM Organic Aerosol Composition Studies. Environmental Science and Technology 50, 10494-10503, 2016.

Zhang, Q., Jimenez, J. L., Worsnop, D. R., and Canagaratna, M.: A case study of urban particle acidity and its influence on secondary organic aerosol, Environ. Sci. Technol., 41, 3213–3219, 2007.

---

## Referee Comment (RC3) · Anonymous Referee #3 · 4 Apr 2017

Overview:

This manuscript presented the highly-time resolved chemical characterization and source apportionment of atmospheric submicron aerosol particles (PM1) in West Africa, along with field on-line measurements (including an ACSM and a 7-wavelength aethalometer) and offline model analysis. The campaign was deployed under the environment affected by anthropogenic emissions (e.g., traffic, cooking, and biomass burning) and natural sources (e.g., desert dust and marine air masses), etc.. The results showed that the ten times lower average concentrations of NR-PM1 were observed

here compared to the results from other megacities with persistent air pollution issues, i.e., Beijing and Paris. Sea breeze phenomena and Saharan desert dust outbreaks may lead to pollution events with high concentrations of PM10 (up to 900 $\mu$g/m3). Organic matter (OM) and sulfate could dominate the major fraction of aerosol particles when air massed could be associated to different influences, i.e., continental and sea breeze and oceanic region related. The authors also estimated the mass concentrations of particulate Fe from the Aethalometer data, for which an average contribution (4.6%) of Fe to PM1 was obtained. A new organic aerosol factor (LCOA), Local Combustion Organic Aerosol, was resolved by PMF analysis, with relatively low contribution (3%). Both regional and local photochemistry processes could contribute the formation of oxygenated organic aerosols in this area. The results seem to be interesting. The manuscript is well written and organized. I would recommend this paper could consider to be published in ACP once the following comments are addressed.

Specific comments:

1. Page 4, line 30: Should keep the same abbreviations for those species throughout the manuscript. For example, what's the different between "NO3-" and "NO3" (Page 5 and line 5), and somewhere else "SO42-" and "SO4", "Cl-", "Cl" and "Chl", etc.. If they are different for the discussion in this manuscript, please the authors give the related text to explain them.

2. Page 5, line 3: It was interesting to perform the chloride calibration with ammonium chloride particles. Could the authors also present the related calibration results in supporting information, as showing in Fig. S1, since your RIEChl is much higher than the default value (1.3). In addition, did the authors try to validate chloride data based on your calibration results? It's also interesting to know how it works about the chloride calibration.

3. Page 7, line 32: Would it be possible that the authors could give uncertainties of estimated Fe concentrations with this method for your study?

4. Page 8, line 19: In this paragraph, I suggest that the authors could consider to also mention some brief information about ME-2 algorithm how it works for constraining organic aerosol factors (Canonaco et al., 2013), since this will be helpful and easier for readers to quickly understand the SoFi how it works in this study.

5. Page 10, line 20: The authors should consider to explain which kind of combustion sources for the ACSM m/z 57 tracer is. When I am reading here, I immediately realized that why the authors did not perform the source apportionment of black carbon by using the Aethalometer model (Sandradewi et al., 2008). As described in Section 2.1, there are both traffic and biomass-burning emissions that would potentially contribute the ambient black carbon burden at this sampling site. So, is it possible to identify black-carbon aerosols related to traffic and biomass-burning emissions here? This will be very helpful for the source apportionment of organic aerosol.

6. Page 11, line 3: Here is a little bit confusion about the ratio of Fe/PM10, since your Fe concentrations were estimated by the PM1 aethalometer, but not PM10 Fe. Is this the case? If yes, the authors could consider to add "PM1" in front of "Fe" when you discuss this ratio in the main text of the manuscript.

7. Page 11, lines 27-28: Why didn't the authors consider to give the contribution of chloride to total NR-PM1? I suggest the authors could also mention it.

8. Page 11, lines 32-34: Is there evidence to support this discussion? Otherwise, the authors should give related reference(s).

9. Page 14, Section 3.2.1: It would be more convinced about the discussion on geographical origins, if the authors would also combine with some modeling methods, e.g., potential source contribution function (PSCF) that might be easily performed on highly-time resolved data (Petit et al., 2017).In addition, it could be also interesting to perform PSCF on organic aerosol factors.

10. Page 15, lines 24-25: Should "CnH2n-1" and "CnH2n+1" be "CnH2n-1+" and

"CnH2n+1+", respectively?

11. Page 15, line 13: Change "Source apportionment" to "Source apportionment of OM".

12. Page 15, Section 3.2.2: The biggest question I have in this manuscript is about the PMF-OA solution.

1) Page 15, lines 15-17: Please the authors could also present the results of 3 – 10 factors from the PMF-free runs in supporting information.

2) About the LCOA factor related to local open waste-burning emissions, did the authors try to compare it with any tracers? For example, please try to do some correlation analysis between LCOA with f60 (an AMS/ACSM biomass-burning tracer), f36 (as mentioned by the authors), and black-carbon aerosols from different sources, etc.. At the same time, the authors did not find any reasonable BBOA profile instead of LCOA factor. Did the authors try to compare the LCOA profile with any BBOA related profiles? What's different from them? And did the authors find any similar such mass spectrum of LCOA during your unconstrained runs and/or constrained HOA and COA runs, respectively? I am thinking whether the LCOA factor is a kind of splitted factors due to constrained HOA and COA? Could the authors give those results from your 3-10 factor runs and make comparison with your LCOA profiles. Based on the results of Fig. S6, the constrained five-factor solution seems also good. Why didn't the authors choice this one for the finally PMF-factors solution? Did the authors also try to constrain all BBOA, COA, and HOA factors together to check the five-factor solution? And the authors could also try to just constrain BBOA and HOA factors to check the results. There is in factor that the BBOA factor/profile could be affected when HOA and COA factors were constrained together.

3) In addition, could the authors explain why the median mass concentrations of COA and HOA show relatively similar diurnal variations (Fig. 9), and the similarly high peaks of HOA, COA, and LCOA around 8 am? I suggest that the authors could also make

correlation analysis among HOA, COA, and LCOA each other in supporting information. In addition, the authors should also consider to perform the source apportionment of black carbon as mentioned above. It will be very useful to evaluate the PMF solution by checking the relationships between HOA, COA, and LCOA with black carbon from fossil fuel combustion and biomass burning, respectively.

4) Why did the authors select a-value = 0.6 for both HOA and COA factors (Fig. 9)? I suggest that the authors could also perform the sensitivity test of a-values (e.g., from 0 to 1, with delta a = 0.1/0.05) on HOA and COA factors for your data sets. And why did the authors apply the reference mass spectrum of COA from HR-AMS (Crippa et al., 2013) for ME-2 constraining runs, but not from the ACSMs (Fröhlich et al., 2015;Ng et al., 2011)? Please check "Ng et al., 2016" in the plot?

5) Page 18, lines 3-23: It would be also interesting to discuss the different types of OOA factor, i.e., LO-OOA and MO-OOA, as showing in Fig. S6. Why didn't the authors keep both them for OOA factors in the final PMF solution? The authors would consider to try to take a look at the relationship between OOA and Fe concentrations. This might make sense to find something new.  

References:

Canonaco, F., Crippa, M., Slowik, J. G., Baltensperger, U., and Prévôt, A. S. H.: SoFi, an IGOR-based interface for the efficient use of the generalized multilinear engine (ME-2) for the source apportionment: ME-2 application to aerosol mass spectrometer data, Atmos. Meas. Technol., 6, 3649-3661, 10.5194/amt-6-3649-2013, 2013.

Crippa, M., El Haddad, I., Slowik, J. G., DeCarlo, P. F., Mohr, C., Heringa, M. F., Chirico, R., Marchand, N., Sciare, J., Baltensperger, U., and Prévôt, A. S. H.: Identification of marine and continental aerosol sources in Paris using high resolution aerosol mass spectrometry, J. Geophys. Res. Atmos., 118, 1950-1963, 10.1002/jgrd.50151, 2013.

Fröhlich, R., Crenn, V., Setyan, A., Belis, C. A., Canonaco, F., Favez, O., Riffault, V.,

[Figure]

Slowik, J. G., Aas, W., Aijälä, M., Alastuey, A., Artiñano, B., Bonnaire, N., Bozzetti, C., Bressi, M., Carbone, C., Coz, E., Croteau, P. L., Cubison, M. J., Esser-Gietl, J. K., Green, D. C., Gros, V., Heikkinen, L., Herrmann, H., Jayne, J. T., Lunder, C. R., Minguillón, M. C., Močnik, G., O'Dowd, C. D., Ovadnevaite, J., Petralia, E., Poulain, L., Priestman, M., Ripoll, A., Sarda-Estève, R., Wiedensohler, A., Baltensperger, U., Sciare, J., and Prévôt, A. S. H.: ACTRIS ACSM intercomparison – Part 2: Intercomparison of ME-2 organic source apportionment results from 15 individual, co-located aerosol mass spectrometers, Atmos. Meas. Tech., 8, 2555-2576, 10.5194/amt-8-2555-2015, 2015.

Ng, N. L., Canagaratna, M. R., Jimenez, J. L., Zhang, Q., Ulbrich, I. M., and Worsnop, D. R.: Real-Time Methods for Estimating Organic Component Mass Concentrations from Aerosol Mass Spectrometer Data, Environ. Sci. Technol., 45, 910-916, 10.1021/es102951k, 2011.

Petit, J. E., Favez, O., Albinet, A., and Canonaco, F.: A user-friendly tool for comprehensive evaluation of the geographical origins of atmospheric pollution: Wind and trajectory analyses, Environ. Modell. Softw., 88, 183-187, http://dx.doi.org/10.1016/j.envsoft.2016.11.022, 2017.

Sandradewi, J., Prévôt, A. S. H., Szidat, S., Perron, N., Alfarra, M. R., Lanz, V. A., Weingartner, E., and Baltensperger, U.: Using Aerosol Light Absorption Measurements for the Quantitative Determination of Wood Burning and Traffic Emission Contributions to Particulate Matter, Environ. Sci. Technol., 42, 3316-3323, 10.1021/es702253m, 2008.

---

## Author Response (AR1)

Journal: ACP
Title: Chemical characterization and source apportionment of submicron aerosols measured in Senegal during the 2015 SHADOW campaign
Author(s): Laura-Hélèna Rivellini et al.
MS No.: acp-2016-1127

The authors want to thank Reviewer #1 for his/her helpful comments. They are addressed below in blue. Changes in the manuscript are written in red.

**Anonymous Referee #1**

Manuscript is a well-written and organized and it offers important information about the aerosols and their sources in West-Africa, Senegal. I recommend it for publishing in ACP after addressing some comments below.

1. Aerosol acidity approach (Chapter 3.1.3, equation 11?) is valid if the influence of metal ions, as well as organic acids and bases on NH4+ concentration is negligible (Zhang et al., 2007). When the sulfate to NH4+ ratio is high, the amount of atmospheric ammonium is not sufficient to neutralize all SO4, NO3 and Cl anions. In such a case at least a fraction of NO3 and Cl anions must be associated with cations other than NH4+ and Eq. (11) is no longer valid. This should be discussed and clarified in the revised manuscript.

**Author's response:** 23% of the data are associated to a $NH_{4,meas}/NH_{4,predict}$ ratio inferior to 0.75 and correspond to points under the 1:1 line in Figure S3a. For such points, the amount of $NH_4$ predicted is overestimated in comparison to the amount measured on site, and considering that $NH_4$ will preferentially react with $SO_4$, we agree with Reviewer #1 that other anions like $NO_3$ and Chl are partially under chemical states other than ammonium nitrate and chloride.

Chloride species have been observed in this study as emitted by local combustion processes (m/z 36, $HCl^+$). Besides, $HNO_3$ adsorption on dust as already been reported in the literature (Fairlie et al., 2010; Savoie et al., 1989) and this is consistent with ratio values inferior to 0.75 which are mainly observed while the site is under dust influence (Figure S3b).

If only $SO_4$ is taken into account in the ion balance as described in Tiitta et al. (2014) who observed a lack of ammonium to fully neutralize the inorganic anions during the wet season, the $NH_{4,meas}/NH_{4,predict}$ ratio increases up to 1.20 (see Figure R1 below). This clearly indicates that sulfates are fully neutralized in our case and that additional $NH_4$ is available to neutralize at least partially other inorganic anions.

[Figure]

Figure R1. Scatter plot of measured-to-predicted $NH_4$ using only $SO_4$ in the ion balance equation. Data points are colored by sulfate concentrations.

**Changes in the manuscript:** The equation number associated with the neutralization equation has been corrected from (10) to (11), page 12 line 29.

The following paragraph has been added page 13 line 2: "On the other hand, 23% of the data are associated to a $NH_{4,meas}/NH_{4,predict}$ ratio inferior to 0.75 and correspond to points under the 1:1 line in Figure S3a. For such points, the amount of $NH_4$ predicted is overestimated in comparison to the amount measured on site, and considering that $NH_4$ will preferentially react with $SO_4$, other anions like $NO_3$ and Chl are partially under chemical states other than ammonium nitrate and chloride. As mentioned previously and developed in section 3.2, this can be explained by chloride species emitted by local combustion processes but also by $HNO_3$ adsorption on dust as already reported in the literature (Fairlie et al., 2010; Savoie et al., 1989) and this is consistent with ratio values inferior to 0.75 which are mainly observed while the site is under dust influence (Figure S3b). Nevertheless these periods also correspond to low levels of inorganic species in $PM_1$."

2. Recently, it has been demonstrated that some inorganic salts (e.g. (NH4)2SO4) have a positive bias on the $CO_2^+$ signal through reaction on the aerosol mass spectrometer vaporizer (Pieber et al., 2016). This interference is highly variable between instruments and with measurement history. How big is this interference value for your ACSM and possible impacts on data analyses including PMF.

**Author's response:** Pieber et al. (2016) have evidenced that the bias is significant when the inorganic fraction is > 50%, 3-10 times more important for ammonium nitrate than for sulfate, and also dependent on the instrument history.
The dataset has been acquired in 2015, when no calibration performed at that time took into account the impact of $CO_2^+$ from inorganic salts on m/z 44 (using the automated procedure of the instrument). For this reason we are not able to determine the b value for this specific campaign, that would allow to estimate the magnitude of the bias and introduce an appropriate correction into the fragmentation table. Nevertheless, the maximum of possible interferences on m/z 44 (observed when $NO_3/OM$ and $SO_4/OM$ ratios are high) also correspond to lower $CO_2^+$ mass concentrations (Figure R2, left). Moreover our last ammonium nitrate calibration (performed recently at a site in Northern France after 6-month sampling of air masses showing high levels of ammonium nitrate), led to an estimation of the m/z 44 vs $NO_3$ ratio of 9% (Figure R2, right).

[Figure]

Figure R2. (left) Ratios of $NO_3$ and $SO_4$ over OM colored by $CO_2^+$ signal and (right) m/z 44 vs $NO_3$ signals (in Amps) obtained during $NH_4NO_3$ calibration.

Based on this measurement, and considering the rather low levels of $NO_3$ observed at M'Bour and the predominance of $SO_4$ in the inorganic fraction, we consider this interference on m/z 44 as likely negligible.

3. How can authors explain high NO3-concentrations in air masses from South-West? The air mass history can be investigated e.g. using backtrajectories calculated with the HYbrid Single-Particle Lagrangian Integrated Trajectory (HYSPLIT).

**Author's response:** We tend to attribute these $NO_3$ concentrations from the SW wind sector (oceanic) to $NO_x$ emissions from Dakar which are transported and transformed above the ocean before reaching our sampling site. Adon et al. (2016) have observed an annual average for $NO_2$ concentrations as high as 32 ppb (60 µg m$^{-3}$). This regional transport is supported by $NO_3$ NWR plot and PSCF map now added in the Supplementary Information, Figure S8.

4. What are the following steps for aerosol research in this region of the world? Are these results valuable for policy makers to guide cleanup and decision making for future industry? Please, improve discussion.

**Author's response:** We have added some additional comments in the Conclusion section regarding Reviewer #1's request.

**Changes in the manuscript:** The following paragraph has been added page 19 line 31: "As shown during this field campaign, at least half of the organic aerosols measured in the submicron fraction are from anthropogenic origins (HOA + COA + LCOA) and we were able to attribute them to specific sources. On the contrary, little is known about the oxygenated fraction – often associated to secondary organic aerosols, which constitutes the other half of OA and therefore efforts should be directed toward better characterizing SOA precursors (anthropogenic and biogenic) and their concentration levels in West Africa. Moreover, the specific LCOA source puts an emphasis on open waste burning, which is highly problematic in terms of health issues, and should be addressed through the implementation of waste disposal facilities and an effective waste collection infrastructure."

····································································································

**References cited in this reply**

Adon, M., Yoboué, V., Galy-Lacaux, C., Liousse, C., Diop, B., Doumbia, E. H. T., Gardrat, E., Ndiaye, S. A. and Jarnot, C.: Measurements of $NO_2$, $SO_2$, $NH_3$, $HNO_3$ and $O_3$ in West African urban environments, Atmos. Environ., 135, 31–40, doi:10.1016/j.atmosenv.2016.03.050, 2016.

Fairlie, T. D., Jacob, D. J., Dibb, J. E., Alexander, B., Avery, M. A., van Donkelaar, A. and Zhang, L.: Impact of mineral dust on nitrate, sulfate, and ozone in transpacific Asian pollution plumes, Atmos Chem Phys, 10(8), 3999–4012, doi:10.5194/acp-10-3999-2010, 2010.

Savoie, D. L., Prospero, J. M. and Saltzman, E. S.: Non-sea-salt sulfate and nitrate in trade wind aerosols at Barbados: Evidence for long-range transport, J. Geophys. Res. Atmospheres, 94(D4), 5069–5080, doi:10.1029/JD094iD04p05069, 1989.

Journal: ACP
Title: Chemical characterization and source apportionment of submicron aerosols measured in Senegal during the 2015 SHADOW campaign
Author(s): Laura-Hélèna Rivellini et al.
MS No.: acp-2016-1127

The authors want to thank Reviewer #2 for his/her helpful comments. They are addressed below in blue. Changes in the manuscript are written in red.

**Anonymous Referee #2**

1. Please consider revising the calculated Fe concentrations and/or the PM1 concentrations and/or the ACSM concentrations (RF for NO3). These data together have inconsistencies that need to be addressed or commented and the possible sources of uncertainty should be stated. The authors report a mean value of 4.6% for the Fe/PM1 ratio. This means 16% for the Fe/RefractoryPM1, since 71% of $PM_1$ is NR-PM1. (Or 20% for the ratio Fe/(PM1-ACSM-BC) according to Figure S2, considering 5% of Fe and 25% of PM1-ACSM-BC). Hence, it is this 16% (or 20%) which should be compared to the data in the literature, given that the literature data that the authors quote make reference to Dust (and not total PM, regardless of the size fraction). Please see some additional comments related to this one below.

**Author's response:**
The reviewer is correct in stating that comparison to literature data should be made considering the iron content in dust (and not in total $PM_1$). The dust amount in our case corresponds to the sum of the unaccounted fraction (assuming negligible influence of sea salt), and Fe obtained by deconvolving absorption measurements, that is to say 25% (Figure S2). The Fe/(Fe + Unacc.) ratio is therefore 20% on average (varying between 12% for marine and 23% for continental days on average, Figures R1a-c).

Table R1 below summarizes the iron content determined in Saharan samples, which shows that the relative contribution of iron determined in this work is in the same order of magnitude but still significantly higher. However iron oxides can be found mostly (for ~2/3) in the clay fraction (~$PM_{2.5}$) and ~1/3 in the silt (coarse) fraction (Journet et al., 2014; Kandler et al., 2009), which is consistent with increased ratios in the submicron fraction compared to larger ones. It is also worth noting that Val et al. (2013) measured the iron content in the ultrafine and fine fractions (corresponding to $PM_1$) of particles collected in Dakar, and measured a ratio in the upper range of those already reported in the literature, even in the absence of dust event.

**Table R1.** Comparison of iron content (in %) determined in Saharan dust and soil samples

| Reference | Location | Method [a] | Size fraction | %Fe [b] |
|---|---|---|---|---|
| Dust samples | | | | |
| (Lafon et al., 2004) | Banizoumbou (Niger) | XRF; CBD | TSP | 6.3; 7.8 |
| (Lafon et al., 2006) | Banizoumbou, | XRF; CBD | TSP | 4.3 – 6.1 |
| (Lafon et al., 2006) | Cape Verde | XRF; CBD | TSP | 5.3 – 6.0 |
| (Formenti et al., 2008) | Banizoumbou | CBD | 40 μm | 5.8 |
| (Val et al., 2013) | Dakar (Senegal) | ICP-MS | 1 μm | 7.8 |
| This work | M'Bour | cf. text | 1 μm | 23 (continental) 21 (sea breeze) 16 (marine) |
| Soil samples | | | | |
| (Moreno et al., 2006) | Saharan region (9 samples) | ICP-AES/ ICP-MS | TSP | 2.0 – 4.7 |
| (Lafon et al., 2006) | Banizoumbou, | XRF; CBD | 10.2 μm [*] 2.5 μm [*] | 5.3 5.8 |
| (Joshi et al., 2017) | M'Bour, Bordj (Algeria), Nefta (Tunisia) | XRD | 100 μm | < 0.5 |

[a] XRF: X-ray Fluorescence (XRF) Spectrometry for elemental analysis; CBD: chemical method based on citrate-bicarbonate-dithionite (CBD) reagent for quantification of iron oxides adapted from soil analysis (Mehra and Jackson, 1960)

[b] Percentages of iron relative to the mass of all oxides, classically taking into account $Na_2O$, $MgO$, $Al_2O_3$, $SiO_2$, $K_2O$, $CaO$, $TiO_2$ and $Fe_2O_3$.

[*] Soil samples resuspended using wind tunnel and collected with a 13-stage impactor

The uncertainties in the calculation of the Fe/(Fe + Unacc.) ratio can come from the measurements themselves (those for the ACSM, in particular regarding RF($NO_3$), are detailed in the reply to comment 4 and will influence the determination of the unaccounted fraction); but are mostly related to the BC and dust absorption Angström exponents (AAE) corresponding to α and β values, respectively, in the deconvolution method. This method is indeed highly sensitive to even small variations of these parameters, with values quite well known for BC from fossil fuel ranging from 0.8 to 1.1 (Hansen, 2005; Zotter et al., 2017) but not so much for dust. In the manuscript, we chose to use β = - 4, according to Fialho et al. (2006) values determined at the Azores Islands for samples influenced by Saharan dust events. But other values can be found in the literature (Table R2), ranging from -1.6 to -6.5 and largely influenced by the wavelength range as well as dust origins and size fractions since the iron content differ depending on emission sources and particle size (Journet et al., 2014). Even during the SAMUM campaign (May to June 2006 in Morocco), a wide range of AAE values have been reported from 1.6 up to 5.1 for ground-based measurements in the same size fraction, as shown in Table R2.

**Table R2**. Mineral dust AAE values reported from field campaigns around the Saharan region.

| Reference | Location / Period | Wavelengths (nm) | Fraction | β |
|---|---|---|---|---|
| Fialho et al. (2006) [a] | Azores Islands Jul. 2001 – Jun. 2005 | 370-950 | - | -4 |
| Müller et al. (2009) [a] | Tinfou, Morocco (SAMUM) Summer 2006 | 467/660 | $PM_{10}$ | -2.25 to -5.13 |
| Petzold et al. (2009) [b] | South-East Morocco (SAMUM) Summer 2006 | 467/660 | $PM_{2.5}$ | -2 to -6.5 |
| Schladitz et al. (2009) [a] | Tinfou, Morocco (SAMUM) Summer 2006 | 537/637 | $PM_{10}$ | -1.6 to -4.73 |
| (Linke et al., 2006) [c] | Morocco Egypt | 266/532 | $\sim PM_4$ | -4.2 -5.3 |
| (Caponi et al., 2017) [c] | Morocco Lybia Algeria Mali | 375-850 375-532 375-850 375-532 | $PM_{2.5}$ ($PM_{10.6}$) | -2.6 -4.1 (-3.2) -2.8 (-2.5) -3.4 |

[a] In situ ground-based measurements; [b] Airborne measurements through dust plumes; [c] Laboratory experiments with resuspended soil samples

[Figure]

**Figure R1.** Box plots of (a) Fe, (b) BC concentrations and (c) Fe/(Fe+Unacc.) ratio for continental, sea breeze and marine days. (d) Scatter plot of iron concentrations (in μg m⁻³) obtained from Fialho's deconvolution method using an AAE value of ± 10% compared to the one from the literature and used in the manuscript.

Applying a relatively small increase (resp. decrease) of 10% on the value of β for our dataset led to a 33% decrease (resp. 50% increase) of iron concentrations, as shown in Figure R1d, but no change in the temporal behavior.

In conclusion, the approach used here leads to an estimate of the absolute concentrations of iron, although with high uncertainties given all the necessary assumptions and the empirical algorithm used to deconvolve BC and Fe from absorption measurements. However the temporal profiles, non-parametric wind regression (NWR) plots and potential source contribution function (PSCF) maps (now provided in Figures S5b and S5c, respectively) are all consistent with the expected behavior of such a desert dust tracer and show that it can be useful in determining the contribution of dust to absorption measurements (see also reply to comment 24). We nonetheless agree with reviewer #2 that there is quite some room for improvement, in particular for a better estimation of the AAE value for dust similar to the efforts carried out to determine the AAE values for BC from fossil fuel and wood burning (Zotter et al., 2017). We strongly believe the lack of information for submicron particles in terms of chemical composition of refractory species and optical properties should be better addressed, but is beyond the scope of this work.

**Changes in manuscript**:
A new appendix (S2) in the Supplementary Information now includes the whole discussion above. Changes in the main text have been also done page 8, line 18 with a new sentence added: "Applying the propagation for uncertainties approach on the values of $K_{Fe}$ (10%) and the slope b (39%, calculated using a variability of 0.2 for α and β (Fialho et al., 2006)) gives an overall uncertainty of ~40% for iron concentrations. However the deconvolution algorithm is highly sensitive to the values of the Angström absorption exponents (α and β) and a more detailed discussion can be found in Appendix S2."

2. The sampling period is actually 3 months (20 March to 22 June), and not 4 as stated.

**Author's response**: Page 1 line 27, this mistake has been corrected.

3. Please homogenize the dry and wet period definition: in the introduction it says that the dry season extends from November to May; in section 2.1 it says that the dry season is from December to March; in section 2.3.1 it says that dry season is generally defined from November to April; in the same section 2.3.1 it is coherent within the section and it says that "Our study taking place from March to June allowed for the observation of both the late part of the dry season (March-April) and the beginning of the wet season (May-June)"; in the conclusions section it says "during four months of the 2015 dry season". Given that precipitation data for the specific campaign is available, according to section 2.2.3, could you please provide this data, or make the classification based on these data? (Although the info in literature about the usual dry-wet periods can still be included).

**Author's response**:
We understand our wording can be confusing. The two contrasted dry and wet seasons observed around the Equator originate from the closeness of the Intertropical Convergence Zone (ITCZ), which brings moist air masses and heavy precipitations. Kaly et al. (2015), based on 5 years of observations (2006-2010) at M'Bour, defined the dry season as the period during which no precipitation occurs from November to April and the wet season from May to October, where significant precipitation is measured, with a transition during April/May.
In Mortier et al. (2016), who analyzed data from 2006 to 2012 at M'Bour, the seasons are defined based on RH levels: from December to March/April (RH < 40%) for the dry season and from June to September (RH ~ 80%) for the wet season. They also observed different wind patterns at the ground level, that is to say trade winds coming mostly from the North-East during the dry season, whereas the wet season was characterized by winds from the west. During the AMMA field campaign in the Sahelian belt, Haywood et al. (2008) defined the period from May to June as the monsoon onset. Finally, Slingo et al. (2008) also mentioned "*large interannual variability in the seasonal progression of humidity, with no clearly reproducible pattern from year-to-year*" in Niamey, Niger.

Therefore we based the definition of the dry and wet seasons in this work on the observed weather parameters during the field campaign. Since absolutely no precipitation was observed during the whole period, but differences in RH levels (Figure R2) – though not as pronounced as reported by Mortier et al. (2016) – and wind patterns were clearly visible, we considered March and April to belong to the dry season and May-June to the transition period.

[Figure]

**Figure R2.** Time series of relative humidity (in %; 2-hour averages).

**Changes in manuscript:**
Complementary information concerning the distinction between dry and wet seasons has been added in section 2.3.1 "Classification of air masses", and some sentences have been simplified to hopefully make things clearer.
Page 2 lines 26-27, sentence modified: "During the months of January-February (dry season)"
Page 3 lines 30-31, sentence modified: "the dry season", with no mention of the month range.
Page 6 lines 16 - 24, paragraph modified: "The station of M'Bour is under the influence of a typical Sahelian climatic cycle composed of two contrasted dry and wet seasons observed around the Equator, which originate from the closeness of the Intertropical Convergence Zone (ITCZ), bringing moist air masses and heavy precipitations. Kaly et al. (2015), based on 5 years of observations (2006-2010) at M'Bour, defined the dry season as the period during which no precipitation occurs from November to April and the wet season from May to October, where significant precipitation is measured, with a transition during April/May. In Mortier et al. (2016), who analyzed data from 2006 to 2012 at M'Bour, the seasons are defined based on RH levels: from December to March/April (RH < 40%) for the dry season and from June to September (RH ~ 80%) for the wet season. They also observed different wind patterns at the ground level, that is to say trade winds coming mostly from the North-East during the dry season, whereas the wet season was characterized by winds from the west. During the AMMA field campaign in the Sahelian belt, Haywood et al. (2008) defined the period from May to June as the monsoon onset. Finally, Slingo et al. (2008) also mentioned large inter-annual variability in the seasonal progression of moisture, with no clearly reproducible pattern from year-to-year in Niamey, Niger.
Therefore we based the definition of the dry and wet seasons in this work on the observed weather parameters during the field campaign. Since absolutely no precipitation was observed during the whole period, but differences in RH levels – though not as pronounced as reported by Mortier et al. (2016) – and wind patterns were clearly visible (Figure 2a), we considered March (RH = 49%) and April (68%) to belong to the dry season, and May (82%) and June (84%) to the transition period."

4. Section 2.2.1. Please confirm that no major changes (such as filament replacement) occurred to the ACSM during the time between Feb 2014 and Jan 2015, for which the calibration constants were taken. This is necessary to be able to take the average of all calibrations. Especially between Dec 2014 and Jan 2015 there is a big difference for the RF of NO3 (Fig S1). This is a source of uncertainty that should be acknowledged in the manuscript, especially considering that the absolute concentrations are used and that the differences with the bulk PM1 concentrations are taken as very valid and interpreted.

**Author's response:**
We confirm that no major changes that could impact RF(NO₃) calibration occurred between Feb. 2014 and Jan. 2015. In particular, we have operated our ACSM with the same filament since its purchase in

2013. Additional calibrations performed since the end of the SHADOW field campaign have confirmed the stability of this value with an average of $(3.75 \pm 0.67) \times 10^{-11}$ Amps / ($\mu$g m$^{-3}$). It must be noted however that the uncertainties on mass concentrations with aerosol mass spectrometers are estimated at 20-35% ($2\sigma$) for the total mass (Bahreini et al., 2009). Furthermore, Crenn et al. (2015) reported reproducibility expanded uncertainties of Q-ACSM concentration measurements of 9, 15, 19, 28, and 36% for NR-PM$_1$, nitrate, organic matter, sulfate, and ammonium, respectively, during an intercomparison that involved 13 Q-ACSM in the Paris area during springtime.

**Changes in the manuscript**:
Page 5 lines 9 - 12, the text now reads: "It must be noted however that the uncertainties on mass concentrations with aerosol mass spectrometers are estimated at 20-35% ($2\sigma$) for the total mass (Bahreini et al., 2009). Furthermore, Crenn et al. (2015) reported reproducibility expanded uncertainties of Q-ACSM concentration measurements of 9, 15, 19, 28, and 36% for NR-PM$_1$, nitrate, organic matter, sulfate, and ammonium, respectively, during an intercomparison that involved 13 Q-ACSM in the Paris area during springtime."

5. Section 2.3.1. "During IOP-1 two main prevailing directions were found (Fig. 2a). The first one [...] and North-West to South-West (315-225_, dominant in May-June)"
Maybe you could indicate which wind direction prevails for the first and second periods (Mar-Apr and May-Jun), since the wind directions for the periods are given based on literature info, but the data for the specific campaign is available and then the wind roses are commented for the entire period.

**Author's response:**
The period from end of March to April was dominated by winds coming from NW to NE (~62%) with some occurrences (~33%) of Western winds during the sea breezes, while from May to June winds were mainly originating from the West (72%).

**Changes in the manuscript**:
Figure 2 has been modified to include wind frequency rose plots for March-April (dry season) and May-June (transition period). Besides, the following sentence has been added page 6, lines 30-31: "The period from end of March to April was dominated by winds coming from NW to NE (~62%) with some occurrences (~33%) of Western winds during the sea breezes, while from May to June winds were mainly originating from the West (72%)."

6. Section 2.3.1. "In summary, among the 91 days of IOP-1, 19% were classified as continental days, 32% as sea breeze days and 49% as marine days". You could say these percentages for the dry and wet periods? Or the rain for each of the 3 types of days? Somehow the info of the day types and the info on rain (dry-wet) should be linked. This is related to Fig 2 as well.

**Author's response:**
As indicated in reply to comment #3, no precipitation was observed during the whole campaign. Nonetheless, we provide in Table R3 below the number of days associated with the dry season (March-April) and the transition period (May-June).

**Table R3.** Number of days (relative contribution in parenthesis) associated with the continental, sea breeze and marine influences for the dry season, the transition period and the whole IOP-1.

|  | Continental | Sea breeze | Marine | Total |
|---|---|---|---|---|
| Mar-Apr. | 13 (32%) | 19 (48%) | 8 (20%) | 40 (100%) |
| May-Jun. | 4 (8%) | 10 (20%) | 37 (73%) | 51 (100%) |
| IOP-1 | 17 (19%) | 29 (32%) | 45 (49%) | 91 (100%) |

**Changes in the manuscript:**
Figure 2 in the manuscript now includes the wind roses for the dry season (March-April) and the transition period (May-June).

7. Section 3.1.1. "the fraction of unaccounted material therefore corresponded to DD and SS contributions". Note that if Fe is 2-5% of DD, then according to Fe concentrations, DD>unaccounted mass.

**Author's response:** As correctly pointed out in comment #1 by this reviewer and discussed in our reply, the Fe contribution to DD is 20% on average in $PM_1$, and not 2-5%.

8. Section 3.1.1. "The unaccounted fraction (determined as the difference between the gravimetrically measured $PM_1$ mass concentration and the sum of chemical species from ACSM and aethalometer measurements) corresponds to 27%, 26% and 16% of the PM1 mass for continental, sea breeze and marine days, respectively (see Figure S2)". Please specify that for these numbers you already applied the model from Fialho for this calculation, so that you derived already BC and Fe concentrations from the aethalometer measurements.

**Changes in the manuscript:**
Page 8 line 18, now added in the methodology section: "In the rest of the paper, when BC and Fe concentrations are mentioned, it corresponds to the deconvolved values based on the above-mentioned method."
Page 10 lines 4-7: "The unaccounted fraction was determined as the difference between the gravimetrically measured $PM_1$ mass concentration and the sum of chemical species from ACSM (Org, $NO_3$, $SO_4$, $NH_4$, Chl) and aethalometer (BC, Fe) measurements. It corresponded to 27%, 26% and 16% of the $PM_1$ mass for continental, sea breeze and marine days, respectively (see Figure S3)."

9. Section 3.1.1. Related with comment 4, please comment on the uncertainty of ACSM measurements since you took RF of NO3 as an average of previous calibrations and not determined on site.

**Author's response:**
Indeed we were not able to perform calibrations on site due to technical and regulatory constraints (for instance shipping our SMPS with a radioactive source to Senegal would have been nearly impossible). As indicated in the reply to comment #4, additional calibrations performed since the end of the SHADOW field campaign have confirmed the stability of the averaged value used for this campaign. Another (indirect) way to confirm that the calibrations are not too far off is the slope close to unity of Figure S4a that shows $NH_4$ measured vs. $NH_4$ predicted since these two parameters depend on both RF(NO3) and RIE values (at least of the main inorganic species that neutralize $NH_4$), as mentioned page 12 lines 32-34.

10. Section 3.1.2. "Although a weak correlation (r = 0.55) was found between Fe and total PM1 concentrations, Fe concentrations showed higher correlations with $PM_{10}$ (r = 0.70, see Figure 4)". You could check the correlation between Fe and PM1-ACSM-BC.

**Author's response:** The correlation observed between Fe and the unaccounted $PM_1$ is even worse for the whole IOP-1 (r = 0.47). It is partly due to the absence of TEOM-FDMS $PM_1$ measurements during the periods with major dust events (whereas TEOM $PM_{10}$ measurements were available), that leads to excluding Fe concentrations above 4 µg m$^{-3}$.

**Changes in the manuscript:**
Page 10 lines 32-35 now reads: "Although weak correlations were found between Fe and total $PM_1$ concentrations (r = 0.55) and unaccounted $PM_1$ (r = 0.47), Fe concentrations showed higher

correlations with $PM_{10}$ (r = 0.70, see Figure 4). This could be explained by the lack of $PM_1$ mass concentration measurements during intense dust events, as well as DD domination in the coarse fraction, while the fine fraction is mainly driven by NR and BC species during most of the IOP-1 (Figure 3c)."

11. Section 3.1.2. "the highest Fe concentrations (> 8.0 µg m-3) are generally associated with continental and sea breeze days. These maxima also coincide with PM10 highest concentrations (> 400 µg m-3)". Note that 8 µg m-3 of Fe corresponds to 160 µg m-3 of DD (if Fe is 5% of DD on average according to literature values). Even if we assume a high percentage of refractory PM1 for this data point (higher than the average 29% according to data in page 9, line 24), let's estimate 60% of PM1 is refractory, this would mean that the PM1 is 267 _g m-3. If PM10 is 400 _g m-3, then the ratio PM1/PM10 for this event would be 67%, much higher than the average 10% reported.
Is this the case? Please check for consistency. Either Fe is overestimated, or PM1 is underestimated, or both.

**Author's response:** During the whole IOP-1, the $PM_1/PM_{10}$ ratio average was indeed 10% but it varied between 2 and 55% for 2-hour averages. We expect it to be closer to the lower values during dust events (according to Figure 3b), for which unfortunately we had to invalidate $PM_1$ mass concentrations. Additionally, we recalculated the percentage of {Fe + Unacc.} in $PM_1$ for the highest $PM_{10}$ concentrations (> 400 µg m$^{-3}$) and found an average value of 0.77 ± 0.04, slightly higher than the 60% used in the reviewer's calculations.
Therefore, using our corrected ratio of 20% of Fe in submicron DD (see comment 1), 8 µg m$^{-3}$ of Fe corresponds to 40 µg m$^{-3}$ of DD, 52 µg m$^{-3}$ of $PM_1$ and a $PM_1/PM_{10}$ ratio of 13%, consistent with the values presented in the manuscript. Note that this is an upper value since the concentrations of $PM_{10}$ reached 950 µg m$^{-3}$ at the site, while the highest Fe concentration was 11.2 µg m$^{-3}$.

12. Section 3.1.2. Last paragraph. "Fe contributions to PM10 estimated in M'Bour (average Fe/$PM_{10}$ ratio of 0.51% over IOP-1 and 0.89% for continental days)". The Fe concentrations determined in this study correspond to PM1, since the aethalometer was equipped with a $PM_1$ inlet. If this is correct, then the authors are taking the Fe in $PM_1$ with respect to bulk $PM_{10}$, whereas the Fe concentration in $PM_{10}$ corresponding to the Fe concentrations in PM1 determined in the present study would be much higher. Hence the comparisons with the ratios of Fe/DD or Fe/soil in the literature are not direct. Regarding the sentence "Nonetheless they (Formenti et al in $PM_{40}$) measured for the same samples an averaged iron concentration of 10 µg m$^{-3}$, in the same order of magnitude as our maximum concentration of 11.2 µg m$^{-3}$ in $PM_1$"; this is not directly comparable, Fe in PM40 with Fe in $PM_1$.

**Author's response:** As mentioned in reply to comment 1 (Table 1), we only found one study that determined the iron content in $PM_1$ DD, leading to a value of 7.8% in the absence of dust events, which is within the same order of magnitude with the one found here, i.e. 20%.
We agree with reviewer #2 that our wording is confusing in this paragraph since the Fe/$PM_{10}$ ratio refers indeed to $Fe_{PM1}$ / $PM_{10}$ and not to the proportion of Fe in the $PM_{10}$ size fraction.
Direct comparisons with other size fractions cannot be straightforward since it would assume that the contribution of Fe is constant whatever the particle size, although literature has shown that, as mentioned above (comment 1), iron oxides belong mostly (for ~2/3) to the clay fraction (~$PM_{2.5}$) and ~1/3 to the silt (coarse) fraction (Journet et al., 2014; Kandler et al., 2009), which is consistent with increased ratios in the submicron fraction compared to larger ones.

**Changes in the manuscript:**
Page 11 lines 3-13: "From the only study in the literature focusing on iron concentrations in the submicron fraction in West Africa (Val et al., 2013), we could infer an elemental iron contribution of 7.8% to $PM_1$ dust, in Dakar, in the absence of dust events. Other studies focused on dust gave the iron contribution for size fractions higher than $PM_1$, thus no straightforward comparisons can be made with our average ratios of Fe/$DD_{PM1}$ (20, 23, 21 and 16% for respectively IOP-1, continental, sea breeze

and marine days). It can nevertheless be interesting to have in mind values retrieved within the same region as it is known that iron oxides mainly belong to the finest fraction (Journet et al., 2014; Kandler et al., 2009) and therefore the elemental iron contribution should be lower for larger sizes, which is consistent with values reported in Table S2.2."

13. Section 3.1.2. To compare with % determined in DD or soil samples, the ratios that should be taken from this study are the Fe/(PM1-ACSM-BC), assuming PM1-ACSMBC a proxy for DD if we disregard sea salt, as DD or soil samples do not have the NR components that we have in the PM1 in this study. This ratio (Fe/(PM1-ACSM-BC)) is 20% approx for IOP-1 and about 23% for continental days (according to Figure S2).

**Author's response:** See response to comment 1.

14. Section 3.1.3. "regional background sites such as MontSec, Spain". Consider replacing regional by continental, since Montsec site is defined as continental back-ground site in Ripoll et al, 2015.

**Changes in the manuscript:**
Page 11 line 25: "continental background sites such as MontSec"

15. Section 3.1.4. The wind is always from the North (NW-NE), so the variation along the day cannot be explained by transport only, since the transport takes place the entire daytime (during the night the wind velocity is lower, so this can partially explain some variation).

**Author's response:** We agree with the reviewer that for continental days, the wind is always blowing from the NW to NE, whereas during sea breeze days, part of the afternoon is under the influence of marine air masses, and for marine days western winds dominate.
We have indeed highlighted several times in that section that for continental days the observed temporal variations are also coming from the temporal variability of local emissions.
For instance, page 13 lines 13-14: "*suggesting an influence linked to emissions by rather local anthropogenic activities rather than long-range transport sources*"; page 13, line 28: "*probably combining local emissions and reduced dispersion*"; page 13, lines 34-35: "*These peaks are measured for air masses coming from the continent and correspond to traffic and/or cooking hours*"; page 14, line 1: "*(…) tend to confirm combustion sources for these species*".

16. Section 3.2.2. "The HOA rose plot shows marked peaks in the directions of the two open waste burning areas and of the fish-smoking area located northeast of the site in the outskirts of M'Bour". Can you please give a tentative explanation for this?

**Author's response:** HOA rose plot and NWR plot both show higher concentrations in these directions. As for the open waste burning areas, a possible explanation could be the formation of organic compounds which present molecular structures that, when fragmented by electron impact ionization, are very similar to the HOA mass spectrum for m/z below 100. This is for instance the case for phthalate esters (Wienecke et al., 1992), whose mass spectrum is shown below.

[Figure]

NIST Chemistry WebBook (http://webbook.nist.gov/chemistry)

**Figure R3.** Mass spectrum of di-n-octyl phthalate obtained by electron impact ionization

It is less obvious why HOA points out also to the fish-smoking area. We have not been able to observe the process of fish-smoking but were told that they used millet flour as fuel. Since it is extracted from the grains, it should contain almost no cellulose (0.7-1.8% according to Wankhede et al. -1979)), contrary to the stems of plants where it amounts to ~40% (Ververis et al., 2004). This could explain why we do not see any biomass burning tracer, since levoglucosan is formed by the pyrolysis of cellulose. Finally, we cannot exclude higher traffic emissions at this location during the hours when the fish is smoked, since it has to be transported from the harbour to the fish-smoking area.

17. Section 3.2.2. Could you comment on the 5-factors solution constraining HOA and COA? Do you get LCOA and 2 different OOA factors? If this is the case, you could see different origins for OOA, ideally locally formed versus transported? Or is the 5-factors solution resulting in a mix LCOA-OOA factor not well defined?

**Author's response:** The 5-factor solution presented in Appendix S6 of the submitted manuscript (now Appendix S8) was obtained with m/z 36 in the PMF input in addition to the "classical" organics matrix. This mass was consistently attributed to the LCOA factor, as mentioned in the Appendix discussion. Besides, strong constraints on the different POA factors (using profiles of HOA, COA and LCOA obtained with unconstrained solutions) were applied. In these conditions only, two different kinds of OOA factors could be deconvolved: one more oxidized (MO-OOA; 76.5% of OOA) and considered from a more regional origin (mostly marine as highlighted by its NWR plot and PSCF map in Figure R4 below) and the other less oxidized (LO-OOA; 23.5% of OOA), locally emitted as per its NWR plot. Nonetheless, without using m/z 36 as input and literature profiles for constraining HOA and COA none of the solution leads to two completely distinct OOA profiles. If considering MO-OOA only, most of it could be rather due to the oxidation of ship emissions along the Western African coast, which would also explain the better correlation observed with $NO_3$ from $NO_x$ emission processing despite the predominance of this regional oxidized factor over the local one.

[Figure]

**Figure R4.** NWR plots (top) and PSCF map (bottom) for MO-OOA (left) and LO-OOA (right) obtained with the 5-factor constrained solution including m/z 36.

**Changes in the manuscript:**
Abstract, page 2 lines 4-6: "The remaining fraction was identified as oxygenated organic aerosols (OOA), a factor that prevailed regardless of the day type (45%) and was representative of regional (~3/4) but also local (~1/4) sources due to enhanced photochemical processes."

Page 15 lines 34-36: "Since the behavior of Chl had also been suspected to come from the same sources, PMF solutions adding the m/z 36 signal in the input matrix were investigated, and a solution is presented in Appendix S8, where regional OOA accounts for ~3/4 of the OOA and local OOA ~1/4."

Page 18 line 21: "The OOA PSCF map (Figure S5c) seems to trace back its origin along the entire Western African coast, where shipping emissions could be a major source of organic aerosols."

18. Section 3.2.2. The COA profile does not meet the 41>43 characteristic of the COA. Do you have any comment on this?

**Author's response:** HOA and COA profiles were quite difficult to separate over the whole period when running unconstrained PMF, as has been shown in previous studies with Q-ACSM (Fröhlich et al., 2015 and references therein). When applying strong constraints on the primary profile, part of the m/z 43 fragment goes to HOA and OOA, as observed in the 5-factor constrained solution (Appendix S8). However this does not change significantly the contribution of this factor to the total OA.

19. Section 3.2.2. "Our measurements over a large period of four months". Please re-write to state the 3 months period. Maybe 3 months cannot be considered a large period (true it is larger than typical 3 weeks campaign for AMS, but it is not very large).

**Changes in the manuscript:**
Page 18 line 30: "over a period of three months"

20. Conclusions section. "during four months of the 2015 dry season". Please revise if you wanted to say dry season, or dry+wet. Please correct the duration to 3 months.

**Changes in the manuscript:**
Page 18 lines 35-36: "during three months encompassing the end of the dry season and the transition period toward the wet season of 2015"

21. Conclusions section. "This factor (LCOA), although minor on average, could represent as high as 7% on a 30-minute time period when the air masses were blowing from the waste burning areas". Isn't it even more as a maximum? The average contribution of LCOA for marine events is 7%, so there must be some individual 30-min data points with a higher contribution. Or are these points you mention when air masses blow from the waste burning areas not taking place during marine-classified days? Maybe worth to clarify this. Moreover, maybe worth to clarify also the differences in absolute contributions, although the percentage is higher for marine days (by a factor of 2 or 3 with respect to other days), the absolute contribution is not so much higher. Still the absolute contribution for marine is higher (continental: 5.98 µg m$^{-3}$ x 36% of OA x 2% of LCOA= 0.04 µg m$^{-3}$ of LCOA; sea breeze: 6.29 µg m$^{-3}$ x 40% of OA x 3% of LCOA=0.08 _g m-3 of LCOA; marine: 6.09 µg m$^{-3}$ x 25% of OA x 7% of LCOA=0.11 µg m$^{-3}$ of LCOA) (calculations to be improved with the corresponding decimals and not rounded values taken from the plots). Why the absolute average contribution of LCOA is higher for marine days even that during marine days the wind is never coming from the identified waste burning sources, according to map and wind rose in Figs 1 and 2?

**Author's response:** Overall the concentration of LCOA is 0.05 (11%), 0.06 (17%) and 0.07 (34%) µg m$^{-3}$ on average (maximum contribution) for continental, sea breeze and marine days, respectively. Days during which air masses were coming from the identified open waste burning areas of Gandigal or/and Saly Douté were classified either as continental or sea-breeze days. Some strong events also appear during marine days at an average distance from the site (see LCOA NWR plot in Figure S5b) and may be related to air masses carried over Dakar where similar massive anthropogenic emissions from waste burning could be expected from Mbeubeuss, the largest dumpsite in Senegal located 25 km north-east of Dakar along the coast, which receives 250,000 tons of garbage per year from the Dakar region (Cissé, 2012). In the absence of strictly controlled waste regulations, it is however quite likely there are other unidentified open waste burning sites along the coast that could also contribute to this factor. The more regional influence seen in the NWR plot may also be due to chlorine-driven photo-oxidation processes occurring off the coast of Senegal (Hossaini et al., 2016).

**Changes in the manuscript:** The dumpsite of Mbeubeuss in Dakar is now identified in Figure 1.

Page 17 line 29: "Besides, the NWR plots of Chl (local influence) and LCOA (both local and regional) rather suggest the presence of chlorinated organics. The PSCF maps identify two possible origins, one clearly from the ocean that could be related to chlorine-driven photo-oxidation processes (Hossaini et al., 2016) and the other linked to air masses carried over Dakar where similar massive anthropogenic emissions from waste burning could be expected from Mbeubeuss, the largest dumpsite in Senegal located 25 km north-east of Dakar along the coast, which receives 250,000 tons of garbage per year from the Dakar region (Cissé, 2012).

Page 19 lines 19-26: "Three primary OA linked to anthropogenic activities from nearby sources were also identified: HOA (22%), COA (28%) and a new factor LCOA (3%) related to local combustion sources (emissions from open-waste burning and fish smoking areas), for which a good correlation with particulate chloride (m/z 36) was consistently found. Non-refractory chloride fragments from waste burning or fish smoking areas were suggested to originate from local plastic smoldering/flaming processes (for the former) and/or sea salt (for both) submitted to high temperatures under continental influence. This factor, although minor on average, could represent as high as 7% on a 30-minute time period when the air masses were blowing from the local waste burning areas, and very likely resulted in the concomitant emissions of highly-toxic compounds such as dioxins that would require further investigation. Back-trajectories also suggest possible distant sources of combustion, with part of LCOA, OOA and BC associated to processed oceanic air masses which could be influenced by Dakar traffic emissions and waste burning activities, as well as shipping emissions along the West African coast."

22. Figure 5. Maybe choose a different color for Fe (in print it looks same as sulfate).

**Changes in the manuscript:** Figure 5 has been modified as suggested using dark brown for iron.

23. Figure 7a. Consider choosing a different scale for OM and (SO4, NH4, NO3 and Chl) to help seeing the variations, not very evident now for components different from OM.

**Changes in the manuscript:** Figure 7a has been modified as suggested, plotting OM on the right axis and all the other concentrations on the left one.

24. Figure 7b. Fe correlates with BC. This is an indication that the Fe calculation should be revised. The explanation of the co-transport of BC and Fe may not explain completely this parallel behavior. You could isolate the dust events and see the differences in Fe and BC ratios.

**Author's response:** A correlation coefficient of 0.55 between Fe and BC was calculated for the whole IOP-1. Both species show quite distinctive origins (Figure S5b and S5c), iron sources pointing toward the Saharan desert (PSCF map) but also attributed to more local emissions most probably caused by traffic resuspension (NWR plot). BC appeared to be emitted both by cities located along the Western African coast especially Dakar (PSCF) and by local sources and attributable to diesel combustion from traffic which could explain the common peaks encountered in the morning and the evening by the two compounds.

25. Figure S6. Should legend in first plot read LCOA instead of WCOA?
**Changes in the manuscript:** The legend of Figure S8 (formerly Fig. S6) has been corrected.

· · · · · · · · · · · · · · · · · · · · · · · · · · · · · · · · · · · · · · · · · · · · · · · · · · · · · · · ·

**References cited in this reply**

Bahreini, R., Ervens, B., Middlebrook, A. M., Warneke, C., de Gouw, J. A., DeCarlo, P. F., Jimenez, J. L., Brock, C. A., Neuman, J. A., Ryerson, T. B., Stark, H., Atlas, E., Brioude, J., Fried, A., Holloway, J. S., Peischl, J., Richter, D., Walega, J., Weibring, P., Wollny, A. G. and Fehsenfeld, F. C.: Organic aerosol formation in urban and industrial plumes near Houston and Dallas, Texas, J. Geophys. Res. Atmospheres, 114(D7), D00F16, doi:10.1029/2008JD011493, 2009.

Caponi, L., Formenti, P., Massabó, D., Di Biagio, C., Cazaunau, M., Pangui, E., Chevaillier, S., Landrot, G., Andreae, M. O., Kandler, K., Piketh, S., Saeed, T., Seibert, D., Williams, E., Balkanski, Y., Prati, P. and Doussin, J.-F.: Spectral- and size-resolved mass absorption efficiency of mineral dust

aerosols in the shortwave: a simulation chamber study, Atmos Chem Phys Discuss, 2017, 1–39, doi:10.5194/acp-2017-5, 2017.

Cissé, O.: Les décharges d'ordures en Afrique - Mbeubeuss à Dakar au Sénégal, Karthala., 2012.

Crenn, V., Sciare, J., Croteau, P. L., Verlhac, S., Fröhlich, R., Belis, C. A., Aas, W., Äijälä, M., Alastuey, A., Artiñano, B., Baisnée, D., Bonnaire, N., Bressi, M., Canagaratna, M., Canonaco, F., Carbone, C., Cavalli, F., Coz, E., Cubison, M. J., Esser-Gietl, J. K., Green, D. C., Gros, V., Heikkinen, L., Herrmann, H., Lunder, C., Minguillón, M. C., Močnik, G., O'Dowd, C. D., Ovadnevaite, J., Petit, J.-E., Petralia, E., Poulain, L., Priestman, M., Riffault, V., Ripoll, A., Sarda-Estève, R., Slowik, J. G., Setyan, A., Wiedensohler, A., Baltensperger, U., Prévôt, A. S. H., Jayne, J. T. and Favez, O.: ACTRIS ACSM intercomparison – Part 1: Reproducibility of concentration and fragment results from 13 individual Quadrupole Aerosol Chemical Speciation Monitors (Q-ACSM) and consistency with co-located instruments, Atmos Meas Tech, 8(12), 5063–5087, doi:10.5194/amt-8-5063-2015, 2015.

Fialho, P., Freitas, M. C., Barata, F., Vieira, B., Hansen, A. D. A. and Honrath, R. E.: The Aethalometer calibration and determination of iron concentration in dust aerosols, J. Aerosol Sci., 37(11), 1497–1506, doi:10.1016/j.jaerosci.2006.03.002, 2006.

Formenti, P., Rajot, J. L., Desboeufs, K., Caquineau, S., Chevaillier, S., Nava, S., Gaudichet, A., Journet, E., Triquet, S., Alfaro, S., Chiari, M., Haywood, J., Coe, H. and Highwood, E.: Regional variability of the composition of mineral dust from western Africa: Results from the AMMA SOP0/DABEX and DODO field campaigns, J. Geophys. Res. Atmospheres, 113(D23), D00C13, doi:10.1029/2008JD009903, 2008.

Fröhlich, R., Crenn, V., Setyan, A., Belis, C. A., Canonaco, F., Favez, O., Riffault, V., Slowik, J. G., Aas, W., Aijälä, M., Alastuey, A., Artiñano, B., Bonnaire, N., Bozzetti, C., Bressi, M., Carbone, C., Coz, E., Croteau, P. L., Cubison, M. J., Esser-Gietl, J. K., Green, D. C., Gros, V., Heikkinen, L., Herrmann, H., Jayne, J. T., Lunder, C. R., Minguillón, M. C., Močnik, G., O'Dowd, C. D., Ovadnevaite, J., Petralia, E., Poulain, L., Priestman, M., Ripoll, A., Sarda-Estève, R., Wiedensohler, A., Baltensperger, U., Sciare, J. and Prévôt, A. S. H.: ACTRIS ACSM intercomparison – Part 2: Intercomparison of ME-2 organic source apportionment results from 15 individual, co-located aerosol mass spectrometers, Atmos Meas Tech, 8(6), 2555–2576, doi:10.5194/amt-8-2555-2015, 2015.

Hansen, A. D. A.: Aethalometer Operations Manual, Magee scientifique, Berkeley, CA, USA., 2005.

Haywood, J. M., Pelon, J., Formenti, P., Bharmal, N., Brooks, M., Capes, G., Chazette, P., Chou, C., Christopher, S., Coe, H., Cuesta, J., Derimian, Y., Desboeufs, K., Greed, G., Harrison, M., Heese, B., Highwood, E. J., Johnson, B., Mallet, M., Marticorena, B., Marsham, J., Milton, S., Myhre, G., Osborne, S. R., Parker, D. J., Rajot, J.-L., Schulz, M., Slingo, A., Tanré, D. and Tulet, P.: Overview of the Dust and Biomass-burning Experiment and African Monsoon Multidisciplinary Analysis Special Observing Period-0, J. Geophys. Res. Atmospheres 1984–2012, 113(D23), doi:10.1029/2008JD010077, 2008.

Hossaini, R., Chipperfield, M. P., Saiz-Lopez, A., Fernandez, R., Monks, S., Feng, W., Brauer, P. and von Glasow, R.: A global model of tropospheric chlorine chemistry: Organic versus inorganic sources and impact on methane oxidation, J. Geophys. Res. Atmospheres, 121(23), 2016JD025756, doi:10.1002/2016JD025756, 2016.

Joshi, N., Romanias, M., Riffault, V. and Thévenet, F.: Investigating water adsorption on natural mineral dust particles: A DRIFT and BET theory study, under press, Aeolian Research, 2017.

Journet, E., Balkanski, Y. and Harrison, S. P.: A new data set of soil mineralogy for dust-cycle modeling, Atmos Chem Phys, 14(8), 3801–3816, doi:10.5194/acp-14-3801-2014, 2014.

Kaly, F., Marticorena, B., Chatenet, B., Rajot, J. L., Janicot, S., Niang, A., Yahi, H., Thiria, S., Maman, A., Zakou, A., Coulibaly, B. S., Coulibaly, M., Koné, I., Traoré, S., Diallo, A. and Ndiaye, T.: Variability of mineral dust concentrations over West Africa monitored by the Sahelian Dust Transect, Atmospheric Res., 164–165, 226–241, doi:10.1016/j.atmosres.2015.05.011, 2015.

Kandler, K., SchÜTz, L., Deutscher, C., Ebert, M., Hofmann, H., JÄCkel, S., Jaenicke, R., Knippertz, P., Lieke, K., Massling, A., Petzold, A., Schladitz, A., Weinzierl, B., Wiedensohler, A., Zorn, S. and Weinbruch, S.: Size distribution, mass concentration, chemical and mineralogical composition and derived optical parameters of the boundary layer aerosol at Tinfou, Morocco, during SAMUM 2006, Tellus B, 61(1), 32–50, doi:10.1111/j.1600-0889.2008.00385.x, 2009.

Lafon, S., Rajot, J.-L., Alfaro, S. C. and Gaudichet, A.: Quantification of iron oxides in desert aerosol, Atmos. Environ., 38(8), 1211–1218, doi:10.1016/j.atmosenv.2003.11.006, 2004.

Lafon, S., Sokolik, I. N., Rajot, J. L., Caquineau, S. and Gaudichet, A.: Characterization of iron oxides in mineral dust aerosols: Implications for light absorption, J. Geophys. Res. Atmospheres, 111(D21), D21207, doi:10.1029/2005JD007016, 2006.

Linke, C., Möhler, O., Veres, A., Mohácsi, Á., Bozóki, Z., Szabó, G. and Schnaiter, M.: Optical properties and mineralogical composition of different Saharan mineral dust samples: a laboratory study, Atmos Chem Phys, 6(11), 3315–3323, doi:10.5194/acp-6-3315-2006, 2006.

Moreno, T., Querol, X., Castillo, S., Alastuey, A., Cuevas, E., Herrmann, L., Mounkaila, M., Elvira, J. and Gibbons, W.: Geochemical variations in aeolian mineral particles from the Sahara–Sahel Dust Corridor, Chemosphere, 65(2), 261–270, doi:10.1016/j.chemosphere.2006.02.052, 2006.

Mortier, A., Goloub, P., Derimian, Y., Tanré, D., Podvin, T., Blarel, L., Deroo, C., Marticorena, B., Diallo, A. and Ndiaye, T.: Climatology of aerosol properties and clear-sky shortwave radiative effects using Lidar and Sun photometer observations in the Dakar site, J. Geophys. Res. Atmospheres, 121(11), 2015JD024588, doi:10.1002/2015JD024588, 2016.

Müller, T., Schladitz, A., Massling, A., Kaaden, N., Kandler, K. and Wiedensohler, A.: Spectral absorption coefficients and imaginary parts of refractive indices of Saharan dust during SAMUM-1, Tellus B, 61(1), 79–95, doi:10.1111/j.1600-0889.2008.00399.x, 2009.

Petzold, A., Rasp, K., Weinzierl, B., Esselborn, M., Hamburger, T., Dörnbrack, A., Kandler, K., Schütz, L., Knippertz, P., Fiebig, M. and Virkkula, A.: Saharan dust absorption and refractive index from aircraft-based observations during SAMUM 2006, Tellus B, 61(1), 118–130, doi:10.1111/j.1600-0889.2008.00383.x, 2009.

Schladitz, A., Müller, T., Kaaden, N., Massling, A., Kandler, K., Ebert, M., Weinbruch, S., Deutscher, C. and Wiedensohler, A.: In situ measurements of optical properties at Tinfou (Morocco) during the Saharan Mineral Dust Experiment SAMUM 2006, Tellus B, 61(1), 64–78, doi:10.1111/j.1600-0889.2008.00397.x, 2009.

Slingo, A., Bharmal, N. A., Robinson, G. J., Settle, J. J., Allan, R. P., White, H. E., Lamb, P. J., Lélé, M. I., Turner, D. D., McFarlane, S., Kassianov, E., Barnard, J., Flynn, C. and Miller, M.: Overview of observations from the RADAGAST experiment in Niamey, Niger: Meteorology and thermodynamic variables, J. Geophys. Res. Atmospheres, 113(D13), D00E01, doi:10.1029/2008JD009909, 2008.

Val, S., Liousse, C., Doumbia, E. H. T., Galy-Lacaux, C., Cachier, H., Marchand, N., Badel, A., Gardrat, E., Sylvestre, A. and Baeza-Squiban, A.: Physico-chemical characterization of African urban aerosols (Bamako in Mali and Dakar in Senegal) and their toxic effects in human bronchial epithelial cells: description of a worrying situation, Part Fibre Toxicol, 10(10), 2013.

Ververis, C., Georghiou, K., Christodoulakis, N., Santas, P. and Santas, R.: Fiber dimensions, lignin and cellulose content of various plant materials and their suitability for paper production, Ind. Crops Prod., 19(3), 245–254, doi:10.1016/j.indcrop.2003.10.006, 2004.

Wankhede, D. B., Shehnaj, A. and Rao, M. R. R.: Carbohydrate composition of finger millet (Eleusine coracana) and foxtail millet (Setaria italica), Qual. Plant., 28(4), 293–303, doi:10.1007/BF01095511, 1979.

Wienecke, J., Kruse, H. and Wassermann, O.: Organic compounds in the waste gasification and combustion process, Chemosphere, 25(4), 437–447, doi:10.1016/0045-6535(92)90277-X, 1992.

Zotter, P., Herich, H., Gysel, M., El-Haddad, I., Zhang, Y., Močnik, G., Hüglin, C., Baltensperger, U., Szidat, S. and Prévôt, A. S. H.: Evaluation of the absorption Ångström exponents for traffic and wood burning in the Aethalometer-based source apportionment using radiocarbon measurements of ambient aerosol, Atmos Chem Phys, 17(6), 4229–4249, doi:10.5194/acp-17-4229-2017, 2017.

Journal: ACP
Title: Chemical characterization and source apportionment of submicron aerosols measured in Senegal during the 2015 SHADOW campaign
Author(s): Laura-Hélèna Rivellini et al.
MS No.: acp-2016-1127

The authors want to thank Reviewer #3 for his/her helpful comments. They are addressed below in blue. Changes in the manuscript are written in red.

**Anonymous Referee #3**

Overview:

This manuscript presented the highly-time resolved chemical characterization and source apportionment of atmospheric submicron aerosol particles (PM1) in West Africa, along with field on-line measurements (including an ACSM and a 7-wavelength aethalometer) and offline model analysis. The campaign was deployed under the environment affected by anthropogenic emissions (e.g., traffic, cooking, and biomass burning) and natural sources (e.g., desert dust and marine air masses), etc.. The results showed that the ten times lower average concentrations of NR-PM1 were observed here compared to the results from other megacities with persistent air pollution issues, i.e., Beijing and Paris. Sea breeze phenomena and Saharan desert dust outbreaks may lead to pollution events with high concentrations of PM10 (up to 900 µg/m3). Organic matter (OM) and sulfate could dominate the major fraction of aerosol particles when air massed could be associated to different influences, i.e., continental and sea breeze and oceanic region related. The authors also estimated the mass concentrations of particulate Fe from the Aethalometer data, for which an average contribution (4.6%) of Fe to PM1 was obtained. A new organic aerosol factor (LCOA), Local Combustion Organic Aerosol, was resolved by PMF analysis, with relatively low contribution (3%). Both regional and local photochemistry processes could contribute the formation of oxygenated organic aerosols in this area. The results seem to be interesting. The manuscript is well written and organized. I would recommend this paper could consider to be published in ACP once the following comments are addressed.

Specific comments:

1. Page 4, line 30: Should keep the same abbreviations for those species throughout the manuscript. For example, what's the different between "NO3-" and "NO3" (Page 5 and line 5), and somewhere else "SO42-" and "SO4", "Cl-", "Cl" and "Chl", etc.. If they are different for the discussion in this manuscript, please the authors give the related text to explain them.

**Author's response**: As suggested we have harmonized the nomenclature regarding the sum of nitrate ($NO_x$, $ONO_x$…), ammonium ($NH_x$), sulfate ($SO_x$, $H_ySO_x$) and chloride ($Cl$, $HCl$) related fragments. The $SO_4^{2-}$, $NO_3^-$ and $Cl^-$ ions are only used in the neutralization equation.

**Changes in the manuscript:**

Page 4 lines 29-31: "Non-refractory species, such as organic matter (OM), sulfate ($SO_4$), nitrate ($NO_3$), ammonium ($NH_4$) and non-refractory chloride (Chl), are vaporized at this temperature and then ionized by electron impact (70 eV). The abovementioned names of the different NR species correspond to the sum of all m/z fragments related to one given species in the fragmentation table (Allan et al., 2004), that is to say $H_{0\leq x\leq 2}S_{0\leq y\leq 1}O_{0\leq z\leq 4}$ for sulfate, $NH_{0\leq x\leq 2}$ for ammonium, $NO_{0\leq x\leq 2}$ and $HNO_3$ for nitrate, and $H_{0\leq x\leq 1}Cl$ for chloride."

2. Page 5, line 3: It was interesting to perform the chloride calibration with ammonium chloride particles. Could the authors also present the related calibration results in supporting information, as showing in Fig. S1, since your RIEChl is much higher than the default value (1.3). In addition, did the authors try to validate chloride data based on your calibration results? It's also interesting to know how it works about the chloride calibration.

**Author's response**: Chloride RIE calibrations (relative to nitrate) were performed using the same methodology as for ammonium nitrate and sulfate calibrations. A monodisperse aerosol at 300 nm was generated from an aqueous solution of $NH_4Cl$ (> 99.8%, Merck) at $5 \times 10^{-3}$ mol $L^{-1}$. The fragments at m/z 15, 16 and 17 were taken into account to determine the concentration of $NH_4$ whereas those at 35, 36, 37 and 38 were used for Chl.

Chl concentrations (in $\mu g\ m^{-3}$) were then calculated using the following equation:

$$Chl = N_{CPC} \times S \times V_{part} \times \rho \times \frac{M(Cl^-)}{M(NH_4Cl)}$$

where $N_{CPC}$ is the number concentration given by the CPC in particles per $cm^3$, S the shape factor (taken as 1), $V_{part}$ the particle volume (in $cm^3$) corresponding to 300 nm and assuming spherical shape, $\rho$ the ammonium chloride density (1.53 g $cm^{-3}$), M(X) the molar mass in g $mol^{-1}$.

Using the ratio of the slopes obtained by plotting the sum of Chl signals vs Cl mass, and the similar plot for $NH_4$, combined with RIE($NH_4$), allows to retrieve RIE(Chl) through the following equation:

$$RIE(Chl) = RIE(Chl/NH_4) \times RIE(NH_4/NO_3)$$

with RIE(Chl/ $NH_4$) and RIE($NH_4/NO_3$) mean values of $0.39 \pm 0.04$ and $5.72 \pm 0.55$, respectively, we obtained RIE(Chl) = $2.26 \pm 0.02$.

Comparing this value with external chloride calibrations is tricky since only non-refractory chloride can be detected by AMS/ACSM techniques, whereas most chloride ambient analyses (using ion chromatography for instance) would be dominated by (refractory) sea-salt chloride. Furthermore, NR-Chl tend to be rather low at most sites and therefore has not been a major concern in the AMS/ACSM community so far.

**Changes in the manuscript:** An example of chloride calibration with $NH_4Cl$ has been added in the supplementary (Figure S1e). Page 5 line 7 now reads: "S1(a-e)".

3. Page 7, line 32: Would it be possible that the authors could give uncertainties of estimated Fe concentrations with this method for your study?

**Author's response**: Indeed the uncertainties on estimated Fe concentrations can be calculated by applying the propagation for uncertainties on the values of $K_{Fe}$ (10%) and the slope b (39%, calculated using a variability of 0.2 for β and α (Fialho et al., 2006)), which gives an overall uncertainty of ~40%. However this method is highly sensitive to even small variations of α (BC) and β (DD), with values quite well known for BC from fossil fuel ranging from 0.8 to 1.1 (Hansen, 2005; Zotter et al., 2017 and references therein) but not so much for dust. In the manuscript, we chose to use β = - 4, according to Fialho et al. (2006) values determined at the Azores Islands for samples influenced by Saharan dust events. But other values can be found in the literature (Table R1), ranging from -1.6 to -6.5 and largely influenced by the wavelength range as well as dust origins and size fractions since the iron content differ depending on emission sources and particle size (Journet et al., 2014). Even during the SAMUM campaign (May to June 2006 in Morocco), a wide range of AAE values have been reported from 1.6 up to 5.1 for ground-based measurements in the same size fraction, as shown in Table R1.

Table R1. Mineral dust AAE values reported from field campaigns around the Saharan region.

| Reference | Location / Period | Wavelengths (nm) | Fraction | β |
|---|---|---|---|---|
| Fialho et al. (2006) [a] | Azores Islands Jul. 2001 – Jun. 2005 | 370-950 | - | -4 |
| Müller et al. (2009) [a] | Tinfou, Morocco (SAMUM) Summer 2006 | 467/660 | $PM_{10}$ | -2.25 to -5.13 |
| Petzold et al. (2009) [b] | South-East Morocco (SAMUM) Summer 2006 | 467/660 | $PM_{2.5}$ | -2 to -6.5 |
| Schladitz et al. (2009) [a] | Tinfou, Morocco (SAMUM) Summer 2006 | 537/637 | $PM_{10}$ | -1.6 to -4.73 |
| (Linke et al., 2006) [c] | Morocco Egypt | 266/532 | $\sim PM_4$ | -4.2 -5.3 |
| (Caponi et al., 2017) [c] | Morocco Lybia Algeria Mali | 375-850 375-532 375-850 375-532 | $PM_{2.5}$ ($PM_{10.6}$) | -2.6 -4.1 (-3.2) -2.8 (-2.5) -3.4 |

[a] In situ ground-based measurements; [b] Airborne measurements through dust plumes; [c] Laboratory experiments with resuspended soil samples

Therefore applying a relatively small increase (resp. decrease) of 10% on the value of β for our dataset led to a 33% decrease (resp. 50% increase) of iron concentrations, as shown in Figure R1, but no change in the temporal behavior.

[Figure]

Figure R1. Scatter plot of iron concentrations (in µg m$^{-3}$) obtained from Fialho's deconvolution method using an AAE value of ± 10% compared to the one from the literature and used in the manuscript.

**Changes in the manuscript:**
A new appendix (S2) in the Supplementary Information now includes this whole discussion. Changes in the main text have been also done page 8, line 18 with a new sentence added: "Applying the propagation for uncertainties approach on the values of $K_{Fe}$ (10%) and the slope b (39%, calculated using a variability of 0.2 for α and β (Fialho et al., 2006)) gives an overall uncertainty of ~40% for iron concentrations. However the deconvolution algorithm is highly sensitive to the values of the Angström absorption exponents (α and β) and a more detailed discussion can be found in Appendix S2."

4. Page 8, line 19: In this paragraph, I suggest that the authors could consider to also mention some brief information about ME-2 algorithm how it works for constraining organic aerosol factors (Canonaco et al., 2013), since this will be helpful and easier for readers to quickly understand the SoFi how it works in this study.

**Changes in the manuscript:** Page 9 lines 4, sentence added at the end of the paragraph: "In case of mixed (known) factors, the solution can be furthermore constrained by imposing reference factor profiles (F, from the literature) as inputs. The user can apply those constraints with a certain degree of freedom defined by a scalar a-value ranging from 0 (no degree of freedom) up to 1 (totally unconstrained)."

5. Page 10, line 20: The authors should consider to explain which kind of combustion sources for the ACSM m/z 57 tracer is. When I am reading here, I immediately realized that why the authors did not perform the source apportionment of black carbon by using the Aethalometer model (Sandradewi et al., 2008). As described in Section 2.1, there are both traffic and biomass-burning emissions that would potentially contribute the ambient black carbon burden at this sampling site. So, is it possible to identify blackcarbon aerosols related to traffic and biomass-burning emissions here? This will be very helpful for the source apportionment of organic aerosol.

**Author's response**: m/z 57 is mostly the $C_4H_9^+$ fragment, which has been linked to combustion sources, and is one of the key fragments in HOA and COA spectra but appears as well in the BBOA one without being the main tracer (Ng et al., 2011a).
As to the second part of the comment, although biomass burning (BB) events can sometimes be observed in the region, no BB aerosols were detected during IOP-1 as mentioned in section 2.3.2. That was in fact the first condition required to apply the deconvolution method from Fialho et al. between BC from fossil fuel and Fe from mineral dust. Using this method does assume that no brown carbon (BrC) is present since the Angström absorption exponent (AAE) for BC is taken as the one from fossil fuel (AAE ~ 1).
On the other hand, the method proposed by Sandradewi et al. (2008) assumes that absorbing particles are only BC from fossil fuel or wood burning sources, and that there is no significant absorption from dust, which we know to be untrue at the M'Bour site. Therefore, as both dust and BC from wood burning absorb in the shortest wavelengths, the two models cannot be applied at the same time unless we could have constrained the time profile of each source with external tracers. In the absence of external data, a three-factor deconvolution has been tried but would still be highly hypothetical and therefore is not presented here.
Anyway, during the campaign, mineral dust was clearly present, sometimes at high concentrations, whereas sources of BrC were almost unsignificant. Therefore we chose to exclude the very few periods when BrC could be suspected to influence our measurements (1% of the data) and therefore the empirical deconvolution using Fialho's algorithm.

**Changes in the manuscript:**
Page 10 lines 19-20, sentence modified: "The higher concentrations can be attributed to local anthropogenic combustion processes as BC concentrations present a significant correlation (r = 0.79) with the ACSM m/z 57 tracer of all types of combustion."

Page 7 line 36 – page 8 line 1, sentence now reads: "As mentioned by Fialho et al. (2014), this method allows to estimate elemental iron concentrations only in the absence of brown carbon since an absorption Angström exponent of 1 (which correspond to fossil fuel BC) is applied. Therefore, other methods, such as the one proposed by Sandradewi et al. (2008) to deconvolve BC from fossil fuel and biomass burning, cannot be used in our conditions."

6. Page 11, line 3: Here is a little bit confusion about the ratio of Fe/PM10, since your Fe concentrations were estimated by the PM1 aethalometer, but not PM10 Fe. Is this the case? If yes, the

authors could consider to add "PM1" in front of "Fe" when you discuss this ratio in the main text of the manuscript.

**Author's response**: Indeed Fe was deconvolved from $PM_1$ absorption measurements and we agree that using the $Fe/PM_{10}$ ratio was confusing. We have now removed these direct comparisons and focused on the only available study for which we could derive $Fe/DD_{PM1}$ in Dakar, Senegal, although in the absence of significant dust events. We only use the Fe/DD ratios found in the literature to emphasize the influence of the size fraction on the iron contribution to DD since most of it can be found in the clay fraction ($\sim PM_{2.5}$; Journet et al., 2014; Kandler et al., 2009), as part of Appendix S2.

**Changes in the manuscript:**
Page 11 lines 3-13: "From the only study in the literature focusing on iron concentrations in the submicron fraction in West Africa (Val et al., 2013), we could infer an elemental iron contribution of 7.8% to $PM_1$ dust, in Dakar, in the absence of dust events. Other studies focused on dust gave the iron contribution for size fractions higher than $PM_1$, thus no straightforward comparisons can be made with our average ratios of $Fe/DD_{PM1}$ (20, 23, 21 and 16% for respectively IOP-1, continental, sea breeze and marine days). It can nevertheless be interesting to have in mind values retrieved within the same region as it is known that iron oxides mainly belong to the finest fraction (Journet et al., 2014; Kandler et al., 2009) and therefore the elemental iron contribution should be lower for larger sizes, which is consistent with values reported in Table S2.2."

**Table S2.2: Comparison of iron content (in %) determined in Saharan dust and soil samples**

| Reference | Location | Method [a] | Size fraction | %Fe [b] |
|---|---|---|---|---|
| Dust samples | | | | |
| (Lafon et al., 2004) | Banizoumbou (Niger) | XRF; CBD | TSP | 6.3; 7.8 |
| (Lafon et al., 2006) | Banizoumbou, | XRF; CBD | TSP | 4.3 – 6.1 |
| (Lafon et al., 2006) | Cape Verde | XRF; CBD | TSP | 5.3 – 6.0 |
| (Formenti et al., 2008) | Banizoumbou | CBD | 40 μm | 5.8 |
| (Val et al., 2013) | Dakar (Senegal) | ICP-MS | 1 μm | 7.8 |
| This work | M'Bour | cf. text | 1 μm | 23 (continental) 21 (sea breeze) 16 (marine) |
| Soil samples | | | | |
| (Moreno et al., 2006) | Saharan region (9 samples) | ICP-AES/ ICP-MS | TSP | 2.0 – 4.7 |
| (Lafon et al., 2006) | Banizoumbou, | XRF; CBD | 10.2 μm [*] 2.5 μm [*] | 5.3 5.8 |
| (Joshi et al., 2017) | M'Bour, Bordj (Algeria), Nefta (Tunisia) | XRD | 100 μm | < 0.5 |

[a] XRF: X-ray Fluorescence (XRF) Spectrometry for elemental analysis; CBD: chemical method based on citrate-bicarbonate-dithionite (CBD) reagent for quantification of iron oxides adapted from soil analysis (Mehra and Jackson, 1960)
[b] Percentages of iron relative to the mass of all oxides, classically taking into account $Na_2O$, $MgO$, $Al_2O_3$, $SiO_2$, $K_2O$, $CaO$, $TiO_2$ and $Fe_2O_3$.
[*] Soil samples resuspended using wind tunnel and collected with a 13-stage impactor

7. Page 11, lines 27-28: Why didn't the authors consider to give the contribution of chloride to total NR-PM1? I suggest the authors could also mention it.

**Changes in the manuscript:** The contribution of chloride (~1%) was added page 11 line 28.

8. Page 11, lines 32-34: Is there evidence to support this discussion? Otherwise, the authors should give related reference(s).

**Author's response**: We have now lengthened the discussion on that point, plus corrected a typo ($SO_4$ and not $SO_2$).

**Changes in the manuscript:**
Page 11 line 30: "In our case these differences could be explained both by the semi-volatile nature of $NH_4NO_3$ combined with the limited use of fertilizers that prevent $NH_3$ emissions and ammonium nitrate formation, and more sources of non sea salt(nss)-$SO_4$ such as marine DMS oxidation processes. The first point can be assessed by emission inventories that provide annual $NH_3$ emissions (in 2010) of 53 kT in Senegal against 870 kT in France and 204 kT in the Netherlands (source EC-JRC/PBL. EDGAR version 4.2. http://edgar.jrc.ec.europa.eu/, 2011). nss-$SO_4$ comes from secondary origin and has been investigated in $PM_{10}$ at the Cape Verde Atmospheric Observatory (Fomba et al., 2014). This study showed increased concentrations of nss-$SO_4$ during dust events, linked to the oxidation of anthropogenic $SO_2$ transported by continental air masses. They also evidenced a seasonal variability of nss-$SO_4$ for marine air masses, increasing during summer, which was attributed to increased photochemistry and changes in the emission of dimethyl sulfide (DMS) due to higher biological activities in the ocean. This activity can be traced back using satellite data from AQUA/MODIS, in particular the algae concentrations along the Senegalese coast (Ocean biology processing group, 2003)."

9. Page 14, Section 3.2.1: It would be more convinced about the discussion on geographical origins, if the authors would also combine with some modeling methods, e.g., potential source contribution function (PSCF) that might be easily performed on highly-time resolved data (Petit et al., 2017). In addition, it could be also interesting to perform PSCF on organic aerosol factors.

**Author's response**: Indeed we have hopefully improved the discussion on geographical origins by adding a supplementary figure (Figures S5a-c) that presents: (i) back-trajectory clusters for the entire period and the different day categories (continental, sea breeze, marine); (ii) non-parametric wind regression (NWR) plots, which combine atmospheric concentrations with measured wind speed and direction for the different species and PMF factors; (iii) probability source contribution function (PSCF) maps for the variables presenting a regional origin according to the NWR plots.

[Figure]

**Figure S5.** (a) 48-hour back trajectory clusters for (from left to right) IOP-1, continental, sea breeze and marine days. (b) NWR plots (input parameters: angular and radial resolution: 0.1, angle smoothing: 2, radial smoothing: 1; upper limit of the color scale: 75[th] percentile) for PMF factors and NR-PM$_1$, BC and Fe species and corresponding (c) PSCF maps for species showing regional influence (threshold: 75[th] percentile) during IOP-1.

**Changes in the manuscript:**
A new sub-section was added in the methodology section to introduce back-trajectory calculations, NWR plots and PSCF maps page 9 line 4:
"2.3.4. Geographical origins of air masses and chemical species

Air masses reaching the site were characterized through 48-hour back-trajectories (every 3 hours) retrieved from the computer version of the Hybrid Single-Particle Lagrangian Integrated Trajectory model (HYSPLIT; Draxler and Hess (1998)), for an altitude set at one half of the mixed layer depth and coupled with the GDAS (1 degree) meteorological database. Note that sea breeze phenomena, which occur at short time and spatial scales, cannot be satisfactorily reproduced by this type of model. However given the dynamics of sea breezes, only back-trajectories arriving on site at 3 pm and eventually at 6 pm during sea breeze days (< 8.2%) could not be representative of the ground dynamic observations. Therefore all the back-trajectories available for IOP-1 were kept and could be statistically grouped into clusters according to the variation of the total spatial variance, for the whole period and also by day type.

We also used pollution roses to identify local wind directions leading to high concentrations for each species or PMF factors, but also two additional tools provided by the ZeFir Igor-based package developed by Petit et al. (2017):
(i) Non-parametric regression (NWR, Henry et al. (2009)) plots, which combine smoothed surface concentrations and local wind speed and direction, to discriminate between local and more distant/regional sources;
(ii) Potential Source Contribution Function (PSCF, (Polissar et al., 2001)) maps for regional sources, which couple time series of one variable with air mass back-trajectories to redistribute the concentrations observed at the site into geographical emission parcels."

Page 14 line 28: "From back-trajectory analysis (Figure S5a) three different clusters were encountered during the whole period. The prevailing one (77% for the whole IOP-1; 91, 80 and 43% for continental, sea breeze and marine days, respectively) evidences air masses transported along the Western African coast and over Dakar. A second cluster corresponds to air masses purely originating from the ocean (19% of the total back-trajectories) and appeared as two clusters during marine days. A last cluster coming from the Saharan desert contributes only 4% of the IOP-1 air masses but reaches 9 and 10% for the continental and sea breeze days, respectively."

Page 14 line 31: "NWR plots and PSCF maps can be found in Figure S5b and S5c, respectively."

Page 14 line 34: "BC and $NO_3$ both exhibit local and regional influences, as suggested by their NWR plots (Figure S5b). The corresponding PSCF maps (Figure S5c) indicate regional background concentrations could come from anthropogenic emissions from Dakar (~1 million inhab. within city limits and ~3 in the metropolitan area) and possibly from maritime traffic along the Western African coast."

Page 15 line 5: "These species may also be released by anthropogenic activities in distant cities like Dakar, whose emissions may be carried toward the ocean and brought back to M'Bour by western winds. This hypothesis is also supported by back-trajectory analysis (Figure S5a)."

Page 15 line 10: "Regarding the iron pollution rose and NWR plots reported in Figure 8 and Figure S5b, maxima are measured when the site is under the influence of NE winds. The NWR plot evidences both local emissions possibly linked to traffic resuspension of DD and a regional component, that the Fe PSCF map clearly attributes to the Saharan region."

Page 16 line 34: "The HOA rose plot shows marked peaks in the directions of the two open waste burning areas and of the fish-smoking area located northeast of the site in the outskirts of M'Bour. HOA and Chl NWR plots are very similar, which suggests either common sources or a mixture of both compounds in the air masses which resulted into a correlation of 0.64 between these two variables."

Page 17 line 29: "Besides the NWR plots of Chl (local influence) and LCOA (both local and regional) rather suggest the presence of chlorinated organics. The PSCF maps identify two possible origins, one clearly from the ocean that could be related to chlorine-driven photo-oxidation processes (Hossaini et

al., 2016) and the other linked to air masses carried over Dakar where similar massive anthropogenic emissions from waste burning could be expected from Mbeubeuss, the largest dumpsite in Senegal located 25 km north-east of Dakar along the coast, which receives 250,000 tons of garbage per year from the Dakar region (Cissé, 2012)."

Page 17 line 21: "whereas LCOA pollution rose and NWR plots clearly point out toward the local combustion areas already mentioned previously"

Page 18 line 11: "OOA might not be only emitted by long distant sources, as also suggested by its NWR plot (Figure S5b)"

Page 18 line 14: "(Figures 8 and S5b)"

Page 18 line 23: "As shown by the PSCF map (Figure S5c), higher OOA concentrations are associated to air masses that moved along the coast and could transport oxidized anthropogenic species to the receptor site."

10. Page 15, lines 24-25: Should "CnH2n-1" and "CnH2n+1" be "CnH2n-1+" and "CnH2n+1+", respectively?

**Changes in the manuscript:** The "+" charges have been added page 15 lines 24-25.

11. Page 15, line 13: Change "Source apportionment" to "Source apportionment of OM".

**Changes in the manuscript:** Page 15 line 13: the title has been changed.

12. Page 15, Section 3.2.2: The biggest question I have in this manuscript is about the PMF-OA solution.
1) Page 15, lines 15-17: Please the authors could also present the results of 3 – 10 factors from the PMF-free runs in supporting information.

**Changes in the manuscript:** The results of 3-10 factors from the PMF unconstrained runs are now available in Supporting Information (Figure S6.1).

A sentence has been added page 15 line 21: "This factor appeared constantly above 4 factors in the unconstrained runs from 3 to 10 factors (Appendix S6, Figure S6.1) and was associated with one of the OOA for the continental, sea breeze and marine 4-factor unconstrained solutions (Figure S6.2)".

**(a) 3-factor solution**

[Figure]

[Figure]

**(b) 4-factor solution**

[Figure]

[Figure]

**(c) 5-factor solution**

[Figure]

[Figure]

(d) 6-factor solution

(e) 7-factor solution

(f) 8-factor solution

(g) 9-factor solution

[Figure]

[Figure]

(h) 10-factor solution

[Figure]

**Figure S6.1.** PMF unconstrained solutions from 3 to 10 factors, with (left) factor profiles and (right) corresponding daily cycles.

2) About the LCOA factor related to local open waste-burning emissions, did the authors try to compare it with any tracers? For example, please try to do some correlation analysis between LCOA with f60 (an AMS/ACSM biomass-burning tracer), f36 (as mentioned by the authors), and black-carbon aerosols from different sources, etc..

**Author's response**:
LCOA was compared with some tracers, as mentioned in section 3.2.2 (page 17 lines 14-16), namely Chl (r = 0.44), m/z 36 (r = 0.55) and 58 (r = 0.84).
LCOA correlation with m/z 60 remains moderate (r = 0.52), and when considering $f_{36}$ (= mz36 / OA) and $f_{60}$ (= mz60 / OA), correlations drop down to 0.16 and 0.25, respectively. No correlation is observed with BC (r = -0.02). We have also compared LCOA with m/z 39 (r = 0.29), which is commonly attributed to potassium ion $K^+$ but can be emitted by various sources including biomass burning, industrial processes and waste incineration (Olmez et al., 1988; Riffault et al., 2015; Simoneit, 2002; Vassilev et al., 2010).
The following figure shows scatter plots related to the above-mentioned correlations between LCOA and tracers.

[Figure]

**Figure R2.** Scatter plots of LCOA vs. m/z 36, 60, BC (units are in μg m$^{-3}$) and m/z 39 (unit Amps).

At the same time, the authors did not find any reasonable BBOA profile instead of LCOA factor. Did the authors try to compare the LCOA profile with any BBOA related profiles? What's different from them?

**Author's response**: Figure R3 compares our LCOA profile from the 4-factor constrained solution with two BBOA profiles from the literature (Crippa et al., 2013; Ng et al., 2011b). The LCOA factor remains atypical with higher signals at m/z 37, 56, 60, 83, 91; and lower signals at m/z 15, 27, 41, 43, 55.

[Figure]

**Figure R3.** Scatter plots of LCOA m/z profile from the 4-factor constrained solution vs. two BBOA literature profiles, with marker symbols as m/z numbers.

And did the authors find any similar such mass spectrum of LCOA during your unconstrained runs and/or constrained HOA and COA runs, respectively? I am thinking whether the LCOA factor is a kind of splitted factors due to constrained HOA and COA? Could the authors give those results from your 3-10 factor runs and make comparison with your LCOA profiles.

**Author's response**: The 3-10 factor unconstrained solutions are now provided in Appendix S6 Figure S6.1 (cf. reply to comment 12). The LCOA factor consistently appears without constraints as a stable factor (F2) from 4 factors and more as shown in Figure S6.1.

Based on the results of Fig. S6, the constrained five-factor solution seems also good. Why didn't the authors choice this one for the finally PMF-factors solution?

**Author's response**: First, it should be emphasized that both solutions (the 4-factor mildly constrained one and the 5-factor strongly constrained one) lead to physically realistic modelling of the observations. As for any statistical source receptor model, there is no unique solution and therefore it is the responsibility of the user (based on one's modelling expertise and knowledge of the field site) that generally guides toward one solution over the other. This feature can indeed be a weakness since the user can drive the solution toward his/her preconceived expectations. Therefore the constraints should be ideally as low as possible and always consistent with other available information.

In order for our PMF solution to be comparable to other studies in the literature, we chose to present the 4 factor solution in the main text, for which more conventional model input data (that is to say only the organics matrix) were used with (i) profiles from the literature and (ii) mild constraints on the factor profiles. The 5-factor solution was obtained by using m/z 36 ($HCl^+$) as an additional input for the PMF m/z matrix, and strong constraints on all the primary OA profiles (HOA, COA and LCOA all taken from unconstrained runs and not from the literature). It nevertheless gives an interesting insight into the use of non-refractory Chloride species or fragments as possible tracers of specific activities not observed so far in previous studies, as well as an estimation of the attribution of a more oxidized (and therefore more regional) OOA (MO-OOA), and one that is rather locally formed due to enhanced photochemical processes (LO-OOA) and that is why we decided to present it in the Supplementary Information as well.

Did the authors also try to constrain all BBOA, COA, and HOA factors together to check the five-factor solution? And the authors could also try to just constrain BBOA and HOA factors to check the results. There is in factor that the BBOA factor/profile could be affected when HOA and COA factors were constrained together.

**Author's response**: The BBOA factor was constrained for different a-values, either alone or in combination with other primary factors (HOA/COA/LCOA) as mentioned in section 3.2.2 (page 15 line 36 to page 16 line 16) but no realistic profile or solution could be found.

Page 8, lines 1-5, we also reported the average $f_{60}$ signal (0.3%) highlighting the absence of significant BBOA at the site.

3) In addition, could the authors explain why the median mass concentrations of COA and HOA show relatively similar diurnal variations (Fig. 9), and the similarly high peaks of HOA, COA, and LCOA around 8 am? I suggest that the authors could also make correlation analysis among HOA, COA, and LCOA each other in supporting information. In addition, the authors should also consider to perform the source apportionment of black carbon as mentioned above. It will be very useful to evaluate the PMF solution by checking the relationships between HOA, COA, and LCOA with black carbon from fossil fuel combustion and biomass burning, respectively.

**Author's response**: Weak correlations were obtained between HOA vs. LCOA and COA vs. LCOA with respectively r =0.20 and 0.38 while HOA vs. COA present a higher correlation (r=0.77). Indeed the morning wind conditions were really stable from day-to-day for continental and sea breeze days so air masses arriving at the site around 8 am were likely loaded with the three types of POA from the different combustion sources encountered in the NE direction. We have added some pictures of the three different sites identified as possible sources for the LCOA factor in Figure 1.

It can be noted also that the HOA and COA factors were hard to deconvolve because of this repetitive temporal dynamic pattern on continental days (as has already been shown for other datasets, e.g. Fröhlich et al., 2015) but they appeared without any constraints when running the PMF on sea breeze days (see Figure S6.2c, formerly Fig S4.1c) and were further used as anchors for the constrained PMF solutions. HOA was compared with BC (estimated using α = - 1 so mainly resulting from fossil fuel combustion) but we are not able to compare our factors with $BC_{bb}$ (not retrievable using Fialho's

deconvolution, see reply to comment 5) and chose instead to use other tracers like m/z 36 or 58 for LCOA.

**Changes to the manuscript:** Pictures of three sites identified as contributing to the LCOA factor (Gandigal and Saly Douté open waste burning areas and a suburban fish-smoking site in M'Bour) were added to Figure 1.

Page 15 line 33: "(Figure 1 and Appendix S6, Figure S6.3)"

Page 18 line 6: "The hot temperatures and intense solar irradiation encountered in the region enhance these processes and can explain the major contribution (45%) observed for the OOA factor during IOP-1, and the predominance (~3/4) of the more-oxidized fraction in the solution presented in Appendix S8."

[Figure]

**Figure1.** (top left) Dakar and M'Bour locations with city delimitations in orange and (top right) local sources located around the IRD sampling site (red star), with open waste burning areas (green circles), fish-smoking sites (blue triangles) and the M'Bour port (light blue diamond). (bottom) Photographs of (from left to right) smoldering fire in the Gandigal open waste burning area; flaming fire in the Saly Douté open waste burning area; fish-smoking location (drying stage) in the suburb of M'Bour.

4) Why did the authors select a-value = 0.6 for both HOA and COA factors (Fig. 9)? I suggest that the authors could also perform the sensitivity test of a-values (e.g., from 0 to 1, with delta a = 0.1/0.05) on HOA and COA factors for your data sets. And why did the authors apply the reference mass spectrum of COA from HR-AMS (Crippa et al., 2013) for ME-2 constraining runs, but not from the ACSMs (Fröhlich et al., 2015;Ng et al., 2011)?

**Author's response**: We applied different combinations of a-values based on the approach described in Elser et al. (2016); the value of 0.6 was the one offering the best solution without using too strong constraints (in order not to drive the solution to what we are expecting). We also used this approach when constraining 1, 2, 3 or 4 primary factors at a time (LCOA, HOA, COA and BBOA). There were not many differences between both the Ng et al. (2011b) and Crippa et al. (2013) COA profiles especially as we used a-values between 0.3 and 0.9. It must be noted that in the intercomparison study of Fröhlich et al. (2015) mentioned by referee #3, the COA factor could not be deconvolved by Q-ACSMs without constraints and therefore the Crippa et al. (2013) factor obtained in a previous field campaign in Paris was used as the reference profile as well.

Please check "Ng et al., 2016" in the plot?

**Changes in the manuscript:** Fig 9 corrected from "Ng et al., 2016" to "Ng et al., 2011".

5) Page 18, lines 3-23: It would be also interesting to discuss the different types of OOA factor, i.e., LO-OOA and MO-OOA, as showing in Fig. S6. Why didn't the authors keep both them for OOA factors in the final PMF solution? The authors would consider to try to take a look at the relationship between OOA and Fe concentrations. This might make sense to find something new.

**Author's response**: In the 5-factor constrained solution presented in Appendix S6 of the submitted manuscript (now Appendix S8), two different kinds of OOA were obtained: one more oxidized (MO-OOA; 76.5% of OOA) and considered from a more regional origin (mostly marine as highlighted by its NWR plot and PSCF map in Figure R4 below) and the other less oxidized (LO-OOA; 23.5% of OOA), locally emitted as per its NWR plot. Nonetheless, without using m/z 36 as input and literature profiles for constraining HOA and COA none of the solution leads to two completely distinct OOA profiles. When comparing both LO-OOA and MO-OOA with Fe and BC, only low correlations (r < 0.4) were found between the different variables, except for LO-OOA and BC (r = 0.64; n=3854) which might underline common sources for these species. Iron and OOA have completely different origins as also shown in the NWR plots and PSCF maps now presented in Figure S5 (see comment 9). If considering MO-OOA only, most of it could be rather due to the oxidation of ship emissions along the Western African coast, which would also explain the better correlation observed with $NO_3$ from $NO_x$ emission processing despite the predominance of this regional oxidized factor over the local one.

[Figure]

[Figure]

**Figure R4.** NWR plots (top) and PSCF map (bottom) for MO-OOA (left) and LO-OOA (right) obtained with the 5-factor constrained solution including m/z 36.

**Changes in the manuscript:**

Abstract, page 2 lines 4-6: "The remaining fraction was identified as oxygenated organic aerosols (OOA), a factor that prevailed regardless of the day type (45%) and was representative of regional (~3/4) but also local (~1/4) sources due to enhanced photochemical processes."

Page 15 lines 34-36: "Since the behavior of Chl had also been suspected to come from the same sources, PMF solutions adding the m/z 36 signal in the input matrix were investigated, and a solution is presented in Appendix S8, where regional OOA accounts for ~3/4 of the OOA and local OOA ~1/4."

Page 18 line 21: "The OOA PSCF map (Figure S5c) seems to trace back its origin along the entire Western African coast, where shipping emissions could be a major source of organic aerosols."

· · · · · · · · · · · · · · · · · · · · · · · · · · · · · · · · · · · · · · · · · · · · · · · · · · · · · · ·

**References cited in this reply**

Allan, J. D., Bower, K. N., Coe, H., Boudries, H., Jayne, J. T., Caragaratna, M. R., Millet, D. B., Goldstein, A. H., Quinn, P. K., Weber, R. J. C. G. L. and Worsnop, D. R.: Submicron aerosol composition at Trinidad Head, California, during ITCT 2K2: Its relationship with gas phase volatile organic carbon and assessment of instrument performance, J. Aerosol Sci., 35, 909–922, doi:10.1016/j.jaerosci.2004.02.007, 2004.

Caponi, L., Formenti, P., Massabó, D., Di Biagio, C., Cazaunau, M., Pangui, E., Chevaillier, S., Landrot, G., Andreae, M. O., Kandler, K., Piketh, S., Saeed, T., Seibert, D., Williams, E., Balkanski, Y., Prati, P. and Doussin, J.-F.: Spectral- and size-resolved mass absorption efficiency of mineral dust aerosols in the shortwave: a simulation chamber study, Atmos Chem Phys Discuss, 2017, 1–39, doi:10.5194/acp-2017-5, 2017.

Cissé, O.: Les décharges d'ordures en Afrique - Mbeubeuss à Dakar au Sénégal, Karthala., 2012.

Crippa, M., DeCarlo, P., Slowik, J., Mohr, C., Heringa, M., Chirico, R., Poulain, L., Freutel, F., Sciare, J. and Cozic, J.: Wintertime aerosol chemical composition and source apportionment of the organic fraction in the metropolitan area of Paris, Atmospheric Chem. Phys., 13(2), 961–981, 2013.

Draxler, R. R. and Hess, G. D.: An overview of the HYSPLIT_4 modeling system of trajectories, dispersion, and deposition, Aust. Meteorol. Mag., 47, 295–308, 1998.

Elser, M., Huang, R.-J., Wolf, R., Slowik, J. G., Wang, Q., Canonaco, F., Li, G., Bozzetti, C., Daellenbach, K. R., Huang, Y., Zhang, R., Li, Z., Cao, J., Baltensperger, U., El-Haddad, I. and Prévôt, A. S. H.: New insights into PM2.5 chemical composition and sources in two major cities in China during extreme haze events using aerosol mass spectrometry, Atmos Chem Phys, 16(5), 3207–3225, doi:10.5194/acp-16-3207-2016, 2016.

Fialho, P., Freitas, M. C., Barata, F., Vieira, B., Hansen, A. D. A. and Honrath, R. E.: The Aethalometer calibration and determination of iron concentration in dust aerosols, J. Aerosol Sci., 37(11), 1497–1506, doi:10.1016/j.jaerosci.2006.03.002, 2006.

Fomba, K. W., Müller, K., van Pinxteren, D., Poulain, L., van Pinxteren, M. and Herrmann, H.: Long-term chemical characterization of tropical and marine aerosols at the Cape Verde Atmospheric Observatory (CVAO) from 2007 to 2011, Atmospheric Chem. Phys., 14(17), 8883–8904, doi:10.5194/acp-14-8883-2014, 2014.

Formenti, P., Rajot, J. L., Desboeufs, K., Caquineau, S., Chevaillier, S., Nava, S., Gaudichet, A., Journet, E., Triquet, S., Alfaro, S., Chiari, M., Haywood, J., Coe, H. and Highwood, E.: Regional variability of the composition of mineral dust from western Africa: Results from the AMMA SOP0/DABEX and DODO field campaigns, J. Geophys. Res. Atmospheres, 113(D23), D00C13, doi:10.1029/2008JD009903, 2008.

Fröhlich, R., Cubison, M. J., Slowik, J. G., Bukowiecki, N., Canonaco, F., Croteau, P. L., Gysel, M., Henne, S., Herrmann, E., Jayne, J. T., Steinbacher, M., Worsnop, D. R., Baltensperger, U. and Prévôt, A. S. H.: Fourteen months of on-line measurements of the non-refractory submicron aerosol at the Jungfraujoch (3580 m a.s.l.) – chemical composition, origins and organic aerosol sources, Atmos Chem Phys, 15(19), 11373–11398, doi:10.5194/acp-15-11373-2015, 2015.

Hansen, A. D. A.: Aethalometer Operations Manual, Magee scientifique, Berkeley, CA, USA., 2005.

Henry, R., Norris, G. A., Vedantham, R. and Turner, J. R.: Source Region Identification Using Kernel Smoothing, Environ. Sci. Technol., 43(11), 4090–4097, doi:10.1021/es8011723, 2009.

Hossaini, R., Chipperfield, M. P., Saiz-Lopez, A., Fernandez, R., Monks, S., Feng, W., Brauer, P. and von Glasow, R.: A global model of tropospheric chlorine chemistry: Organic versus inorganic sources and impact on methane oxidation, J. Geophys. Res. Atmospheres, 121(23), 2016JD025756, doi:10.1002/2016JD025756, 2016.

Joshi, N., Romanias, M., Riffault, V. and Thévenet, F.: Investigating water adsorption on natural mineral dust particles: A DRIFT and BET theory study, under press, Aeolian Research, 2017.

Journet, E., Balkanski, Y. and Harrison, S. P.: A new data set of soil mineralogy for dust-cycle modeling, Atmos Chem Phys, 14(8), 3801–3816, doi:10.5194/acp-14-3801-2014, 2014.

Kandler, K., SchÜTz, L., Deutscher, C., Ebert, M., Hofmann, H., JÄCkel, S., Jaenicke, R., Knippertz, P., Lieke, K., Massling, A., Petzold, A., Schladitz, A., Weinzierl, B., Wiedensohler, A., Zorn, S. and Weinbruch, S.: Size distribution, mass concentration, chemical and mineralogical composition and derived optical parameters of the boundary layer aerosol at Tinfou, Morocco, during SAMUM 2006, Tellus B, 61(1), 32–50, doi:10.1111/j.1600-0889.2008.00385.x, 2009.

Lafon, S., Rajot, J.-L., Alfaro, S. C. and Gaudichet, A.: Quantification of iron oxides in desert aerosol, Atmos. Environ., 38(8), 1211–1218, doi:10.1016/j.atmosenv.2003.11.006, 2004.

Lafon, S., Sokolik, I. N., Rajot, J. L., Caquineau, S. and Gaudichet, A.: Characterization of iron oxides in mineral dust aerosols: Implications for light absorption, J. Geophys. Res. Atmospheres, 111(D21), D21207, doi:10.1029/2005JD007016, 2006.

Linke, C., Möhler, O., Veres, A., Mohácsi, Á., Bozóki, Z., Szabó, G. and Schnaiter, M.: Optical properties and mineralogical composition of different Saharan mineral dust samples: a laboratory study, Atmos Chem Phys, 6(11), 3315–3323, doi:10.5194/acp-6-3315-2006, 2006.

Moreno, T., Querol, X., Castillo, S., Alastuey, A., Cuevas, E., Herrmann, L., Mounkaila, M., Elvira, J. and Gibbons, W.: Geochemical variations in aeolian mineral particles from the Sahara–Sahel Dust Corridor, Chemosphere, 65(2), 261–270, doi:10.1016/j.chemosphere.2006.02.052, 2006.

Müller, T., Schladitz, A., Massling, A., Kaaden, N., Kandler, K. and Wiedensohler, A.: Spectral absorption coefficients and imaginary parts of refractive indices of Saharan dust during SAMUM-1, Tellus B, 61(1), 79–95, doi:10.1111/j.1600-0889.2008.00399.x, 2009.

Ng, N. L., Herndon, S. C., Trimborn, A., Canagaratna, M. R., Croteau, P. L., Onasch, T. B., Sueper, D., Worsnop, D. R., Zhang, Q., Sun, Y. L. and Jayne, J. T.: An Aerosol Chemical Speciation Monitor (ACSM) for Routine Monitoring of the Composition and Mass Concentrations of Ambient Aerosol, Aerosol Sci. Technol., 45(7), 780–794, doi:10.1080/02786826.2011.560211, 2011a.

Ng, N. L., Canagaratna, M. R., Jimenez, J. L., Zhang, Q., Ulbrich, I. M. and Worsnop, D. R.: Real-Time Methods for Estimating Organic Component Mass Concentrations from Aerosol Mass Spectrometer Data, Environ. Sci. Technol., 45(3), 910–916, doi:10.1021/es102951k, 2011b.

Ocean biology processing group: MODIS Aqua Level 3 Global Daily Mapped 4 km Chlorophyll a. Ver. 6. PO.DAAC, CA, USA., [online] Available from: https://neo.sci.gsfc.nasa.gov/view.php?datasetId=MY1DMM_CHLORA&year=2015 (Accessed 14 June 2017), 2003.

Olmez, I., Sheffield, A. E., Gordon, G. E., Houck, J. E., Pritchett, L. C., Cooper, J. A., Dzubay, T. G. and Bennett, R. L.: Compositions of Particles from Selected Sources in Philadelphia for Receptor Modeling Applications, JAPCA, 38(11), 1392–1402, doi:10.1080/08940630.1988.10466479, 1988.

Petit, J.-E., Favez, O., Albinet, A. and Canonaco, F.: A user-friendly tool for comprehensive evaluation of the geographical origins of atmospheric pollution: Wind and trajectory analyses, Environ. Model. Softw., 88, 183–187, doi:10.1016/j.envsoft.2016.11.022, 2017.

Petzold, A., Rasp, K., Weinzierl, B., Esselborn, M., Hamburger, T., Dörnbrack, A., Kandler, K., Schütz, L., Knippertz, P., Fiebig, M. and Virkkula, A.: Saharan dust absorption and refractive index from aircraft-based observations during SAMUM 2006, Tellus B, 61(1), 118–130, doi:10.1111/j.1600-0889.2008.00383.x, 2009.

Polissar, A. V., Hopke, P. K. and Poirot, R. L.: Atmospheric Aerosol over Vermont: Chemical Composition and Sources, Environ. Sci. Technol., 35(23), 4604–4621, doi:10.1021/es0105865, 2001.

Riffault, V., Arndt, J., Marris, H., Mbengue, S., Setyan, A., Alleman, L. Y., Deboudt, K., Flament, P., Augustin, P., Delbarre, H. and Wenger, J.: Fine and Ultrafine Particles in the Vicinity of Industrial Activities: A Review, Crit. Rev. Environ. Sci. Technol., 45(21), 2305–2356, doi:10.1080/10643389.2015.1025636, 2015.

Sandradewi, J., Prévôt, A. S. H., Szidat, S., Perron, N., Alfarra, M. R., Lanz, V. A., Weingartner, E. and Baltensperger, U.: Using Aerosol Light Absorption Measurements for the Quantitative Determination of Wood Burning and Traffic Emission Contributions to Particulate Matter, Environ. Sci. Technol., 42(9), 3316–3323, doi:10.1021/es702253m, 2008.

Schladitz, A., Müller, T., Kaaden, N., Massling, A., Kandler, K., Ebert, M., Weinbruch, S., Deutscher, C. and Wiedensohler, A.: In situ measurements of optical properties at Tinfou (Morocco) during the Saharan Mineral Dust Experiment SAMUM 2006, Tellus B, 61(1), 64–78, doi:10.1111/j.1600-0889.2008.00397.x, 2009.

Simoneit, B. R. T.: Biomass burning — a review of organic tracers for smoke from incomplete combustion, Appl. Geochem., 17(3), 129–162, doi:10.1016/S0883-2927(01)00061-0, 2002.

Val, S., Liousse, C., Doumbia, E. H. T., Galy-Lacaux, C., Cachier, H., Marchand, N., Badel, A., Gardrat, E., Sylvestre, A. and Baeza-Squiban, A.: Physico-chemical characterization of African urban aerosols (Bamako in Mali and Dakar in Senegal) and their toxic effects in human bronchial epithelial cells: description of a worrying situation, Part Fibre Toxicol, 10(10), 2013.

Vassilev, S. V., Baxter, D., Andersen, L. K. and Vassileva, C. G.: An overview of the chemical composition of biomass, Fuel, 89(5), 913–933, doi:10.1016/j.fuel.2009.10.022, 2010.

Zotter, P., Herich, H., Gysel, M., El-Haddad, I., Zhang, Y., Močnik, G., Hüglin, C., Baltensperger, U., Szidat, S. and Prévôt, A. S. H.: Evaluation of the absorption Ångström exponents for traffic and wood burning in the Aethalometer-based source apportionment using radiocarbon measurements of ambient aerosol, Atmos Chem Phys, 17(6), 4229–4249, doi:10.5194/acp-17-4229-2017, 2017.

[revised manuscript text omitted]

L.-H. Rivellini[1,2,*], I. Chiapello[2], E. Tison[1], M. Fourmentin[3], A. Féron[4], A. Diallo[5], T. N'Diaye[5], P. Goloub[2], F. Canonaco[6], A. S. H. Prévôt[6], and V. Riffault[1,*]

[1] IMT Lille Douai, Univ. Lille, SAGE - Département Sciences de l'Atmosphère et Génie de l'Environnement, F-59000 Lille, France
[1]
[2] Univ. Lille, CNRS, UMR8518 – LOA – Laboratoire d'Optique Atmosphérique, F-59000 Lille, France
[2]
[3] Laboratoire de Physico-Chimie de l'Atmosphère, Université du Littoral Côte d'Opale, Dunkerque, F-59140, France
[4] Laboratoire Interuniversitaire des Systèmes Atmosphériques, CNRS - Université Paris Est Créteil - Université Paris Diderot, Créteil, F-94010, France
[5] Institut de Recherche pour le Développement, M'Bour, Senegal
[6] Laboratory of Atmospheric Chemistry, Paul Scherrer Institute, CH-5232 Villigen, Switzerland

[*] Corresponding authors:

Véronique Riffault

Tel.: +33 327 712 604, Fax: +33 327 712 914, e-mail: veronique.riffault@imt-lille-douai.fr

Laura-Hélèna Rivellini
e-mail: laura.rivellini@imt-lille-douai.fr

[Figure]

**Figure S1.** Values of (a) RF(NO₃), (b) RIE of ammonium, and ratios to ammonium for (c) sulfate and (d) chloride obtained from calibrations performed during a previous campaign ( Zhang et al., in prep.)  and this study ( Nov. 25th 2014 and beyond). (e) Chl and NH₄ calibration curves performed on January 22nd, 2015. (f) time series of CE values corrected following Middlebrook et al. (2012) algorithm colored by the NH₄,meas/NH₄,pred ratio .

**Appendix S2. Deconvolution method uncertainties and comparison with literature data**

The uncertainties on estimated Fe concentrations can be calculated by applying the propagation for uncertainties on the values of $K_{Fe}$ (10%) and the slope b (39%, calculated using a variability of 0.2 for the two Angström absorption exponents (AAE), β and α (Fialho et al., 2006)), which gives an overall uncertainty of ~40%. However this method is highly sensitive to even small variations of α (BC) and β (DD), with values quite well known for BC from fossil fuel ranging from 0.8 to 1.1 (Hansen, 2005; Zotter et al., 2017 and references therein) but not so much for dust. In the manuscript, we chose to use β = - 4, according to Fialho et al. (2006) values determined at the Azores Islands for samples influenced by Saharan dust events. But other values can be found in the literature (Table S2.1), ranging from -1.6 to -6.5 and largely influenced by the wavelength range as well as dust origins and size fractions since the iron content differ depending on emission sources and particle size (Journet et al., 2014). Even during the SAMUM campaign (May to June 2006 in Morocco), a wide range of AAE values have been reported from -1.6 up to -5.1 for ground-based measurements in the same size fraction, as shown in Table S2.1.

**Table S2.1.** Mineral dust AAE values reported from field campaigns around the Saharan region.

| Reference | Location / Period | Wavelengths (nm) | Fraction | β |
|---|---|---|---|---|
| Fialho et al. (2006) [a] | Azores Islands Jul. 2001 – Jun. 2005 | 370-950 | - | -4 |
| Müller et al. (2009) [a] | Tinfou, Morocco (SAMUM) Summer 2006 | 467/660 | $PM_{10}$ | -2.25 to -5.13 |
| Petzold et al. (2009) [b] | South-East Morocco (SAMUM) Summer 2006 | 467/660 | $PM_{2.5}$ | -2 to -6.5 |
| Schladitz et al. (2009) [a] | Tinfou, Morocco (SAMUM) Summer 2006 | 537/637 | $PM_{10}$ | -1.6 to -4.73 |
| (Linke et al., 2006) [c] | Morocco Egypt | 266/532 | $\sim PM_4$ | -4.2 -5.3 |
| (Caponi et al., 2017) [c] | Morocco Lybia Algeria Mali | 375-850 375-532 375-850 375-532 | $PM_{2.5}$ ($PM_{10.6}$) | -2.6 -4.1 (-3.2) -2.8 (-2.5) -3.4 |

[a] In situ ground-based measurements; [b] Airborne measurements through dust plumes; [c] Laboratory experiments with resuspended soil samples

Applying a relatively small increase (resp. decrease) of 10% on the value of β for our dataset led to a 33% decrease (resp. 50% increase) of iron concentrations, as shown in Figure S2.1, but no change in the temporal behavior.

[Figure]

**Figure S2.1.** Box plots of (a) Fe, (b) BC concentrations and (c) Fe/(Fe+Unacc.) ratio for continental, sea breeze and marine days. (d) Scatter plot of iron concentrations (in μg m-3) obtained from Fialho's deconvolution method using an AAE value of ± 10% compared to the one from the literature and used in the manuscript.

Table S2.2 summarizes the iron content determined in Saharan samples, which shows that the relative contribution of iron determined in this work is in the same order of magnitude but still significantly higher. However iron oxides can be found mostly (for ~2/3) in the clay fraction (~$PM_{2.5}$) and ~1/3 in the silt (coarse) fraction (Journet et al., 2014; Kandler et al., 2009), which is consistent with increased ratios in the submicron fraction compared to larger ones. It is also worth noting that Val et al. (2013) measured the iron content in the ultrafine and fine fractions (corresponding to $PM_1$) of particles collected in Dakar, and measured a ratio in the upper range of those already reported in the literature, even in the absence of dust event.

The approach used here leads to an estimate of the absolute concentrations of iron, although with high uncertainties given all the necessary assumptions and the empirical algorithm used to deconvolve BC and Fe from absorption measurements. However the temporal profiles, non-parametric wind regression (NWR) plots and potential source contribution function (PSCF) maps (now provided in Figures S5b and S5c, respectively) are all consistent with the expected behavior of such a desert dust tracer and show that it can be useful in determining the contribution of dust to absorption measurements. There is nonetheless quite some room for improvement, in particular for a better estimation of the AAE value for dust similar to the efforts carried out to determine the AAE values for BC from fossil fuel and wood burning (Zotter et al., 2017). We strongly believe the lack of information for submicron particles in terms of chemical composition of refractory species and optical properties should be better addressed, but is beyond the scope of this work.

**Table S2.2: Comparison of iron content (in %) determined in Saharan dust and soil samples**

| Reference | Location | Method [a] | Size fraction | %Fe [b] |
|---|---|---|---|---|
| Dust samples | | | | |
| (Lafon et al., 2004) | Banizoumbou (Niger) | XRF; CBD | TSP | 6.3; 7.8 |
| (Lafon et al., 2006) | Banizoumbou, | XRF; CBD | TSP | 4.3 – 6.1 |
| (Lafon et al., 2006) | Cape Verde | XRF; CBD | TSP | 5.3 – 6.0 |
| (Formenti et al., 2008) | Banizoumbou | CBD | 40 µm | 5.8 |
| (Val et al., 2013) | Dakar (Senegal) | ICP-MS | 1 µm | 7.8 |
| This work | M'Bour | cf. text | 1 µm | 23 (continental) 21 (sea breeze) 16 (marine) |
| Soil samples | | | | |
| (Moreno et al., 2006) | Saharan region (9 samples) | ICP-AES/ ICP-MS | TSP | 2.0 – 4.7 |
| (Lafon et al., 2006) | Banizoumbou, | XRF; CBD | 10.2 µm [*] 2.5 µm [*] | 5.3 5.8 |
| (Joshi et al., 2017) | M'Bour, Bordj (Algeria), Nefta (Tunisia) | XRD | 100 µm | < 0.5 |

[a] XRF: X-ray Fluorescence (XRF) Spectrometry for elemental analysis; CBD: chemical method based on citrate-bicarbonate-dithionite (CBD) reagent for quantification of iron oxides adapted from soil analysis (Mehra and Jackson, 1960)

[b] Percentages of iron relative to the mass of all oxides, classically taking into account $Na_2O$, $MgO$, $Al_2O_3$, $SiO_2$, $K_2O$, $CaO$, $TiO_2$ and $Fe_2O_3$.

[*] Soil samples resuspended using wind tunnel and collected with a 13-stage impactor

[Figure]

**Figure S23.** Averaged PM$_1$ chemical composition for (a) IOP-1 (n = 2952), (b) continental (n = 307), (c) sea breeze (n = 799) and (d) marine days (n = 1846). Only days with at least 50% of the total PM$_1$ mass concentration measurements by TEOM-FDMS were taken into account for averaging, corresponding to 11 days for continental, 21 days for sea breeze and 42 days for marine days. Unacc.: unaccounted fraction determined as the difference between the gravimetrically measured PM$_1$ mass concentration and the sum of chemical species from ACSM and aethalometer measurements

[Figure]

**Figure S3.** (a) Scatter plot between measured and predicted $NH_4$ colored by relative humidity and (b) associated rose plot, (c) $NH_{4,meas}/NH_{4,pred}$ ratios as a function of $SO_4$, $NO_3$, OM and Chl species, where OM and Chl data are colored by BC concentrations. The red ellipses highlight the data points deriving from the 1:1 ratio corresponding to aerosol neutralization.

[Figure]

**Figure S5.** (a) 48-hour back trajectory clusters for (from left to right) IOP-1, continental, sea breeze and marine days. (b) NWR plots (input parameters: angular and radial resolution of 0.1, angle smoothing of 2 and radial smoothing of 1; upper limit of the color scale: 75th percentile) for PMF factors and NR-PM$_1$, BC and Fe species and corresponding (c) PSCF maps for species showing regional influence (threshold: 75th percentile) during IOP-1.

**Appendix S6. Unconstrained PMF analysis**

(a) 3-factor solution

(b) 4-factor solution

(g) 9-factor solution

(h) 10-factor solution

[Figure]

**Figure S6.1.** PMF unconstrained solutions from 3 to 10 factors, with (left) factors profiles and (right) corresponding daily cycles.

[Figure]

**Figure** S6.. 4-factor PMF unconstrained solutions with (a) factor profiles, pie chart and time series for IOP-1; and factor profiles and pie charts for (b) continental (Q/Qexp = 0.37), (c) sea breeze (Q/Qexp = 0.36) and (d) marine (Q/Qexp = 0.27) days .

[Figure]

**Figure S4S6.23.** (top) Mass spectrum (inset: m/z ≥ 60), daily profile and rose plot of the LCOA factor from the unconstrained 4-factor solution; (bottom) Time series and rose plots of fragments at m/z 58, 60, 83 and 91

[Figure]

**Figure** S7. Box plots of (left)  the m/z 35, 36 and 60 signals and (right) of m/z 60/OM (f$_{60}$) ratio. For each box plot, top line: 75$^{th}$ percentile, bottom line: 25$^{th}$ percentile, middle line: 50$^{th}$ percentile (median); top whisker: 95$^{th}$ percentile, bottom whisker: 5$^{th}$ percentile. The open circle and the cross represent the 99$^{th}$ percentile and maximum value, respectively.

**Appendix S8. PMF 5-factor solution including organics + m/z 36 as input data**

Since the behavior of Chl had also been suspected to come from the same sources, m/z signals at 35 and 36 were investigated in order to possibly implement them in the model input. However the m/z 35 signal presented an important amount of slightly negative values (-3.0 ± 6.2 × 10-13, see Figure S7) which likely resulted from a slow vaporization of refractory chloride species both during filter and non filter measurement as previously observed (Nuaaman et al., 2015). For this reason only m/z 36 was incorporated into the model without additional normalization since the signal intensity was close to organic ones. Uncertainties were estimated as followed.

The detection limits ($DL_x$) for these m/z were assumed to be equal to 3 times their respective signal-to-noise ratio for filtered air. The method to determine the uncertainties has already been used to carry out source apportionment studies based on filter data (Tauler et al., 2009; Jang et al., 2013). When the mass concentrations were below the detection limit, concentrations $C_x$ were replaced by $DL_x/2$ and the uncertainties calculated by Equation 1:

$$S_x = 0.2 \times C_x + LD/3 \qquad \text{(Eq. 1)}$$

If the concentrations were above the detection limit, Equation 2 was used:

$$S_x = 0.1 \times C_x + LD/3 \qquad \text{(Eq. 2)}$$

New unconstrained runs of the PMF model using the combined dataset of OM plus HCl+ signal for IOP-1 led to the almost complete (95%) attribution of the m/z 36 signal to the Local Combustion OA (LCOA), where it represented 40% of the total factor mass. Besides, in order to refine the solutions, and due to the possible specificity of local emissions, the PMF model was run with constraints on the primary factor profiles, that is to say LCOA obtained from the IOP-1 solution, and COA and HOA from the sea breeze solution, using the a-value approach with 10% freedom (a = 0.1).

[Figure]

**Fig. S8**. PMF constrained 5-factor solution including the m/z 36 chloride peak: (left) factor profiles of LCOA, COA, HOA (all primary factors constrained), MO-OOA, LO-OOA; (middle) corresponding daily cycles according to day types (solid lines: median; dotted lines: average); and (right) pollution rose plots colored by hour of day. (bottom) Average pie charts of the contributions to the total organic fraction for IOP-1, continental, sea breeze and marine days

[Figure]

**Fig. S7S9**. (a) Diurnal average profile of BC/OM ratio for continental, sea breeze and marine days and scatter plot of BC vs OM concentrations (in µg m⁻³) for (b) IOP-1, (c) continental, (d) sea breeze and (e) marine days.